# ARViS: a bleed-free multi-site automated injection robot for accurate, fast, and dense delivery of virus to mouse and marmoset cerebral cortex

Shinnosuke Nomura [1,2], Shin-Ichiro Terada [1], Teppei Ebina [1], Masato Uemura[1], Yoshito Masamizu[3,4], Kenichi Ohki [1,5,6] & Masanori Matsuzaki [1,3,5,7] ✉

Genetically encoded fluorescent sensors continue to be developed and improved. If they could be expressed across multiple cortical areas in non-human primates, it would be possible to measure a variety of spatiotemporal dynamics of primate-specific cortical activity. Here, we develop an Automated Robotic Virus injection System (ARViS) for broad expression of a biosensor. ARViS consists of two technologies: image recognition of vasculature structures on the cortical surface to determine multiple injection sites without hitting them, and robotic control of micropipette insertion perpendicular to the cortical surface with 50 μm precision. In mouse cortex, ARViS sequentially injected virus solution into 100 sites over a duration of 100 min with a bleeding probability of only 0.1% per site. Furthermore, ARViS successfully achieved 266-site injections over the frontoparietal cortex of a female common marmoset. We demonstrate one-photon and two-photon calcium imaging in the marmoset frontoparietal cortex, illustrating the effective expression of biosensors delivered by ARViS.

Genetically encoded calcium indicators (GECIs) have revolutionized the fields of neuroscience and functional biology[1–3]. Furthering their use, the recent development of other genetically encoded biosensors and optogenetic and chemogenetic tools has opened opportunities to study the spatiotemporal dynamics of neuronal and glial activity and synaptic transmission[4–7]. If these probes could be expressed uniformly across multiple broad cortical areas in non-human primates (NHP), it would be possible to measure and manipulate the spatiotemporal dynamics of the inter-areal spreading of a variety of activities

occurring in motor, cognitive, and sensory processing, activities that are well developed in primates[8–10]. In addition, the fluorescence intensity change in these sensors can be detected at the resolution of a single-neuron, and sometimes at a subcellular resolution.

To obtain cortex-wide imaging of fluorescence intensity changes in these sensors with a high signal-to-noise ratio, it is necessary that they are expressed broadly, uniformly, and strongly. The most common method for achieving this is the injection of adeno-associated virus (AAV) carrying the sensor gene into multiple sites. AAV-mediated

[1]Department of Physiology, Graduate School of Medicine, The University of Tokyo, Tokyo 113-0033, Japan. [2]International Institute for Integrative Sleep Medicine (WPI-IIIS), University of Tsukuba, Tsukuba, Ibaraki 305-8575, Japan. [3]Brain Functional Dynamics Collaboration Laboratory, RIKEN Center for Brain Science, Saitama 351-0198, Japan. [4]Laboratory of Functional Brain Circuit Construction, Graduate School of Brain Science, Doshisha University, Kyoto 610-0394, Japan. [5]International Research Center for Neurointelligence (WPI-IRCN), The University of Tokyo Institutes for Advanced Study, Tokyo 113-0033, Japan. [6]Institute for AI and Beyond, The University of Tokyo, Tokyo 113-0033, Japan. [7]Department of Biological Sciences, Graduate School of Science, The University of Tokyo, Tokyo 113-0033, Japan. ✉e-mail: mzakim@m.u-tokyo.ac.jp

GECI expression has been successfully demonstrated in macaque and common marmoset cortex[11–21]. Although a GECI-expressing transgenic marmoset was created, its expression level was very weak[22]; in general, gene expression in transgenic animals is weaker than high-titer virus-mediated gene expression because of a smaller copy number of the transgene in the former method than in the latter method, and inhibitory positional effects and repeat-induced gene silencing in the host genome[23]. As an alternative, brain transfection of AAV variants such as AAV-PHP.eB and AAV.CAP-B10 through systemic transvenous administration is more promising[24]. However, transvenous transfection-mediated gene expression in the NHP brain is not sufficiently high for practical use of GECIs. Since transvenous transfection induces expression in other organs, such as liver, it might cause unintended malfunctioning of these organs, thereby limiting the viral dose and gene expression that can be obtained. Therefore, although molecular biologists are struggling to solve these problems, we need to adopt a method involving multiple local injections of the virus solution to image the sensor in NHPs.

Since the cortical area in which the gene should be expressed is broader in NHPs than in rodents, the number of injection sites and the total time for injections are greater in NHPs than in rodents. To make the injection procedure easier and more effective, automation using robotics could be a solution. To achieve this goal, two major problems need to be addressed. First, blood vessels should be avoided at the injection sites to prevent bleeding. Second, it is necessary to insert a glass micropipette filled with the virus solution into the target site perpendicular to the cortical surface to prevent slipping or bending of the pipette on the cortical surface and deformation of the cortical tissue[25]. Although the technology for automated recognition of blood vessels has advanced[26–28], it has rarely been used in real time immediately before surgery on the target tissue. Although micropipette insertion methods for in vivo patch-clamp recording have been automated, the insertion target is determined by the experimenter or by real-time imaging, and these methods do not have the ability to automatically avoid blood vessels[29–31]. For the purpose of deep brain stimulation therapy, CT- and MRI-assisted robotic technology has been developed for the automated insertion of electrodes into the brain while avoiding important intracranial structures such as nerve nuclei and ventricles[32]. Although this system is suitable for deep brain intervention, the misalignment range is approximately 1 mm from the target[33]. Therefore, insertion into the brain parenchyma is usually performed manually, and hemostatic treatment is necessary.

In this study, we developed an Automated Robotic Virus injection System (ARViS) that enables bleed-free multi-site automated injections for accurate, fast, and dense virus delivery into the cerebral cortex of living mice and common marmoset. We solved the first problem of blood vessel avoidance by precisely measuring the three-dimensional (3D) structure of the cortical surface in these animals and applying deep learning-based image recognition of the vasculature structures with clustering-based optimization of multiple injection sites. We solved the second problem of perpendicular insertion by manipulating the micropipette with a six degrees-of-freedom robot under high-precision calibration. Bleeding probability after injection with ARViS was 0.1% in seven mice and 0% in one marmoset. To demonstrate the imaging of ARViS-assisted expression of the GECI, we imaged different neuronal representations of orofacial and body movements over a 14 × 7 mm area of the frontoparietal cortex in an awake adult marmoset.

## Results
### Workflow of ARViS
ARViS consists of two technologies: robotic technology to precisely, smoothly, and sequentially insert a micropipette into multiple virus-injection sites in the cortex (Fig. 1a–d), and image recognition technology to determine multiple virus-injection sites in the cerebral

cortex without hitting blood vessels on the cortical surface (Fig. 1e–i). The baseplate of a hexapod robot that had six degrees of freedom was fixed on the desk parallel to the vertical plane. The stage of the robot was attached to an injector connected to a micropipette, a laser distance sensor to measure the 3D cortical surface shape, and a CMOS camera to image cortical blood vessels (Camera S) (Fig. 1a). Anesthetized animals with the skull removed were set in a head-fixation apparatus on the base plate. The cortical surface and injection sites were determined in the XYZ coordinates in the robot base system frame {B} (Fig. 1a, c, i), and the micropipette filled with solution was inserted into the cortical tissue at each site by a combination of XYZ movements, V rotation (along the Y axis), and W rotation (along the Z axis), under robotic control.

### Deep learning-based vessel segmentation
In the first step of the image recognition, the 3D cortical surface (an area over 8 × 8 mm centered on the bregma) of the anesthetized mouse was accurately scanned with the laser distance sensor. On the basis of this surface scanning data, the distance between Camera S and the cortical surface was automatically adjusted and finely-focused surface images were acquired at 62 points over the neocortex (Supplementary Fig. 1a). Next, the blood vessels within a cropped area of each image (approximately 3.8 × 3.8 mm; Figs. 1f, 2a and Supplementary Fig. 1b) were segmented using a deep learning-based convolutional neural network (CNN) named Spatial Attention-UNet (SA-UNet)[34], which was originally designed for the segmentation of retinal fundus vasculature (Supplementary Fig. 2a). We initially trained the neural network using twenty images of the retinal fundus vasculature from the Digital Retinal Images for Vessel Extraction (DRIVE) dataset (DRIVE model) (Supplementary Fig. 2b, c). The DRIVE model worked well for relatively narrow vessels, but not for thick vessels (Fig. 2b), which might reflect the fact that the widths of the vessels on the mouse cortex range from less than 10 μm to more than 200 μm, thereby differing substantially from retinal vessels. Therefore, we re-trained the SA-UNet using an additional six images of cortical vessels in mouse brain, with vessel labels that were determined by the experimenter (initial model) (Supplementary Fig. 2d, e). Although this addition improved the detection of thick vessels, some parts of them remained undetected (Fig. 2b). To add more effective mouse cortical images (27 images) to the training dataset, we adopted EquAL[35], which calculates the difference in prediction between the original image and its left-right inverted image as the "consistency" of the inference, and used CNN-corrected segmentation that uses active learning[36]. This CNN-corrected segmentation reduced the time for the EquAL data labeling by approximately one-third (Supplementary Fig. 2f). An SA-UNet that was trained with the DRIVE dataset and 33-mouse dataset (final model) reliably detected thick vessels, even the thickest one in the midline of the cortex (Figs. 1f, 2b and Supplementary Fig. 2g, h; see Methods for details).

We evaluated the segmentation performance of SA-UNets trained with four different datasets: the DRIVE model, the initial model, SA-UNet trained with the 33-mouse dataset (non-DRIVE model), and the final model. To compare the segmentation performance between these four models, we used three images that were common across all four models as the test dataset (Supplementary Fig. 2i, j). Then, we compared four metrics (accuracy, AUC, F1, and Matthews correlation coefficient [MCC]; see legend of Supplementary Fig. 2k for details) that are known as good indicators for evaluating model performance. The final model showed the best accuracy, F1, and MCC, and the second best AUC (Supplementary Fig. 2k). However, this second best AUC value for the final model was almost the same as the highest AUC value obtained for the non-DRIVE model (Supplementary Fig. 2k). Thus, we considered that the final model was the best of the four models, and used this final model for all subsequent experiments.

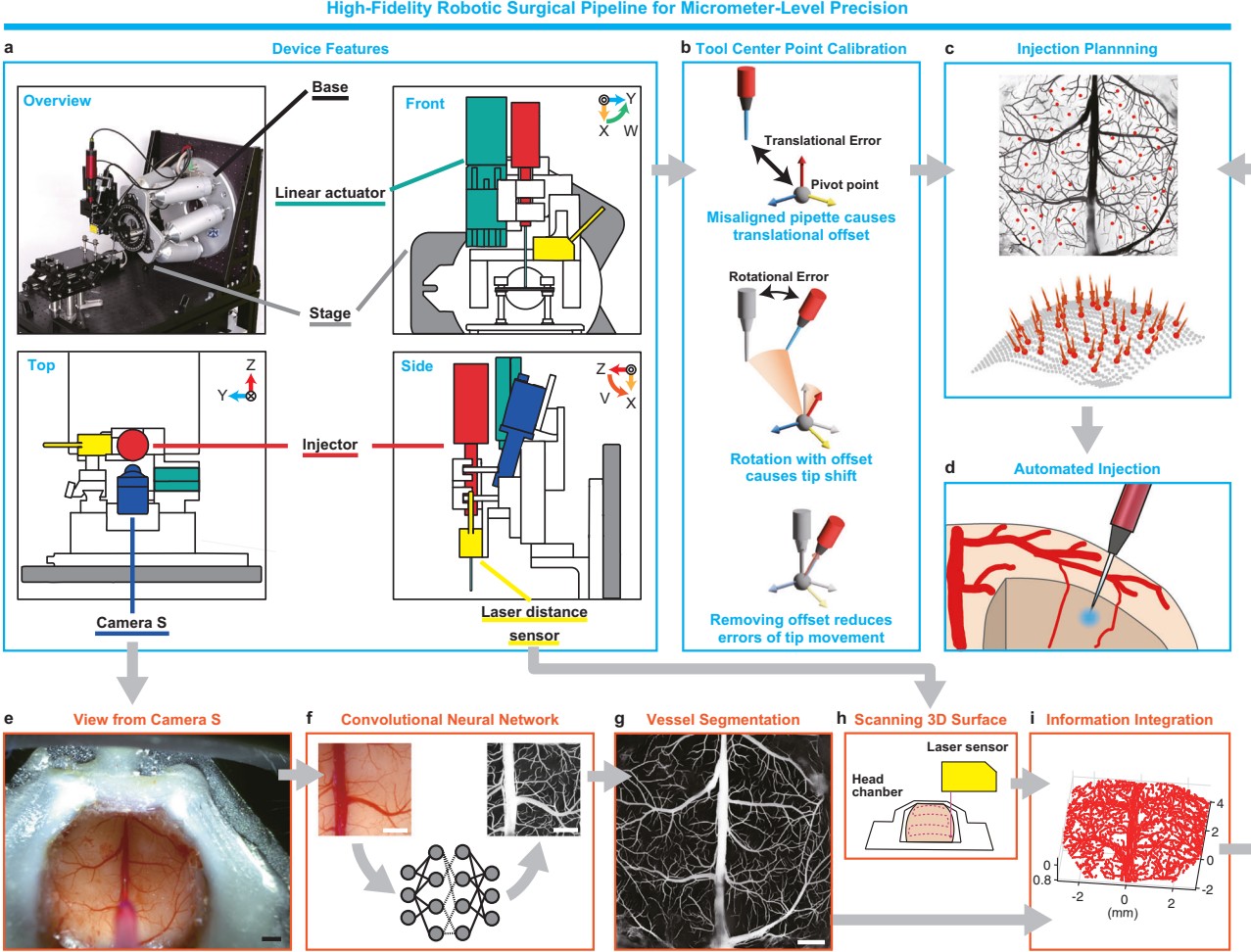

**Fig. 1 | Workflow of an automated injection. a** Device picture and schematic of the device features. An injector (red), a linear actuator (green), a laser distance sensor (yellow), and Camera S (blue) were mounted on the stage of a six-axis hexapod. The animal's head was fixed in the apparatus on the base plate (the animal is not shown). Light orange, blue, and red arrows shown in the upper right of each inset represent the X, Y, and Z axes in the robot base system frame, respectively. The V (orange) and W (green) rotation directions are also shown. **b** Schematic of tool center point calibration. Top, any misalignment of the pipette tip position from the pivot point can cause an unintended translational offset from the desired position. Middle, when the pipette rotates around the pivot point, this offset can induce an unexpected shift in the position of the pipette tip. Bottom, minimizing this offset is essential for 3D manipulation of the micropipette. **c**, A representation of multiple injection sites (red points) on the mouse cortical surface with vessel segmentation (black) and the pipette insertion directions (red lines in the bottom). The injection site positions and pipette angles are designed based on data from the surface recognition pipeline (**e–i**). **d** Illustration of the automated injection into the target point within the cortical tissue. **e** A representative image of the mouse cortical surface captured by Camera S shown in (**a**). Scale bar, 1 mm. **f** Schematic of the segmentation of blood vessels on the cortical surface with a convolutional neural network. An example pair of input and output images are shown on the left and right, respectively. Scale bars, 1 mm. **g** A representative stitched image of the vessel segmentation over the whole dorsal cortex of a mouse. Scale bar, 1 mm.
**h** Schematic of 3D cortical surface scanning with the laser distance sensor. **i** An example of the projection of the vessel pattern onto the 3D cortical surface. Source data are provided as the Source Data file.

To determine the minimum size of the extracted blood vessels, we compared the model-segmented vessels with vessel structures that were manually determined from cortical images taken at a higher magnification under the microscope (Fig. 2c, d). Vessels of ≥30 μm width were recognized by the model-based segmentation with 100% accuracy, whereas vessels with approximately 20 μm width were recognized with nearly 80% accuracy (Fig. 2e). We assumed that if we avoided vessels with a width of approximately 20 μm or more, then serious bleeding would not occur when the injections were performed.

### Determination of multiple injection sites in the two-dimensional vessel segmentation image

The vessel segmentation images were stitched together to make one image covering the whole dorsal cortex (left in Fig. 3a; original stitched image). However, the angle of the camera was tilted by approximately

16° against the optical axis of the laser (Fig. 1a). This meant that the greater the angle between the optical axis of Camera S and the perpendicular line of the cortical surface, the more the imaged vessel thickness and position deviated. Thus, the original stitched image needed to be mapped onto the 3D cortical surface anew. The reference points on the 3D cortical surface that were estimated from the laser scanning were projected onto a two-dimensional (2D) image at a given angle and a geometric transformation was performed in this projected image to minimize the difference in the reference point positions between the transformed projected image and the original stitched image. Then, the angle with the minimal difference in the reference point positions was regarded as the angle between the optical axes of the laser distance sensor and Camera S. The vessel segmentation image in the original stitched image was then transformed to the image on the 3D surface shape according to this angle (middle in Fig. 3a).

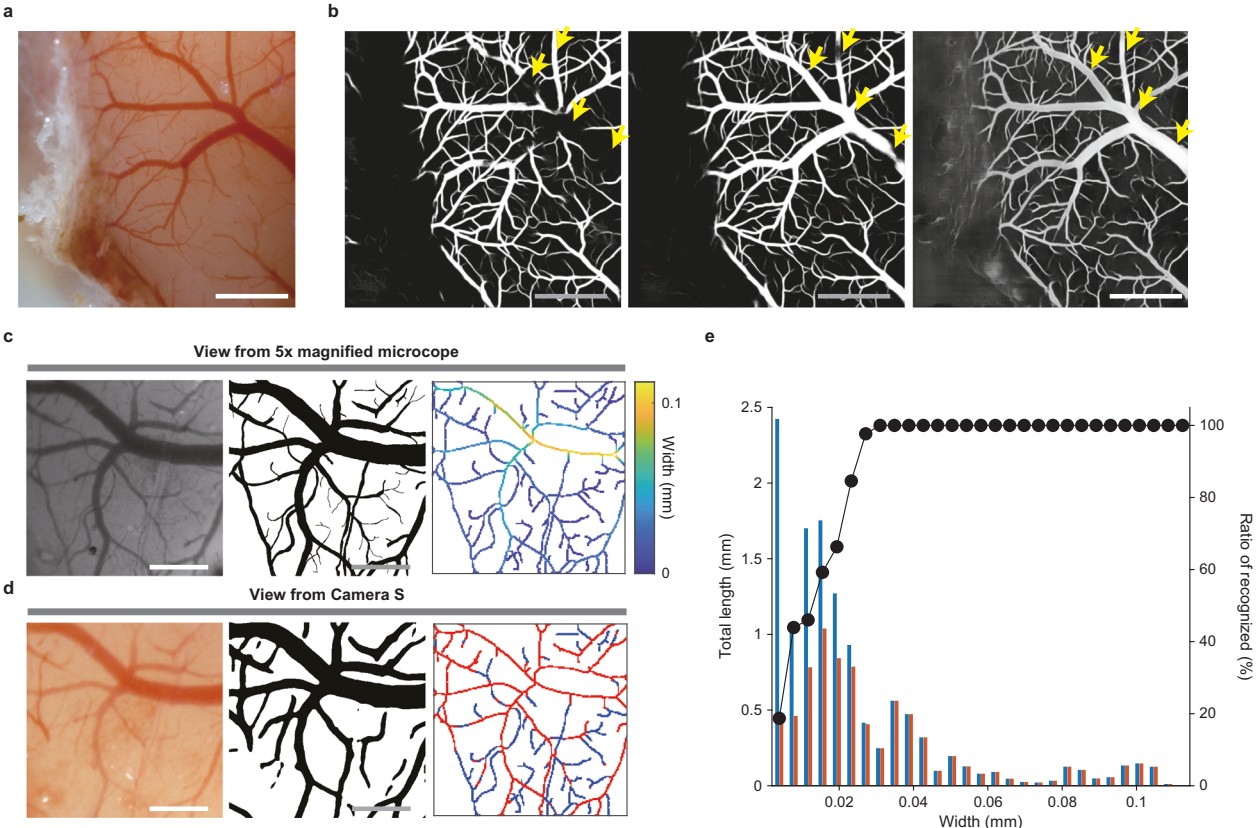

**Fig. 2 | Vessel segmentation on the cortical surface. a** An example input image of the mouse cortical surface for SA-UNet. Scale bar, 1 mm. **b** Example output images of vessel segmentation in the DRIVE model (left), the initial model (middle), and the final model (right). The input image is the image shown in (**a**). Scale bar, 1 mm. Some parts of thick vessels were only segmented in the final model (arrows). **c** Left, an example image of the cortical surface captured by the microscope with a 5× magnification objective. Middle, manually created vessel segmentation of the left image. Right, skeletonized blood vessel mask of the middle image. The color of the skeleton corresponds to the width of the vessel at the corresponding point. Scale bar, 0.3 mm. **d** Left, an image captured by Camera S. The imaged area is the same as in (**c**). Middle, a vessel mask of the left image generated by the final model. Right, skeletonized blood vessel mask of the middle image. Vessels overlaid by the mask in (**c**) are shown in red, while the rest are in blue. Scale bar, 0.3 mm. **e** Histogram of the total lengths of the vessels that were detected on the 5× magnification image (blue) and those that were recognized by the final model (red) for each bin width. A black plot for each bin width indicates the ratio of the latter length to the former length ($n$ = 1 mouse). Source data are provided as the Source Data file.

When the 3D segmentation image was projected into the YZ plane to create a 2D image (YZ stitched image; right in Fig. 3a), differences in vessel structure between the original stitched image and YZ stitched image were apparent around the areas where the curvature of the cortical surface was prominent (Fig. 3b and Supplementary Fig. 1a). Thus, we considered that the image transformation procedure worked well.

Next, we used the YZ stitched image to determine multiple injection sites over the dorsal cortex. It was not possible to insert the pipette into each target injection site outside of the blood vessels without any error. To prevent it from hitting a blood vessel, we designated the cortical region that was a certain distance away from the blood vessels (safety margin distance) as the safety region. To distribute multiple injection sites as evenly as possible within the safety regions, these regions were divided into a number of clusters according to the k-means method[37]. The centroid of each cluster was chosen as a candidate injection site (Fig. 3c). After this process was repeated approximately 10 times to remove the sites that were very close to their nearest neighbors, the target injection sites were determined. When the safety margin distance was set at approximately 65 μm and the number of final injection sites was 97 ± 11.8 ($n$ = 7 mice), the distance from the nearest neighbor sites was 0.360 ± 0.005 mm and the distance from the nearest blood vessel was 0.100 ± 0.001 mm ($n$ = 679 sites from seven mice) (Fig. 3d, e). Although the injection site could not be set in highly vascularized areas without any safety region, if there was a small safety region within highly vascularized areas, the injection site could be placed there, allowing such areas to be covered to some extent (Supplementary Fig. 3).

It took approximately 17 min to finish the laser scanning and take photographs of the cortical surface, perform vessel segmentation, create the YZ stitched image, and determine the target injection sites (Supplementary Table 1).

### High-precision tool point calibration for 3D manipulation of the pipette tip with the hexapod robot

As described, it is necessary to control the robot to precisely translate and rotate the pipette tip to each target injection site on the cortical surface of the anesthetized animal. Since a pipette must be connected for every injection experiment, the position of the pipette tip should be calibrated for each experiment. The problem of how to find the coordinates of a tool tip added to a robot is generally referred to as the tool center point (TCP) calibration problem (Fig. 1b). The traditional approach to the TCP calibration problem is to manually position the needle tip at various angles to align it with a mark[38]. However, this method is labor-intensive and lacks the precision needed for the current study. In addition, the use of fragile glass micropipettes in our study required a non-contact methodology. Therefore, for the initial step in the calibration we adopted a non-contact binocular-vision-based approach[39].

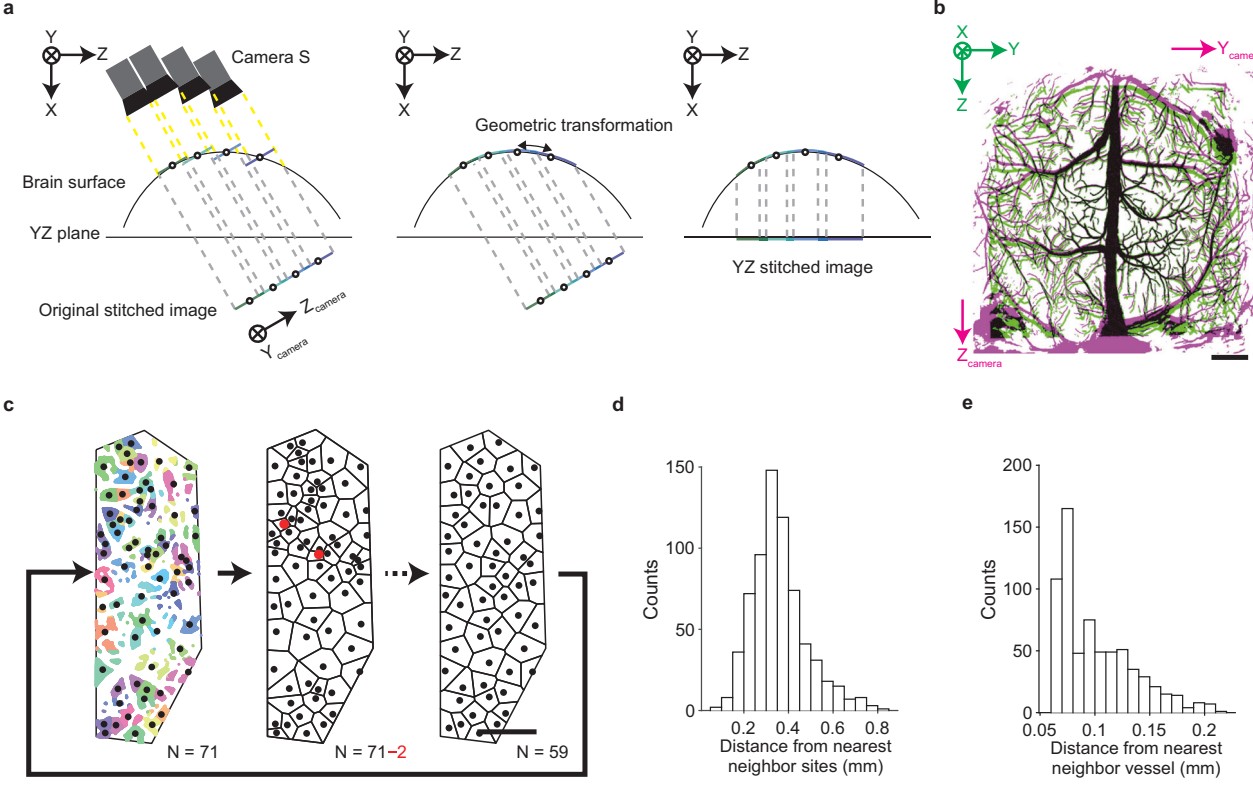

**Fig. 3 | Determination of multiple injection sites in the two-dimensional vessel segmentation image. a** Schematic of the transformation process from the original stitched image to the YZ stitched image. Left: Each image was acquired by adjusting the X position so that the position of the reference point where the pipette tip should be imaged matched the cortical surface, and these images were then stitched to form the original stitched image ($Y_{camera}$-$Z_{camera}$ plane). Circles on the cortical surface indicate the reference points and circles on the stitched images are their projection. Middle: By inferring the angle between the camera and laser optical axes, the original stitched image was mapped to the corresponding cortical surface with geometric transformation. Right: The image mapped on the cortical surface was projected to the YZ plane as the YZ stitched image. The camera angle and the curvature of the cortical surface are more emphatic than factual. **b** Representative overlay of the vessel segmentations of the original stitched image (magenta) and

the YZ stitched image (green). Scale bar, 1 mm. **c** Diagram for the injection site planning algorithm. The initial number ($n = 71$ in this mouse) of sites (black dots) within the safety regions (colors) were determined according to the k-means clustering method. Different colors indicate different clusters. White areas indicate the safety margin region (with a safety margin distance of 65 μm in this mouse) and blood vessels. Next, two sites whose cluster areas including blood vessels were the smallest and second smallest (red dots) were removed and one new site was added. After these procedures were repeated approximately 10 times (eleven times in this mouse), the target injection sites were determined. Scale bar, 1 mm. **d** Histogram of the distance between the target injection site and the corresponding nearest neighbor site ($n = 679$ sites from seven mice). **e** Histogram of the distance between the target injection site and the corresponding nearest blood vessel ($n = 679$ sites from seven mice). Source data are provided as the Source Data file.

First, we set two cameras (Camera R and Camera L) to capture the pipette tip from the side. Then, the three-dimensional position of the pipette tip in the binocular camera coordinate system determined by these two cameras was inferred (Fig. 4a, b). Frame {C} is the binocular camera coordinate system frame. Next, the coordinates of the pipette tip in frame {C} are transformed to those in the robot base system frame {B} (Fig. 4c). In the robot coordinate system, the "pivot point", which serves as the effector point, can be set as the origin in the effector coordinate system (Fig. 4c). Frame {E} is the effector frame. When the pipette tip and the pivot point are completely matched, the translation of the pivot point should move the pipette tip the same distance and the rotation around the pivot point should rotate the pipette tip by the same angle without any change in its position (Fig. 1b). To make this match as precise as possible, the translation and rotation of the pipette tip under the control of the robot were repeated, its position at each movement end was measured with Camera L and Camera R, and then, the coordinate transformation matrix was determined to minimize the deviation of the pipette tip coordinates between frames {C} and {B} (see Methods for details).

Although this calibration was able to bring the pipette tip close to the pivot point, some errors remained when translating or rotating the pipette tip. The first error was the error derived from the pipette

detection on the captured images and camera calibration (Fig. 4d). To more accurately estimate the coordinate transformation between frames {C} and {B} in the data with errors at many pipette-tip points, we implemented a robust estimation technique, the M-estimator Sample Consensus (MSAC) algorithm[40]. As a result, it was possible to estimate the pipette tip movement from the camera images with error of less than 15 μm for translational motion of 0.5 mm (Fig. 4e, f).

The second error was caused by the robotic rotation. We noticed that the center point of the rotation could be shifted by a few hundred micrometers and that this shift depended on the rotation angle (Fig. 4g and Supplementary Fig. 5a–d). To reduce the shift, we first used a voxel grid search to narrow down the deviation of the position of the pipette tip predicted from the pivot point (Supplementary Fig. 5e–h). As a result, the shift of the pipette tip in a rotation pattern (V of −7° and W of −15°) was reduced to less than 0.05 mm (Fig. 4h). However, a maximum shift of 0.3 mm occurred with other rotation angles (Fig. 4h). Therefore, we inferred the distribution of the shift in the V-W space by combining the grid research and regression model with a thin plate spline method[41] (Fig. 4i; see Methods for details). By compensating for the shift predicted by this model, 98.9% of the actual shift of the pipette tip position was reduced to less than 50 μm, and the average shift was $0.018 \pm 0.001$ mm ($n = 175$) (Fig. 4j, k and Supplementary Movie 1).

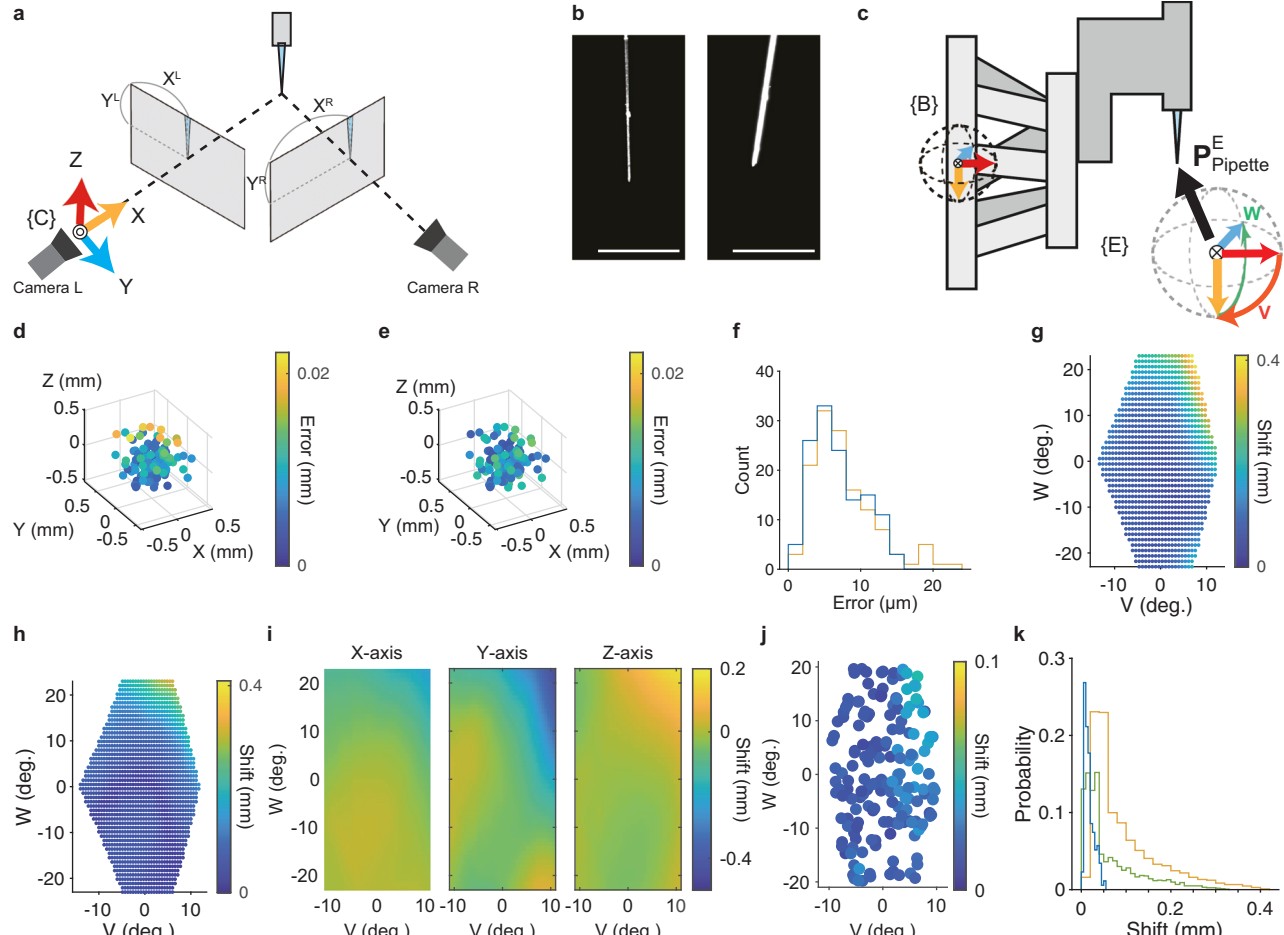

**Fig. 4 | Tool center point (TCP) calibration. a** Schematic of a binocular camera system for the 3D reconstruction of the pipette tip coordinates. 3D coordinates of the pipette tip are inferred from two 2D coordinates from the Camera L image ($X^L$, $Y^L$) and Camera R image ($X^R$, $Y^R$). **b** Example images of the glass micropipette captured by Camera L (left) and Camera R (right). The images were filtered, and the contrast was adjusted. Scale bar, 1 mm. **c**, Coordinate system used for solving the tool center point calibration problem. $P^E_{Pipette}$ is a vector that indicates the initial position of the pipette tip in frame {E}. Light orange, X axis; cyan, Y axis; red, Z axis. The directions of V (orange) and W (green) rotations are also shown. **d**, **e** Representative plots of estimated pipette tip coordinates relative to the original tip coordinates before (**d**) and after (**e**) MSAC algorithm implementation. Only the coordinates regarded as inliers with the MSAC algorithm are shown for comparison ($n = 131$). The pseudo-color bar indicates the distance between the estimated pipette movement and the scheduled movement of the robot. **f** Histogram of the distance between the estimated pipette movement and scheduled movement of the robot without (orange) and with (blue) the MSAC algorithm. **g** A representative map of the pipette tip shift resulting from specific V and W rotation patterns before compensation for rotation error. **h** A representative map of the pipette tip shift resulting from specific V and W rotation patterns after the first part of the compensation for rotation error. **i** X- (left), Y- (middle), and Z- (right) axial maps of the pipette tip shift of the map shown in (**h**) after the interpolation using the thin-plate spline method. **j** A map of the pipette tip shift resulting from specific V and W rotation patterns after the compensation for rotation error. **k** Histogram of the pipette tip shift for rotations with only conventional TCP calibration (orange), with grid searching after TCP calibration (green), and with all methods including the compensation (blue). Source data are provided as the Source Data file.

When the XYZ position of the pipette tip was moved within a $3 \times 8 \times 8$ mm space, which covered the mouse dorsal cortex (Supplementary Fig. 4a–c), and the rotation pattern was changed within a V range from −10° to 5° and a W range of ±20°, 99.8% of the shift caused by the rotation was less than 50 μm and the average shift was $0.021 \pm 0.0002$ mm (mean ± s.e.m., $n = 1274$) (Supplementary Fig. 5i, j).

It took 80–100 min to finish all processes of the TCP calibration (Supplementary Table 1). All codes for the image recognition and robotic technology are listed in Supplementary Table 1 and are publicly available at (https://github.com/nomurshin/ARViS_Automated_Robotic_Virus_injection_System/wiki). This Wiki also provides a full parts list, assembly guide, and workflow instructions.

### Validation of the precision of ARViS in the mouse neocortex
Next, we estimated the accuracy of the pipette insertion in the mouse cortex. We injected a fluorescent dye from the pipette tip into the mouse cortex and traced the pipette trajectory according to the fluorescence signal on post-fixed brain slices (Fig. 5a). In this experiment, the target depth of insertion was set to 0.5 mm from the cortical surface. The injection depth was estimated at $0.50 \pm 0.01$ mm (mean ± s.e.m., $n = 32$ sites from one mouse) (Fig. 5b) and the absolute misalignment was $0.05 \pm 0.01$ mm (Fig. 5c). The lateral distance from each target injection site to the actual point of insertion was $0.038 \pm 0.002$ mm ($n = 32$ sites from one mouse) (Fig. 5d, e). These results show that even in the soft tissue of the cerebral cortex, a thin and bendable glass micropipette can be manipulated with a precision of approximately 50 μm.

The risk of bleeding should depend on the distance between the target injection site and the nearest blood vessel. We therefore examined how the bleeding risk probability decreased as the safety margin distance increased. The bleeding risk probability was simulated as the proportion of injection sites corresponding to a blood vessel structure. Based on the distribution of the measured misalignments (Fig. 5e), the displacement was probabilistically added to 120 target

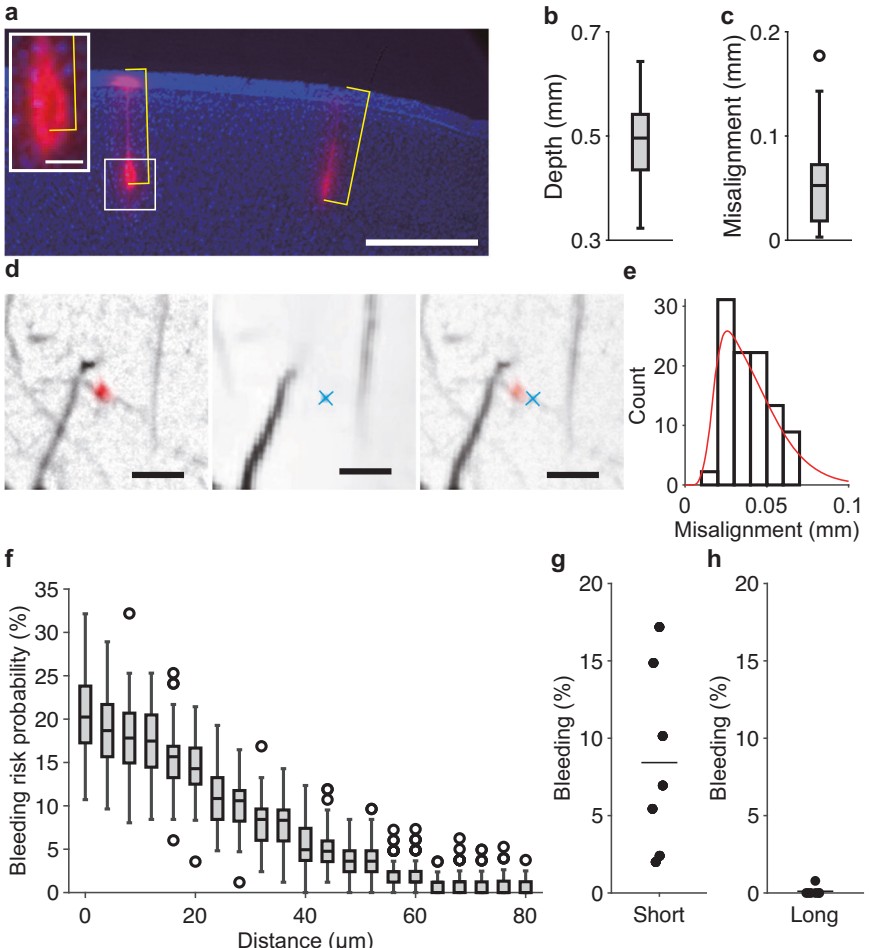

**Fig. 5 | The accuracy of the pipette injection in vivo. a**, A representative coronal section of the post-fixed cerebral cortex of a mouse into which fluorescent dye (CM-DiI; red) was injected. The section was Nissl-stained (blue). The yellow lines indicate the depth of the injection from the cortical surface. When the putative pipette location was considered as a black void surrounded by the fluorescent dye, the deepest point of the void was measured as the depth of the injection (inset). Otherwise, the injection's depth was determined by measuring the deepest point of the fluorescent dye. Scale bar, 0.5 mm. Scale bar in the inset, 50 μm. **b** Box-and-whisker diagram of the measured depth (*n* = 32 sites from one mouse). **c** Box-and-whisker diagram of the absolute difference between the targeted and measured depths (*n* = 32 sites from one mouse). For panel (**b**, **c**), the center line, box limits and whiskers denote the median, quartiles, and 1.5 × interquartile ranges in the boxplot, respectively. **d** Representative images showing the lateral misalignment of the

pipette injection site. Left: A fluorescence (red, CM-DiI) image of the actual injection site superimposed on a bright field image. Middle: The target injection site (a blue cross) superimposed on a bright field image of the SA-UNet segmentation. Right: A composite image of the left and middle images. Scale bar, 100 μm. **e** Histogram of the lateral misalignment of the pipette tip (*n* = 32 sites from one mouse). The red curve indicates a non-parametric approximation of the lateral misalignment distribution, calculated using a Gaussian kernel. **f** Box-and-whisker diagram of the bleeding risk probability against the safety margin distance (*n* = 100 simulations). The center line, box limits and whiskers denote the median, quartiles, and 1.5 × interquartile ranges in the boxplot, respectively. **g**, **h** Rates of short (**g**) and long (**h**) bleeding when the pipette was injected perpendicular to the cortical surface at each site (*n* = 7 mice). The horizontal line represents the mean. Source data are provided as the Source Data file.

injection sites that were determined when the safety margin distance was set in one cortical image. Then, the total number of virtual hits of the vessel was counted as the bleeding risk probability. The bleeding risk probability was near zero when the safety margin distance was more than 60 μm (e.g., 0.85 ± 0.09% at 64 μm) (Fig. 5f). Therefore, we set the safety margin distance to 65 μm.

**Multiple injections in the mouse dorsal cortex without bleeding**
The robot moved the pipette to the 3D coordinates of a target site along the perpendicular line at that site and inserted the tip into the cortical tissue (Supplementary Fig. 6a–h). After injecting the fluid, the pipette was pulled back and moved into position above the next target injection site (Supplementary Fig. 6i–l). This procedure was performed continuously for all sites in each animal. It took approximately 1 min to complete the series of movements at one injection site.

To determine the frequency of actual bleeding occurrence in the mouse cortex after automatic multiple injections, we conducted a total

of 679 injections in seven mice (AAV solution in six mice [5 males and 1 female, 12–14 weeks old] and the dye solution in one mouse [male, 8 weeks old]). In these mice, the range of the insertion angles was within ±10° in the V rotation and ±20° in the W rotation (Supplementary Fig. 7a, b). In 8.43% ± 2.24% of the injection sites, short-duration bleeding occurred, with this defined as bleeding that stopped within 30 s after starting (Fig. 5g). However, this short bleeding did not apparently affect the cortical structure after it had disappeared. By contrast, long bleeding, which was defined as bleeding lasting for more than 30 s, caused brain swelling or changed the vascular structure on the cortical surface and needed some hemostatic treatment. Long bleeding occurred in only one site in the seven mice (0.11% ± 0.11% of the injection sites) (Fig. 5h). Histological analysis one month after injection of the GCaMP-carrying AAV did not indicate any apparent cell death around the fluorescent areas (Supplementary Fig. 7c–f). The lateral spreading of the fluorescence in the most lateral region, which was relatively far from other expression sites, was 185.3 ± 16.1 μm

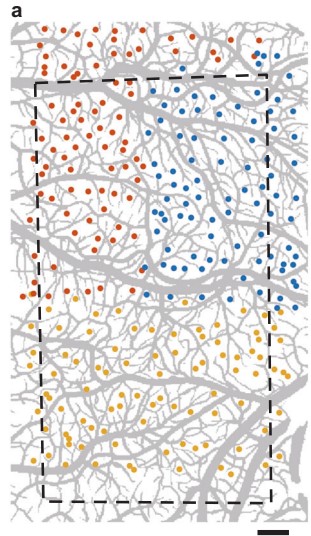
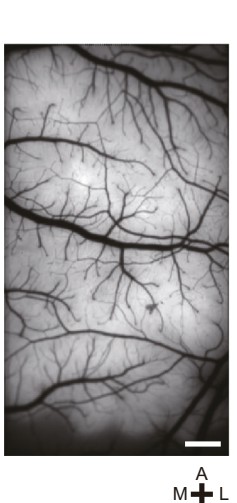
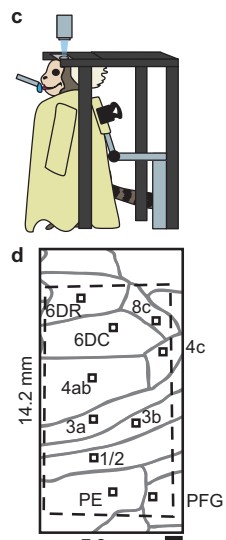
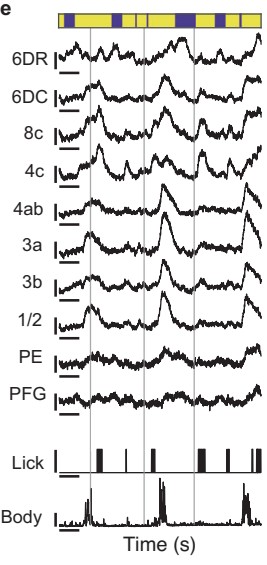

**Fig. 6 | Wide-field calcium imaging of the frontoparietal cortex in an awake marmoset. a** A vascular image of the marmoset frontoparietal cortex that was segmented by the SA-UNet (in gray) with target injection sites (colored dots). Blue, red, and yellow dots represent the first, second, and third sections, respectively. The black dashed frame represents the field of view for the fluorescence microscopy. The slightly distorted shape was caused by the affine transformation of the frame shown in (**b**), which was based on the blood vessel structure. This might be due to the elapsed time from the injection and distortion of the cortex caused by installation of the glass window. Scale bar, 1 mm. **b** Fluorescence image of the same area as (**a**) 7 weeks after the virus injection. Scale bar, 1 mm. A, P, M, and L stand for anterior, posterior, medial, and lateral, respectively. **c** Schematic of the experimental setup for wide-field one-photon calcium imaging of the marmoset in an awake head-fixed condition. **d** Putative cortical areas within the imaging window (black frame). The dashed frame indicates the field of view (14.2 × 7.2 mm). Parcellation is based on the stereotaxic atlas and ICMS (see Supplementary Fig. 8c). Small black squares represent ROIs for the corresponding cortical areas. Scale bar, 1 mm. **e** Representative time courses of ΔF/F at the ROIs of ten cortical areas shown in (**c**), licking, and body movement. The vertical gray lines show the reward timing. The vertical scale bars on the fluorescence traces represent 0.1 ΔF/F. The vertical scale bars with a lick represent a binary value of 1 and those with normalized body movement represent an intensity value of 1. The horizontal scale bars represent 5 s. The blue bars at the top indicate the quiet periods, whereas the yellow bars at the top indicate the periods outside of the quiet periods. Source data are provided as the Source Data file.

($n$ = 9 slices from two mice [males, 13 and 14 weeks old]) (Supplementary Fig. 7g). Considering that the average distance between neighboring injection sites was approximately 0.36 mm (Fig. 3d), the injection site spacing and the spread of GCaMP expression in mice seemed reasonable. Thus, we concluded that the estimation of the blood vessel structure on the 3D cortical surface, determination of the injection sites in the 2D vessel segmentation image, and 3D manipulation of the glass micropipette worked very well.

After these assessments, to further ensure that the glass pipette did not hit the vessel position when the 3D position determined as the injection site corresponded to the vessel position at the time of injection, an optional closed-loop function to avoid vessels was implemented (see "Realtime closed-loop evaluation of vessels" subsection in the Methods for details).

### Multipoint injections into the marmoset frontoparietal cortex without bleeding

Once we were able to demonstrate the effectiveness of multi-site injections in mice using ARViS, we applied it to the adult marmoset cortex. To enhance GECI expression by a tetracycline-inducible gene expression system[12], a mixture of two AAVs encoding hsyn-tTA and TRE-tandem-GCaMP6s[13,14] was injected into a total of 266 sites over the right frontoparietal cortex (16 × 8 mm) in one marmoset (Fig. 6a). The range of the insertion angles was −20° to 0° in the V rotation and −10° to +20° in the W rotation (Supplementary Fig. 8a, b). The injected cortical areas were roughly inferred from intracortical microstimulation (ICMS) and a stereotaxic atlas[13,42,43] as the area from the rostral part of the dorsal premotor cortex (6DR) to areas PE and PFG of the parietal cortex (Supplementary Fig. 8c). The entire injection process was divided into three cycles and the calibration was performed for each cycle because of changes in the surface shape over the surgery period and

the limited volume of solution that could be loaded into the glass micropipette (Fig. 6a). Cycles 1, 2, and 3 consisted of 89, 87, and 90 injections, respectively, at a depth of 0.5 mm from the cortical surface. Short bleeding was observed at seven sites (2.6%) in total (zero in cycle 1, two in cycle 2, and five in cycle 3), but no long bleeding was observed. In addition, no injection site was determined by the realtime closed-loop function to hit a blood vessel. Thus, ARViS was also effective in the marmoset cortex.

### Effect of pulsation of the marmoset brain on the estimation of the positions of the cortical surface

Although there was no long bleeding, the heartbeat could have caused significant errors in the cortical surface measurements[44]. Therefore, we estimated these possible errors. During the measurement of the marmoset cortical surface, the laser distance sensor moved in a zigzag pattern in the YZ plane (Supplementary Fig. 9a). The sensor moved in two phases: 0.1-mm steps in the Y direction and 8 mm steps in the Z direction (Supplementary Fig. 9a, b). Closer observation of the first phase with slow-speed sensor movement revealed depth signal fluctuations of up to 0.1 mm, which did not depend on the sensor's movement timing in the Y direction (Supplementary Fig. 9c). We speculated that these fluctuations were induced by animal movements. We selected one continuous period of time when the sensor position was in the most medial and lateral sides (red and blue lines in Supplementary Fig. 9a, b respectively) as a segment for further analysis to isolate these fluctuations. We constructed a linear regression model to fit the XYZ positions of the laser distance sensor to the measured depth of the cortical surface in 36 segments (in the medial and lateral sides) and extracted the residual (Supplementary Fig. 9d). This residual showed oscillation, so we suspected that the residual reflected the oscillatory heartbeat. In fact, the peak-to-peak intervals were 0.398 ± 0.0125 s (2.51 ± 0.07 Hz,

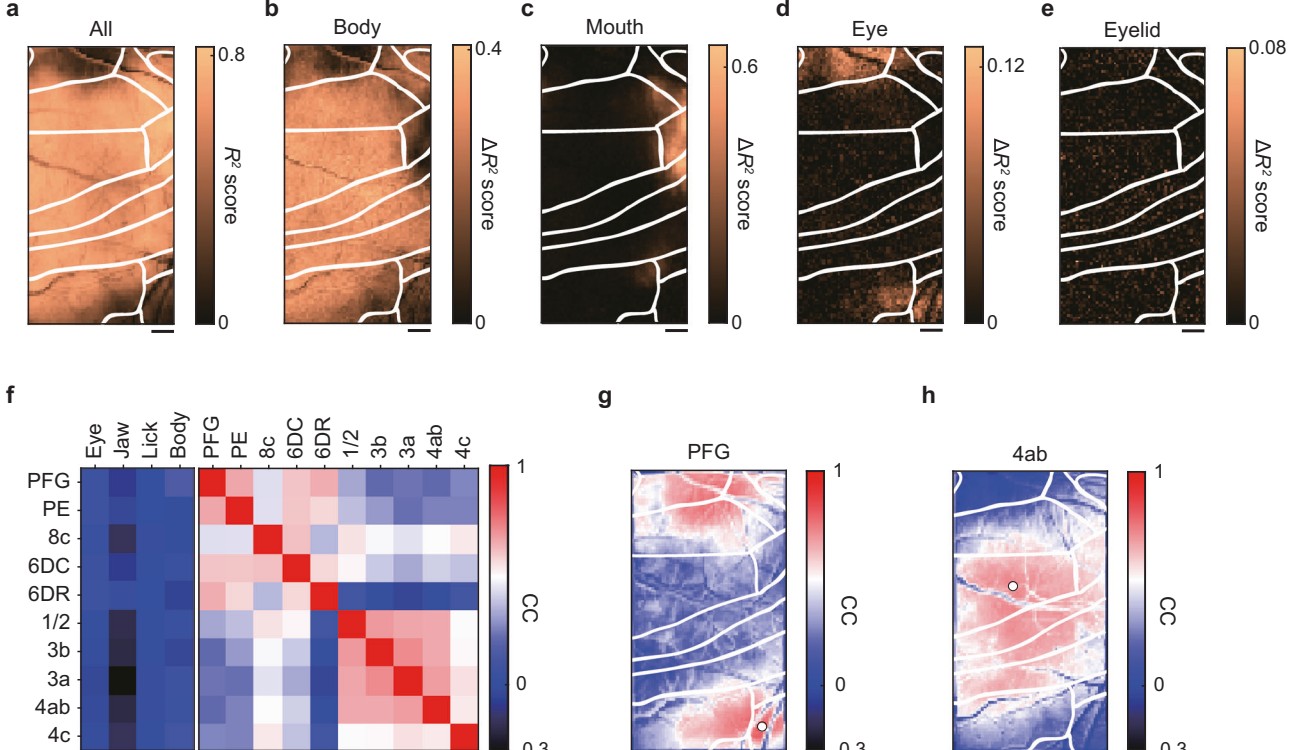

**Fig. 7 | Movement representations and activity correlations in the marmoset frontoparietal cortex. a** Spatial map of the accuracy ($R^2$) of the encoding model for predicting the neural activity with all behavioral variables. The pseudo-colors represent $R^2$ values. Scale bar for (**a**–**e**), 1 mm. **b**–**e** Spatial maps of the unique contribution ($\Delta R^2$) of the body (**b**), mouth (**c**), eye (**d**), and eyelid (**e**) variables to the explanation of the neural activity. The pseudo-colored bar represents the $\Delta R^2$ score. **f** Correlation matrix during the quiet periods between the cortical areas and movement variables (left), and between cortical areas (right). Two clusters between

cortical areas were determined using hierarchical clustering. The pseudo-colored bar represents the correlation coefficient (CC). Note that although the eye movement was not considered in the definition of the quiet periods, all neuronal activity showed very weak correlations with eye movement. **g**, **h** Correlation maps between seed pixels (indicated by white filled circles) located in area PFG (**g**) and area 4ab (**h**) with the other pixels. The pseudo-colored bar represents the correlation coefficient (CC). Source data are provided as the Source Data file.

$n = 18$ segments) in the medial side and $0.396 \pm 0.006$ s ($2.53 \pm 0.04$ Hz, $n = 18$ segments) in the lateral side, matching our recording of the heartbeat rate during surgery ($2.42 \pm 0.14$ Hz, $n = 6$ recorded timepoints), and the reported heartbeat rate of anesthetized marmosets ($3.13 \pm 0.10$ Hz)[45] (Supplementary Fig. 9e). The amplitude of the oscillation was $30.6 \pm 3.6$ μm ($n = 18$ segments) in the medial side and $46.2 \pm 1.2$ μm ($n = 18$ segments) in the lateral side (Supplementary Fig. 9f). Thus, the oscillation amplitude in the Z direction was larger in the lateral side than in the medial side.

Assuming that brain pulsation caused by heartbeat is mostly perpendicular to the cortical surface[46], the maximal heartbeat-induced displacement perpendicular to the cortical surface was estimated to be $29.9 \pm 3.5$ μm ($n = 18$ segments) in the medial side and $39.9 \pm 1.1$ μm ($n = 18$ segments) in the lateral side (Supplementary Fig. 9 g, h). These values are only $5.98\% \pm 0.70\%$ and $7.98\% \pm 0.22\%$ of the planned depth (500 μm), respectively. If this heartbeat-induced displacement perpendicular to the cortical surface also occurred while picturing the cortical surface with Camera S (Supplementary Fig. 3i), the possible maximal displacement of the reference position in a plane perpendicular to the optical axis of Camera S was estimated to be $1.6 \pm 0.2$ μm ($n = 18$ segments) in the medial side and $13.0 \pm 0.4$ μm ($n = 18$ segments) in the lateral side (Supplementary Fig. 9j). Given that the distance from the planned injection site to the nearest blood vessels was set at more than 65 μm, this possible displacement would not have a substantial impact on the injection safety. Thus, we consider that heartbeat-induced misalignment of the injection point during the cortical surface measurement and AAV injection is not negligible, but would have only a minor effect.

## Behavior-related cortical activities revealed by wide-field one-photon calcium imaging of an awake marmoset

Wide-field epi-fluorescent imaging successfully showed uniform fluorescence expression over the area of $14 \times 7$ mm at 7 weeks after the injections (Fig. 6b). To verify whether ARViS-assisted fluorescence expression was effective for functional imaging in the marmoset, we conducted wide-field one-photon calcium imaging in an awake head-fixed condition[13,15] over three sessions (days) (Fig. 6c). Licking frequently occurred following the water delivery from a spout in front of the animal's mouth, whereas other body movements also occurred apart from the licking timing. When the neuronal activity (relative fluorescence change $\Delta F/F$) in regions of interest (ROIs) in 10 cortical areas was extracted, the somatosensory-primary motor areas (1/2, 3a, 3b, and 4ab) appeared to increase activity when body movements occurred, and the ventral areas 4c and 8c appeared to increase the activity when licking occurred (Fig. 6d, e). The time course of $\Delta F/F$ shows that area 4c activity was related to licking and area 4ab activity was related to body movements (Supplementary Fig. 10a–d).

To reveal how the behaviors were coded over the imaging field, we constructed linear ridge regression models with behavioral variables to reconstruct the fluorescence signals in each pixel[47,48]. To assess the models' effectiveness in capturing neural activity across various cortical areas, we calculated the 10-fold cross-validated coefficient of determination (denoted as $R^2$)[47,48]. The model accounted for $54.9\% \pm 2.5\%$ (mean ± s.e.m., $n = 3$ sessions) of the neuronal activity (Fig. 7a). Subsequently, we estimated the unique contribution ($\Delta R^2$) of each variable[47] (see Methods for details). Body movement was widely represented, with areas 3 and 4 at the center (Fig. 7b). Mouth

movement contributed strongly to the activity of area 4c (Fig. 7c). Eye movement contributed to the activity of parts of area 6DR, area 8c, and a region near the boundary between areas PE and PFG, but eyelid movement was not strongly represented in any area (Fig. 7d, e). Area 4c is responsible for facial movement, area 4ab is responsible for forelimb and body movements, and area 8c and the intraparietal areas are responsible for eye movement[43,49–51]. Thus, ARViS-assisted, wide-field one-photon calcium imaging showed the different representations of different movements in the marmoset frontoparietal cortex.

### Wide-field calcium imaging reveals distinct frontoparietal networks

To assess the functional connectivity that was not directly related to movements, we extracted the neuronal activity during the quiet state, which we defined as the time period excluding that from 0.3 s before to 2 s after the occurrence of licking or body movement. Then, we calculated correlations in the fluorescence change between the areas during this quiet state. A hierarchical clustering method was used to group the areas into two clusters: one included most of the somatosensory-primary motor areas, and the other encompassed the regions anterior and posterior to those in the first cluster (Fig. 7f). The spatial patterns of these correlations were also visualized in a seed correlation map with areas PFG and 4ab serving as the seeds (Fig. 7g, h). Anatomically, area 6DR receives axons from lateral and caudal parietal areas (e.g., area PFG and the intraparietal area)[50]. Thus, the correlation structure revealed by the linear regression analysis should reflect the anatomical connectivity, although the cortical structural map was only roughly inferred. These results indicate that ARViS-assisted wide-field calcium imaging showcased correlation patterns over the marmoset frontoparietal cortex.

### Behavior-related neurons revealed by two-photon calcium imaging of the marmoset motor cortex

Finally, we conducted two-photon calcium imaging of a field in area 4c (field 1) and a field in area 4ab (field 2) in the awake condition (Fig. 8a–f). Each field was imaged at different depths on different days. We detected 159 and 182 active neurons in fields 1 and 2, respectively. Consistent with the results from one-photon imaging, the neurons in field 1 responded more strongly to the water delivery than the neurons in field 2 (Fig. 8g–i). The licking was decoded much better by individual neurons in field 1 than in field 2, whereas the body movement was decoded better by those in field 2 than in field 1 (Fig. 8j, k). The difference in decoding prediction accuracy was also prominent when the activities of individual neurons were summed over each field as an estimate of the population activity (Fig. 8l, m). These findings suggest that the behavior-related population activity revealed by wide-field one-photon calcium imaging well reflected the average behavior-related activity of individual neurons. The half-decay time of calcium transients in individual neurons was $0.94 \pm 0.05$ s ($n = 119$ neurons with a certain range of amplitudes) (Supplementary Fig. 10e, f), which was comparable to that in the original paper that reported GCaMP6s[52]. This result suggests that the multiple injections did not induce apparent functional damage to active GCaMP6s-expressing neurons. Thus, ARViS-assisted broad gene expression allowed us to reveal both the neuronal-averaged activity and the activity of multiple individual neurons in the marmoset cortex.

## Discussion

In the current study, we developed an automated injection system capable of performing sequential multipoint injections into the dorsal cortex of mice and a marmoset. The injection point accuracy was 50 μm along the horizontal and perpendicular axes. The probability of long bleeding requiring hemostatic treatment was only 0.1% per injection. Furthermore, wide-field calcium imaging in a marmoset showed that this system allowed the broad expression of sensor proteins.

Although the technology for blood vessel recognition is advanced, especially in fundus examination, image recognition in fundus examination and robotic surgery have rarely been performed consecutively. A neurosurgical instrument that uses deep learning-based segmentation to automatically recognize blood vessels in the brain during robot-assisted surgery and suppress high-risk movements has been designed[53], and while it showed good accuracy in a clay-based phantom study, it was not tested in vivo living tissue. For this neurosurgical instrument, the distance between the target and vessel was set to 2 mm, which is much longer than the distance in our study (65 μm). Therefore, to our knowledge, the present study is the first to apply vessel recognition technology to biological intervention with high precision. In the current study, we segmented blood vessels only from the cortical surface, although there are also many vessels within the cortical tissue. However, cortical vessels tend to run perpendicular to the surface[54], and thus many of them exist beneath the vasculature structure on the cortical surface, and they might therefore be well separated from the pathway of pipette insertion and the bleeding probability might be very small.

In the field of neurosurgery, the ROSA® system is a representative example of a robotic platform that was developed to automate the insertion of electrodes for deep brain stimulation therapy. By superimposing CT and MRI images on the patient's head using a metal frame attached to the patient's skull as a reference point, the system allows electrodes to be inserted while avoiding nerve nuclei and ventricles[32]. Although this system is suitable for deep brain intervention, it has an error of approximately 0.8 mm from the target[33], which is not sufficiently accurate for virus injection into rodents and NHPs. In other robotic systems, such as NeuroMate and iSYS1[33,55–57], CT images are used for registration. The diameter of the electrodes used for deep nuclei stimulation is more than 1 mm[58], which is much longer than the diameter of the micropipette tip (0.03 mm). Therefore, these electrode insertion systems require hemostatic treatment during insertion into the brain parenchyma (Supplementary Table 2). The present study solved this problem by performing high-precision surface measurement using a laser distance sensor and high-precision calibration using cameras to accurately position the needle tip with an error of less than 50 μm. Furthermore, we demonstrated that even in soft cortical tissue with a curved surface, the misalignment was still approximately 50 μm along the axes parallel and perpendicular to the cortical surface.

Needle interventions in the brain are very common in neuroscience, and include single electrode recording and stimulation, multi-electrode recording[59,60], and patch-sequencing, a combination of whole-cell patch-clamp recording and RNA sequencing[61]. The techniques presented in this study could be incorporated into the automated large-scale implementation of these methods. Even in deep brain stimulation, some of the algorithms developed in the current study may be applicable to CT angiographic images or magnetic resonance angiography to facilitate automatic determination of the best no-bleeding penetration pathway for the electrode and accurate manipulation of the electrode to the target area. The real-time control of image recognition and robotic technologies in ARViS would potentially be beneficial in many research and clinical medicine fields.

Apart from multiple microinjections, there are other methods for transfecting AAVs into NHP brains. For example, in-utero virus injections may be possible. However, developmental effects on the transfected gene are inevitable; it takes a year to obtain a young adult, even in the marmoset, and it is not clear whether a sufficiently large number of neurons will strongly express the biosensors. Although transvenous transfection of AAV variants such as AAV-PHP.eB and AAV.CAP-B10 will undoubtedly be further improved, even in NHPs[24], the effectiveness of the transfection and the gene expression level in cortical neurons is unlikely to be sufficiently high for functional imaging in adult NHPs[62,63].

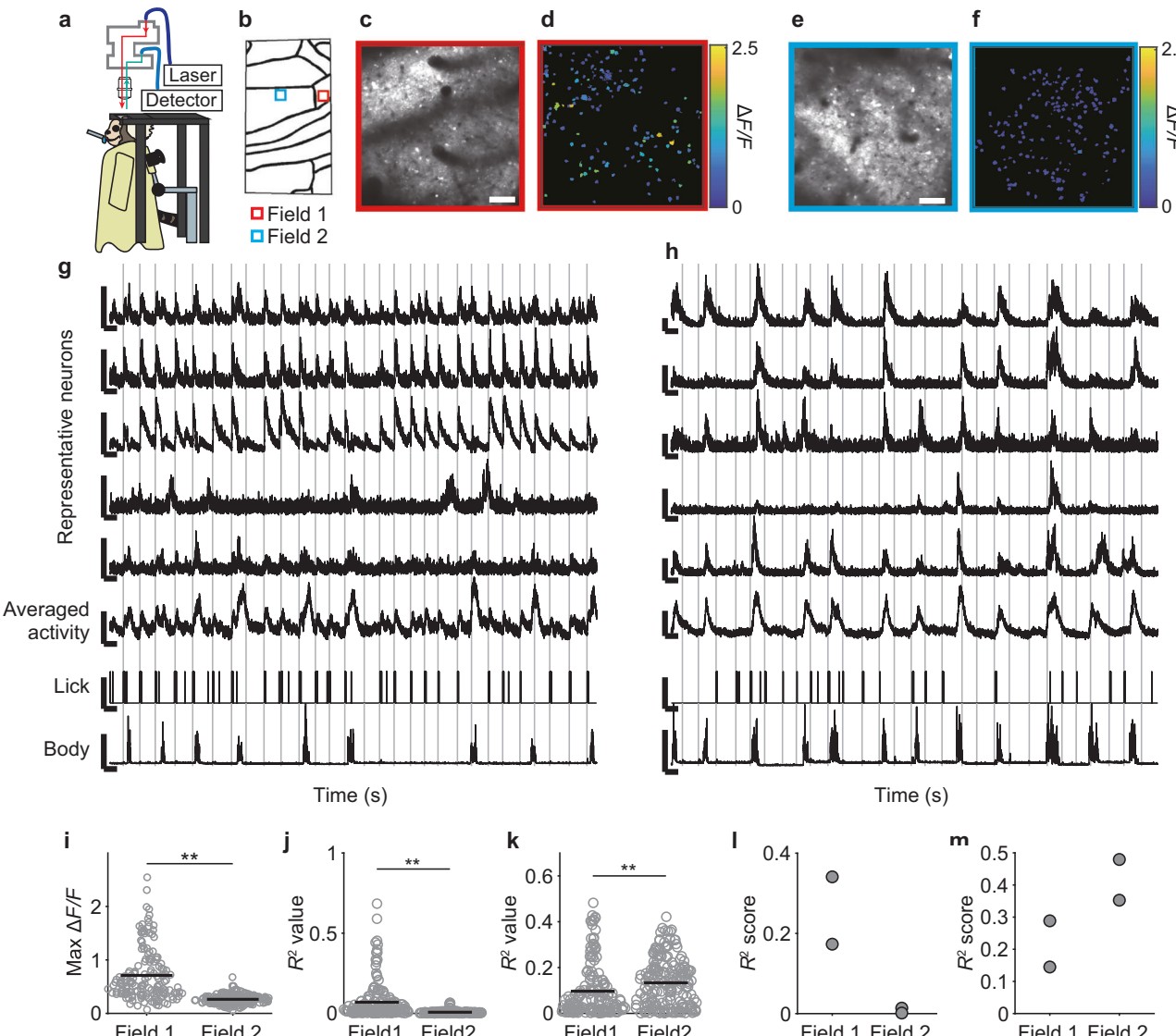

**Fig. 8 | Two-photon calcium imaging of movement-related neurons in the marmoset motor cortex. a** Schematic of the experimental setup to perform two-photon calcium imaging in the awake condition. **b**, Locations of two fields (red for field 1 and blue for field 2) in which two-photon imaging was conducted. **c**, **e** Time-averaged and contrast-adjusted two-photon images for field 1 (**c**) and field 2 (**e**). Scale bars, 100 μm. **d**, **f** Neuronal somata extracted by the CNMF algorithm (Pnevmatikakis et al., 2016) from field 1 (**d**) and field 2 (**f**). Each neuronal soma is colored according to the maximum value from the trial-averaged $\Delta F/F$ trace during the period spanning 1 s before to 8 s after the reward timing. **g**, **h** Representative traces of five neurons from field 1 (**g**) and field 2 (**h**), averaged activity of all active neurons in each field, lick frequency, and body movement. The vertical gray lines show the reward timing. The vertical scale bars with the fluorescence traces, averaged activity, lick, and movement represent 500% $\Delta F/F$, 100% $\Delta F/F$, an intensity value of 1 (arb. unit), and an intensity value of 5 (pixel), respectively. The horizontal scale bars represent 10 s. **i** Maximum values of trial-averaged $\Delta F/F$ traces for a period spanning 1 s before to 8 s after the reward timing in field 1 (red, $n = 159$ from two sessions) and field 2 (blue, $n = 182$ from two sessions). Horizontal bars indicate the means. **p = 5.0 \times 10^{-23}$, two-sided Welch's test. **j**, **k** The prediction accuracy ($R^2$) of individual neurons in fields 1 (red, $n = 159$ from two sessions) and 2 (blue, $n = 182$ from two sessions) for the licking frequency (**j**) and body movement (**k**). Horizontal bars indicate the means. **p = 2.6 \times 10^{-10}$ (**j**) and 0.0028 (**k**), one-sided Welch's test. **l**, **m** The prediction accuracy ($R^2$) of the averaged activity in fields 1 ($n =$ two sessions) and 2 ($n =$ two sessions) for the lick frequency (**l**) and body movement (**m**). Source data are provided as the Source Data file.

Thus, the systematic bleed-free multiple microinjection provided by ARViS should be highly useful for cortex-wide imaging of a variety of biosensors in adult NHPs.

In contrast to NHPs, we now discuss whether ARViS is useful in research using mice that have many transgenic lines and Cre-lines. Relatively uniform expression or cell type-specific expression of biosensors is possible by crossbreeding the appropriate Cre-line mice and sensor gene-expressing flox-lines. Even if there are no new sensor gene-expressing lines, they can be created within a few months. However, it takes more time to crossbreed multiple lines with different color sensors. By contrast, transvenous infection with AAV-PHP.eB is easy and quick, different sensors can be simultaneously transfected, and the sensors are widely expressed over the whole brain. However, a problem with this approach is that it is not possible to limit gene expression to areas that are not genetically defined. ARViS will also be useful for expressing opto-genetic or chemogenetic probes, and RNAi or constitutively active forms of the related genes, in any cell type in any region of the mouse cerebral cortex. An example of the future application of ARViS could be in addressing some of the biggest neuroscience questions: which areas of the neocortex, and how much neuronal and glial cell activity and molecular activity in the neocortex, are required for sleep regulation[64-66]. The subcortical areas, as well as the cerebral cortex, regulate sleep[67]. ARViS can label many cerebral

cortical neurons, but not subcortical neurons, which is very difficult to perform with AAV-PHP.eB. Systematically varying the transgenic area is easy with ARViS, but manual injection requires a great deal of labor and skill.

ARViS could also be applied to the neocortex of other mammals, including rats and macaques. The area of the whole dorsal cortex in the rat is approximately $12 \times 12$ mm, which is close to the current craniotomy area in the marmoset ($8 \times 16$ mm). Thus, ARViS may be applied to the whole rat dorsal cortex. By contrast, the macaque cortex is much larger than the marmoset cortex, and it would therefore be difficult to use the present ARViS to perform cortex-wide injection. However, ARViS would be more efficient than manual injections, even if the injected area is limited to a part of the neocortex (probably one functional cortical area) with an area that is comparable to the area imaged in the current study. An advantage of ARViS is its ability to realize spatially-selective gene expression independent of endogenous expression patterns and anatomical boundaries in both rodent and NHP cerebral cortices. ARViS promises a cross-species approach to many questions related to a variety of functions of the cerebral cortex, questions that have been difficult to solve with previous methods alone. Wide-field one-photon imaging can detect the dynamics of inter-areal interactions while two-photon imaging can detect the multicellular activity at the single-cell level. These methods can be applied to the same individual animal as demonstrated in the current study. Thus, once ARViS is used to express GECI or other probe over a broad region of the cerebral cortex, the strategy of clarifying the dynamics across multiple cortical areas and then revealing the multicellular activity at the single-cell level, especially in regions involved in those dynamics, would be beneficial for understanding cortical information processing across species. We summarize the advantages that have not been realized in previous practices and possible future applications of ARViS in Supplementary Fig. 11.

### Limitations of this study

The limitations of ARViS include the long calibration time (90 min) for the glass pipette position and the limited volume of solution (~4 μL) for each pipette. The currently used injector limits the amount of virus that can be loaded. This problem may be overcome by using a large capacity syringe as an injector. However, the accuracy of injection pressure and single injection volume might be lower. We suspect that the main obstacle to refilling the same pipette with the solution is debris that becomes attached to the pipette tip during insertion. This may be cleared with detergent, such as that used for repetitive patch-clamp recordings[68]. In addition, the current system was unable to compensate for deformation of the surface landscape after the AAV injection started. If cerebral edema is caused by vessel damage, it induces further bleeding and changes the cortical surface structure. To overcome this, we added the realtime closed loop algorithm to avoid the glass pipette from hitting the blood vessels. However, this algorithm cannot cope with a major overall change, and ARViS cannot avoid blood vessels of less than 20 μm on the cortical surface or blood vessels within the brain parenchyma. Thus, our method does not guarantee elimination of risk to all blood vessels. Finally, the injection conditions (the density and number of injection sites, the injection pressure and duration, and the virus titer and solution volume) should be determined for each experiment because the solution viscosity, virus toxicity, and required expression level depend on virus type, probe type, and experiment type.

## Methods

### Robot construction

The base of a six-axis hexapod robot with a maximum travel range of 100 mm in translation and 60° in rotation (Hexapod H-820; Physik Instrumente, Karlsruhe, Germany) was fixed vertically onto the base plate, and an injector (Nanoject III; Drummond Scientific, Broomall, USA), a laser distance sensor (HG-C1050-P; Panasonic, Osaka, Japan), and a CMOS camera (acA4024-29um; Basler, Germany) (Camera S) were attached to the stage of the robot via the original holding fixture. The original holding fixture was first created with a 3D printer but was then replaced with a metal one. The injector was attached to a programmable linear actuator (MTS50/M-Z8; Thorlabs, Newton, USA) that was connected to a computer via a USB port and could be moved backward during surface scanning. The injector and the laser distance sensor were connected to the Hexapod controller through analog signal cables, allowing them to communicate with the computer via the Hexapod. The camera (referred to as Camera S) was fixed approximately 16° relative to the vertical axis to capture the cortical surface (Fig. 1e) and the tip of the glass micropipette. During the step of the calibration of the micropipette tip, two additional cameras (acA1440-220um; Basler, Germany) were set to detect it on the surgical plate via attachments. All these cameras were connected to the computer via USB ports. The code for the ARViS program was written and run in MATLAB (R2020a, R2021b; MathWorks, USA).

### Tool center point calibration problem

Calculating the position of the pipette tip is equivalent to solving a tool center point (TCP) calibration problem. Normally, this problem is solved by bringing the needle tip into contact with a specific point and rotating the needle while maintaining that contact[38]. However, since the micropipettes used in this study were very fragile, a non-contact method was required. Zhang et al.[39] proposed a binocular vision-based approach to this problem. Therefore, we adapted this method as: "1. Inference of the 3D coordinates of the pipette tip with the binocular camera system" and "2. Transformation of the 3D pipette coordinates to the frame centered at the pivot point". Furthermore, since the AAV injection into the neocortex required a higher level of precision than that of Zhang et al., we developed two calibration methods to minimize errors that were not considered in the previous research: "3. Compensation for translation error" and "4. Compensation for rotational error".

### Inference of the 3D coordinates of the pipette tip with the binocular camera system

**Pipette tip detection.** A glass micropipette filled with the desired solution was connected to the injector. With the actuator, the position of the pipette tip was fixed to be at the focal plane of Camera S. In the images acquired by the two other cameras (Camera L and Camera R) for taking pictures of the pipette tip from the side, the origin of the two-dimensional coordinate frame is located at the top left corner (Fig. 4a). The coordinates of the pipette tip within this frame were obtained as follows. First, the positions of the two cameras were fine-tuned so that the pipette tip could be captured. Second, the background gradation in illuminance was removed through applying a top hat filter with a square structuring element to the captured images. Then, the images were binarized by thresholding them according to intensity values (Fig. 4b). Finally, those binarized pixels with a value of one and the top five highest y coordinates were detected, and the x and y coordinates of these pixels were averaged.

**Camera calibration.** The two-dimensional frames on the captured images need to be transformed to a three-dimensional camera coordinate system. Frame {C} is the camera coordinate system frame. We aimed to estimate the camera matrices $\mathbf{C}^L$ and $\mathbf{C}^R$, which project the three-dimensional coordinates of the pipette tip $\mathbf{P}^C_{\text{Pipette}} = (XYZ1)^T$ in the frame {C} centered at Camera L into the two-dimensional coordinates of the pipette tip $\mathbf{P}^L_{\text{Pipette}} = (x^L y^L 1)^T$ and $\mathbf{P}^R_{\text{Pipette}} = (x^R y^R 1)^T$, which were captured by Camera L and Camera

R, respectively.

$$w^L \mathbf{P}_{\text{Pipette}}^L = \mathbf{C}^L \cdot \mathbf{P}_{\text{Pipette}}^C$$

$$= \begin{bmatrix} a_x^L & 0 & c_x^L \\ 0 & a_y^L & c_y^L \\ 0 & 0 & 1 \end{bmatrix} \cdot [\mathbf{I}|\mathbf{0}] \cdot \mathbf{P}_{\text{Pipette}}^C , \quad (1)$$

$$w^R \mathbf{P}_{\text{Pipette}}^R = \mathbf{C}^R \cdot \mathbf{P}_{\text{Pipette}}^C$$

$$= \begin{bmatrix} a_x^R & 0 & c_x^R \\ 0 & a_y^R & c_y^R \\ 0 & 0 & 1 \end{bmatrix} \cdot [\mathbf{R}|\mathbf{t}] \cdot \mathbf{P}_{\text{Pipette}}^C , \quad (2)$$

where

$$a_x = m_x f, \quad (3)$$

$$a_y = m_y f. \quad (4)$$

Values with a superscript L or R indicate the values for Camera L or Camera R, respectively; $f$ represents the focal length of the camera (mm); and $m_x$ and $m_y$ are the scales in x and y directions (pixel/mm), respectively. $(c_x, c_y)$ represent the optical center in the images (pixel). These parameters are intrinsic parameters of the cameras. $\mathbf{R}$ is a $3 \times 3$ rotation matrix relative to Camera L, and $\mathbf{t}$ is a $3 \times 1$ translation vector from Camera R to Camera L. These rotations and translations are extrinsic parameters of the cameras. $\mathbf{C}$ is a camera matrix that represents the transformation, which is composed of intrinsic and extrinsic parameters. Camera L needs neither translation nor rotation to project to Camera L itself; thus, the extrinsic parameters are expressed as $[\mathbf{I}|\mathbf{0}]$, where $\mathbf{I}$ is an identity matrix. For the camera calibration, the inference of both intrinsic parameters and extrinsic parameters is necessary. The intrinsic parameters of the camera were estimated at the first use because they were unique to each camera and were generally invariant in daily use. For the extrinsic parameters, the position and angles were changed to allow focus on the pipette tip, so they were therefore estimated every time the pipette was replaced.

The intrinsic parameters were calculated by correlating grid coordinates obtained by moving the pipette tip in 0.2 mm intervals using the robot. This procedure was performed using the "estimate-CameraParameters" function in Matlab's Computer Vision Toolbox. The extrinsic parameters, the position and angle of Camera R in the frame {C}, were estimated as follows. First, the pipette tip was moved randomly. Then, $\mathbf{P}_{\text{Pipette}}^L\langle i\rangle$ and $\mathbf{P}_{\text{Pipette}}^R\langle i\rangle$ were obtained for the $i$-th movement. On the basis of these pairs of coordinates, the $3 \times 3$ essential matrix $\mathbf{E}$ that satisfies the constraint below was estimated using the "estimateEssentialMatrix" function,

$$\left(\mathbf{P}_{\text{Pipette}}^L\langle i\rangle \, 1\right) \cdot \mathbf{E} \cdot \left(\mathbf{P}_{\text{Pipette}}^R\langle i\rangle \, 1\right)^T = 0 \ (i = 0, 1, 2, \cdots, 180). \quad (5)$$

Then, the "relativeCameraPose" function was used to calculate the rotation and translation relative to Camera L from the essential matrix.

### Transformation of the 3D pipette tip coordinates to the frame centered at the pivot point

In previous research that proposed a calibration method for the robot arm[39], the edge of the arm was called the effector, and the coordinate frame centered at the effector was defined as frame {E}. Although the Hexapod six-axis stage was used instead of a robotic arm in the current study, the pivot point, which is the rotation center in the Hexapod control system, can be considered as the origin of frame {E} when it is fixed at a certain coordinate (Fig. 4c). Therefore, the pivot point was

regarded as the effector in the current study. The frame of the robot base system is represented as frame {B} (Fig. 4c).

The goal of this section is to estimate $\mathbf{P}_{\text{Pipette}}^B$, the coordinates of the pipette tip in the frame {B} ($3 \times 1$ vector). $\mathbf{P}_{\text{Pipette}}^B$ is expressed in two different ways. The first one is:

$$\mathbf{P}_{\text{Pipette}}^B = \mathbf{R}_E^B\langle i\rangle \cdot \mathbf{P}_{\text{Pipette}}^E + \mathbf{t}_E^B\langle i\rangle \ (i = 0, 1, 2, \cdots, N), \quad (6)$$

where $\mathbf{P}_{\text{Pipette}}^E$, a $3 \times 1$ vector, is the initiation position of the pipette tip in frame {E}, and $\mathbf{R}_E^B\langle i\rangle$ and $\mathbf{t}_E^B\langle i\rangle$ are the rotation and position of the effector in frame {B} after the $i$-th movement, respectively. $\mathbf{P}_{\text{Pipette}}^E$ is the only unknown variable required to calculate $\mathbf{P}_{\text{Pipette}}^B$ in Eq. (6). The second expression of $\mathbf{P}_{\text{Pipette}}^B$ is:

$$\mathbf{P}_{\text{Pipette}}^B = \mathbf{R}_C^B \cdot \mathbf{P}_{\text{Pipette}}^C\langle i\rangle + \mathbf{t}_C^B \ (i = 0, 1, 2, \cdots, N), \quad (7)$$

where $\mathbf{R}_C^B$ and $\mathbf{t}_C^B$ are the rotation matrix and translation vector in the camera coordinate system observed from frame {B}, respectively, and which are unknown here, and $N$ is the total number of movements. $\mathbf{P}_{\text{Pipette}}^C\langle i\rangle$ is the position of the pipette tip in frame {C} after the $i$-th movement. By solving Eqs. (6) and (7) simultaneously, the following equation is obtained:

$$\mathbf{R}_C^B \cdot \mathbf{P}_{\text{Pipette}}^C\langle i\rangle + \mathbf{t}_C^B = \mathbf{R}_E^B\langle i\rangle \cdot \mathbf{P}_{\text{Pipette}}^E + \mathbf{t}_E^B\langle i\rangle \ (i = 0, 1, 2 \cdots, N). \quad (8)$$

Although it is impossible to obtain $\mathbf{P}_{\text{Pipette}}^C$ directly, the following Eq. (9) can be obtained by subtracting Eq. (8) at $i = 0$ from Eq. (8) at the $i$-th movement while keeping a fixed rotation of the end effector ($\mathbf{R}_E^B$), since $\mathbf{R}_C^B$, $\mathbf{t}_C^B$, and $\mathbf{P}_{\text{Pipette}}^E$ are constant:

$$\mathbf{R}_C^B \cdot \left(\mathbf{P}_{\text{Pipette}}^C\langle i\rangle - \mathbf{P}_{\text{Pipette}}^C\langle 0\rangle\right) = \mathbf{t}_E^B\langle i\rangle - \mathbf{t}_E^B\langle 0\rangle (i = 0, 1, 2 \cdots, N). \quad (9)$$

In Eqs. (6–9), N was set at 200. Here, we can obtain $\mathbf{R}_C^B$ using singular value decomposition (SVD). Next, by subtracting Eq. (8) at $i = 0$ from Eq. (8) after the $i$-th movement with rotation, the following equation is obtained:

$$\left(\mathbf{R}_E^B\langle i\rangle - \mathbf{R}_E^B\langle 0\rangle\right) \cdot \mathbf{P}_{\text{Pipette}}^E = \mathbf{R}_C^B \cdot \left(\mathbf{P}_{\text{Pipette}}^C\langle i\rangle - \mathbf{P}_{\text{Pipette}}^C\langle 0\rangle\right) - \left(\mathbf{t}_E^B\langle i\rangle - \mathbf{t}_E^B\langle 0\rangle\right)$$
$$(i = 0, 1, 2 \cdots, 40). \quad (10)$$

Then, $\mathbf{P}_{\text{Pipette}}^E$ can be calculated using SVD. Finally, $\mathbf{P}_{\text{Pipette}}^B$ is estimated from Eq. (6).

### Compensation for translation error

There are some error-inducing factors that were not discussed in the previous research. In the current study, two additional compensations for these errors were performed. The first error is the error derived from the pipette detection on the captured images and camera calibration, which mainly affects the estimation of $\mathbf{R}_C^B$ in Eq. (9). To better estimate $\mathbf{R}_C^B$ from error-laden point data in the stereo images, the robust estimation technique of MSAC was used. This algorithm consisted of five steps as follows:

Step 1: Determination of the sample number for estimation of $\mathbf{R}_C^B$. Since there are nine unknown variables in the $\mathbf{R}_C^B$ matrix, at least nine equations must be solved simultaneously in the estimation. Each pair consisting of the effector translation vector $\mathbf{t}_E^B\langle i\rangle - \mathbf{t}_E^B\langle 0\rangle$ in frame {B} and the pipette translation vector $\mathbf{P}_{\text{Pipette}}^C\langle i\rangle - \mathbf{P}_{\text{Pipette}}^C\langle 0\rangle$ in frame {C} contributes three equations, one for each axis. Therefore, at least three pairs of translation vectors are necessary to estimate $\mathbf{R}_C^B$. Considering the stability of the estimation, nine samples were randomly chosen from 200 samples.

Step 2: Estimate $\mathbf{R}_C^B$ ($\hat{\mathbf{R}}_C^B$) using the samples obtained in Step 1.

Step 3: Estimate the translation vector $\hat{\mathbf{t}}_{E\langle i\rangle}^B - \hat{\mathbf{t}}_{E\langle 0\rangle}^B$ from $\mathbf{P}_{\text{Pipette}}^R$ and $\hat{\mathbf{R}}_C^B$.

Step 4: Estimate the cost function:

$$C_2 = \sum_i \rho_2(e_i^2),$$ (11)

where

$$e^2 = \left( \left( \hat{\mathbf{t}}_{E\langle i\rangle}^{B} - \hat{\mathbf{t}}_{E\langle 0\rangle}^{B} \right) - \left( \mathbf{t}_{E\langle i\rangle}^{B} - \mathbf{t}_{E\langle 0\rangle}^{B} \right) \right)^2$$ (12)

and $\rho_2$ is defined as follows:

$$\rho_2(e^2) = \begin{cases} e^2 & e^2 < T^2 \\ T^2 & e^2 \geq T^2, \end{cases}$$ (13)

where $T = 0.015$ is a threshold for the errors. The samples above this value are called outliers and are virtually omitted from the estimation by replacing $e^2$ with $T^2$, which means ignoring outlying deviating points.

Step 5: Repeat Steps 1–4 and adopt $\hat{\mathbf{R}}_C^B$, which minimizes the cost function.

From this procedure, the movement of the pipette tip was estimated from the captured images with an error rate of less than 15 μm for a 0.5 mm movement (Fig. 4e).

## Compensation for rotation error

The second aspect of the error is deformation of the hexapod robot due to its weight. During the system development, we found that when the pipette rotated around the pivot point that should correspond to the pipette tip, the tip location shifted a maximum of 0.4 mm (Fig. 4g). The estimation of $\mathbf{R}_E^B$ was not affected by this shift because the repeatability of the translation was stable as long as the angle of the stage was constant, but the estimation of $\hat{\mathbf{P}}_{Pipette}^E$ was biased and shifts of a few hundred micrometers were unacceptable for our purpose. This kind of error is commonly encountered when a commercial device is used. We handled this problem by grid searching of the pipette tip around the estimated $\hat{\mathbf{P}}_{Pipette}^E$ and measuring the shift of the pipette tip directly.

The shift of the pipette tip caused by the rotation was found to be invariant within the possible range of the injection position. Thus, the translation of the robot stage can be expressed as follows:

$$\mathbf{t}_E^B\langle i\rangle = \tilde{\mathbf{t}}_E^B\langle i\rangle + \Delta\mathbf{t}(\mathbf{R}_E^B\langle i\rangle),$$ (14)

where $\Delta\mathbf{t}(\mathbf{R}_E^B\langle i\rangle)$, a $3\times1$ vector, represents the 3D shift between the expected position from the rotation $\mathbf{R}_E^B\langle i\rangle$, and $\tilde{\mathbf{t}}_E^B\langle i\rangle$ is the expected movement after the $i$-th robot movement. Without rotation,

$$\Delta\mathbf{t}(I) = (0\ 0\ 0)^T.$$ (15)

Considering these effects, Eq. (10) should be impossible to solve because there are two additional unknown terms, $\Delta\mathbf{t}(\mathbf{R}_E^B\langle i\rangle)$ and $\Delta\mathbf{t}(\mathbf{R}_E^B\langle 0\rangle)$:

$$\left( \mathbf{R}_E^B\langle i\rangle - \mathbf{R}_E^B\langle 0\rangle \right) \cdot \mathbf{P}_{Pipette}^E = \mathbf{R}_C^B \cdot \left( \mathbf{P}_{Pipette}^C\langle i\rangle - \mathbf{P}_{Pipette}^C\langle 0\rangle \right)$$
$$- \left( \tilde{\mathbf{t}}_E^B\langle i\rangle - \tilde{\mathbf{t}}_E^B\langle 0\rangle \right) - \left( \Delta\mathbf{t}\left(\mathbf{R}_E^B\langle i\rangle\right) - \Delta\mathbf{t}\left(\mathbf{R}_E^B\langle 0\rangle\right) \right) (i = 0, 1, 2 \cdots, N).$$ (16)

Using a single effector position can introduce bias into our estimate of $\hat{\mathbf{P}}_{Pipette}^B$. To address this, we conducted multiple estimations using 10 different effector positions. This approach allowed us to better infer the likely distribution of $\hat{\mathbf{P}}_{Pipette}^B$, which we found to vary within a cube with sides measuring approximately 0.7–0.8 mm (Supplementary Fig. 5e). To align the pipette tip more precisely with the

pivot point, the effector was positioned within a 0.8 mm square cube. The aim of this section is to find the position that minimizes the shift of the pipette tip position during rotation. The procedure consists of eight steps as follows:

Step 1: Start by creating a grid of 64 evenly spaced points within a cube of side $x$ mm. Initially, $x = 0.8$ mm (Supplementary Fig. 5f).

Step 2: Move the effector to the $i$-th point within the grid.

Step 3: Rotate the stage by −7° around the Y axis (referred to as the V rotation), and by −15° around the Z axis (referred to as the W rotation).

Step 4: Measure the pipette shift for both the V and W rotations, denoted as $e_v$ and $e_w$, respectively.

Step 5: Calculate the squared error for the $i$-th position using the formula $e^2\langle i\rangle = e_v^2 + e_w^2$.

Step 6: Repeat Steps 2–5 for all 64 points in the grid and identify the position where $e^2\langle i\rangle$ is minimized.

Step 7: Narrow the search grid to a cube centered around the point identified in Step 6. The new $x$ is set to two-thirds of the previous $x$ (Supplementary Fig. 5g). Return to Step 1 and repeat the process.

Step 8: Repeat Steps 1–7 over four cycles. This allows the possible volume where the pipette tip could exist to be narrowed to a 0.16 mm square cube (Supplementary Fig. 5h).

Performing these procedures decreased the shift to less than 0.03 mm at the rotation that was used for the calibration. However, the error at the maximum of 0.3 mm was still produced at the other angles (Fig. 4g). Finally, we measured $\Delta\mathbf{t}(\mathbf{R}_E^B\langle i\rangle)$ in a grid-based approach for both the V and W rotations, and calculated regression models to estimate $\Delta\mathbf{t}(\mathbf{R}_E^B\langle i\rangle)$ from V and W rotations. First, 40 × 40 pairs of V and W grid search points were arranged within the ranges of ± 14° for V rotation and ± 23° for W rotation. The number of actual sets of angles was approximately 1000, excluding those that could not be moved due to the physical constraints of the robot. The pipette tip movement was measured twice at each search point and averaged. Then, we used the thin plate spline method to construct a regression model,

$$\mathbf{F}(V,W) = (f_x(V,W), f_y(V,W), f_z(V,W))^T,$$ (17)

where $f_x(V,W), f_y(V,W),$ and $f_z(V,W)$ represent the functions of $(V, W)$ to output the shift by rotation of $V$ and $W$ angles in the $X$, $Y$, and $Z$ axes, respectively (Fig. 4h). The shift that was estimated from this model was compensated for by translation of $-\mathbf{F}(V,W)$. Thus, the final error can be written as follows:

$$E = \left\| \Delta\mathbf{t}\left(\mathbf{R}_E^B(V,W)\right) - \mathbf{F}(V,W) \right\|.$$ (18)

By developing these procedures, 99.8% of the shift caused by the rotation was less than 50 μm and the average shift was 0.021 ± 0.0002 mm ($n = 1274$) at any position and any angle within the range of the injection experiments (Fig. 4j, k and Supplementary Fig. 5i, j).

## Registration of laser reflection point to the pipette tip and laser scanning on the cortical surface

A 26 G needle (NN-2613S; Terumo Corporation, Japan) was cut to make a hollow metal rod that was used as the reference point to align the laser reflection point with the pipette tip in the robot base system. The cut needle was fixed in a position slightly above the base plate. The glass pipette tip was moved to the center of the metal rod with the robot, and the robot coordinates at this point were recorded. Next, after the pipette was moved up by the actuator, the laser point was matched to the center of the metal rod by translating the robot stage position, and the coordinates of this point in the robot base system were recorded. From these two coordinates, the transformation matrix of the laser reflection point and pipette tip was obtained.

The laser reflection point was moved to the bregma by eye inspection using Camera S. An area of ± 5 mm × ± 4.5 mm (anterior-

posterior [AP] axis × medial-lateral [ML] axis) centered on the bregma was scanned, with surface points measured every 0.2 mm along the AP axis and every 0.13 mm along the ML axis. The AP axis corresponded to the Z axis in frame {B} and the ML axis corresponded to the Y axis in frame {B}. In the mouse brain, the Allen Mouse Brain Atlas CCFv3 3D template data (https://connectivity.brain-map.org/static/referencedata) were fitted to the laser-scanned 3D data using an affine transformation including rotation, translation, and anisotropic scaling. To minimize the difference between the coordinate along the DV axis (X-axis in frame {B}) measured by laser scanning and the X coordinate predicted from the transformed template data, the affine transformation parameters were repeatedly updated. The 3D interpolation model based on the 3D template data after the affine transformation was calculated using the "scatteredInterpolant" function.

In the marmoset brain, nine parameters of the quadric surface, $P_{ij}$ ($i = 0, 1, 2; j = 0, 1, 2$), were defined by the following equation:

$$Z = P_{00} + P_{10} X + P_{01} Y + P_{11} XY + P_{20} X^2 + P_{02} Y^2 + P_{21} X^2 Y + P_{12} XY^2 + P_{22} X^2 Y^2.$$

These parameters were estimated from the laser-measured data using the robust least squares method. The difference between the X coordinate measured by laser scanning and the X coordinate predicted from the estimated surface was calculated, and the top 1% and bottom 1% of the distribution were removed as outliers. A 3D interpolation model of the remaining measurement data was constructed using the "scatteredInterpolant" function.

## Acquisition of cortical surface images by Camera S
The 2D coordinates of the pipette tip near the center of the image captured by Camera S were recorded. When the cortical surface was imaged with Camera S, the pipette was moved up with the actuator. In the mouse brain, the image was acquired at 0.6 mm intervals 64 times and a total area of approximately 8 × 8 mm was covered (Supplementary Fig. 1a). The X coordinate on the cortical surface with the Y and Z coordinates of one of the predetermined 64 YZ points was calculated from the 3D interpolation model obtained in the previous step (Supplementary Fig. 1a) and the robot was moved so that the point that corresponded to the pipette tip coincided with the position to acquire the image (red point in Supplementary Fig. 1b). In the marmoset, the image resolution was two-thirds lower than that in the mouse brain; images were acquired at 1 mm intervals 106 times, and a total area of approximately 9.8 × 17.3 mm was covered.

## Building the neural network for vessel segmentation
For each image, the area of 3.8 × 3.8 mm centered at the pipette tip coordinates was cropped and the cropped image was used as the input image for SA-UNet. For training of SA-UNet, twenty images of retinal fundus vasculature from the Digital Retinal Images for Vessel Extraction (DRIVE) dataset were initially prepared, and six original images and labels of cortical vessels in mouse brains were further prepared, with these six images before we started to train the SA-UNet. This model worked well for relatively narrow vessels but not for bold vessels or poorly conditioned surfaces. Because we obtained additional images during the development and validation process of ARViS, we decided to add 27 images to the training dataset using the methodology called active learning, which selects more "informative" images to add to the training dataset. EquAL[35], which calculates the difference in prediction between the original and inverted images as "consistency" of the inference, was adopted as a criterion. In addition, the training annotations were based on the prediction of the neural network with manual modifications in a process called CNN-corrected segmentation[36]. This EquAL selection and CNN-corrected segmentation cycle was repeated three times with 33 original images (six images were manually annotated and 27 images were CNN-corrected) and 20 DRIVE dataset images being used to train the neural network. The final

metrics of the SA-UNet (final model) were as follows: MCC, 0.706; F1, 0.774; AUC, 0.941; specificity, 0.881; sensitivity, 0.884; and accuracy, 0.881. All training processes were written and performed in Python and Keras. Training and inference were performed on a GPU (NVIDIA GeForce RTX 2070 SUPER; NVIDIA, USA). The training parameters for the model training were as follows: captured images were cropped to 512 × 512 pixels, the number of filters for the first convolution layer was 16, total epochs was 150, learning rate was $10^{-3}$ for the first 100 epochs decaying to $10^{-4}$ for the last 50 epochs. The DropBlock layer had a probability of 0.82, block size of seven, and batch size for training of four. Data augmentation was performed, including vertical and horizontal flipping, random rotation, color filtering, and gaussian filtering applied to the original images. We reserved 5% of the dataset for cross-validation, and three images were prepared exclusively for the test dataset (Supplementary Fig. 2i,j). The DRIVE dataset contains 24-bit RGB images in Tag Image File Format (TIFF) as training images and 8-bit binary images in Portable Network Graphics (PNG) format as label images. These can be downloaded from https://drive.grand-challenge.org/. Our mouse images were aligned with the DRIVE dataset format and are available at https://github.com/nomurshin/ARViS_Automated_Robotic_Virus_injection_System/tree/main/Data/Final_model_dataset.

The mice that were used for SA-UNet training were different from those that were used in the subsequent mouse experiments. The same final SA-UNet model was also used for the marmoset experiment. Thus, new annotations were not needed when measuring a new animal, at least if it was a mouse or a marmoset. However, lighting conditions were important. If reflective points appeared on the cortical surface, the vessel segmentation was incorrect. The direction and intensity of the light need to be adjusted so that reflective points do not appear.

## Transformation of the original stitched image to the YZ stitched image
First, 64 vessel segmentation images were stitched with the automatic panoramic image stitching method (Autostitch.exe[69]). Thus, in each mouse, one large area of approximately 8 × 8 mm was obtained as the original stitched image. To map the target injection sites determined in the 2D stitched image to the 3D curvature of the cortical surface, it is necessary to compensate for image distortion caused by the difference between the camera angle and laser emission angle. The 3D shape of the cortical surface estimated from the laser scanning was rotated through a combination of V and W rotation parameters ($v$, $w$). The coordinates of the pipette tip (reference point) at each image acquisition, which were assumed to be on the cortical surface, were also rotated by ($v$, $w$), and these points were projected to the 2D plane along the corresponding rotation axis. The geometric transformation from the projected 2D coordinates of the pipette tip to the pipette tip coordinates in the original stitched image was performed using the "fitgeotrans" function, which is a function in the Image Processing Toolbox in MATLAB that fits a geometric transformation to control point pairs. Then, the geometric transformation, i.e., the distance between the transformed point of the pipette tip point and the corresponding point in the stitched image, was calculated. This distance was summed over all acquired images to represent the error. Then, after changing the combination of rotation parameters, the error was calculated again. These procedures were repeated to find the combination of rotation parameters that minimized the error. The combination of the rotation parameters that minimized the error was (−17.47 ± 0.19°, 5.09 ± 0.51°) in seven mice and (−18.18°, 3.241°) in one marmoset. Using the rotation and its geometric transformation that minimized the error, the vessel structures in the original stitched image were mapped onto the 3D curvature of the cortical surface. Finally, this 3D image was projected into the YZ plane (YZ stitched image) to determine the multiple injection sites. In this

process, the Parallel Computing Toolbox in MATLAB was utilized to accelerate the calculation.

## Injection sites planning algorithm

If the virus was injected into the centers of the hexagons of the 2D honeycomb structure as the 2D closest packed structure, the gene was assumed to be expressed at the highest filling rate (or uniformly expressed in space). If the distance between the centers is $d$, the area of the hexagon with a side of $\frac{d}{\sqrt{3}}$ is expressed as $\frac{\sqrt{3}}{2}d^2$ (desired single-injection area). If the distance from each injection site to its nearest injection site was $d$, and the total area of the cortical region including the safety region and the other regions (blood vessel and safety margin regions) was $A$, the number of injection sites $N_t$ was calculated as approximately $(\frac{2A}{d^2\sqrt{3}})$. However, the injection sites could not be set in blood vessel or safety margin regions. Therefore, we determined the distribution of injection sites by repeating the clustering method to approach a honeycomb structure. First, the initial number of sites was set to $N_t + 10$. Then, the safety region was divided into ($N_t + 10$) clusters using k-means clustering and the centroid of each cluster was determined as the injection site candidate. Each cortical pixel within the region including the safety region and the other regions was assigned to the nearest candidate. Then, the total cortical area assigned to each final injection site candidate was calculated as the assigned injection area. If the median value of the assigned injection area did not exceed the desired injection area, the two sites with the smallest and second smallest assigned injection areas were removed and a new candidate site was added. Then, the clustering was performed again. This procedure was repeated approximately 10 times until the median value of the assigned injection area exceeded the desired injection area. Based on the misalignment distribution and the simulation of the bleeding risk probability, the safety margin distance was generally set at 65 μm. $d$ was set to 0.35–0.65 mm.

## Sequential injections into the target injection sites

The order of multiple injection sites was determined by solving the traveling salesman problem. The order of the target points near the bregma was set to be earlier than the order of the points around the edge of the craniotomy. Because we left the dura mater of the mice intact, a larger force was needed to penetrate it, which caused surface dimpling, potentially damaging tissue[25]. Previous studies indicate that a faster insertion speed reduces deformation of the cortex and vessel damage during membrane penetration[70,71]. Therefore, we set the series of pipette tip movements as follows (Supplementary Fig. 6).

Step 1. The pipette tip was set 3 mm above the bregma.

Step 2. It was moved to 1 mm above the target injection point and then moved vertically to 0.2 mm above the target injection point on the cortical surface at a speed of 20 mm/s.

Step 3. The angle of the pipette was changed so that it was perpendicular to the surface.

Step 4. A new coordinate system with the injection point on the cortical surface as the origin was set as frame {I}. The X axis of the device was aligned with the vertical line to the cortical surface.

Step 5. For mice, the pipette tip was pulled 2 mm back along the X axis in frame {I}. The pipette was transiently inserted to a depth of 0.6–0.8 mm at a speed of 6 mm/s, penetrating the dura mater. The pipette tip was pulled back to the target depth (0.3 mm) at 0.1 mm/s and held for a time setting of 15–30 s before the injection.

Step 5′. In the marmoset procedure, the pipette was inserted to the target depth (0.5 mm) at a speed of 0.1 mm/s and held for a time setting of 5–10 s before the injection.

Step 6. The virus solution was injected.

Step 7. The pipette was pulled back 0.2 mm along the X axis at a speed of 0.1 mm/s.

Step 8. The pipette was pulled back 1 mm along the X axis at a speed of 20 mm/s.

Step 9. The pipette was moved to 4 mm above the injection point and the angle reset.

The cycle of Steps 2–9 was repeated for subsequent injection sites. For the safety of the subject animal and experimenter, we prepared an interactive dialog box that allowed the injection to be aborted.

## Realtime closed-loop evaluation of vessels

To ensure that the glass pipette did not hit the vessel position when the 3D position determined as the injection site corresponded to the vessel position at the time of injection, an optional closed-loop function to avoid vessels was implemented. This closed-loop function is based on a simple classifier using RGB values of pixels. SA-UNet robustly generates pairs of an RGB image and a vessel labeled image for each experiment. Based on this data, a logistic regression model was created to classify the pixels as to whether they were recognized as vessels or not by SA-UNet. Training was performed using the "fitglm" function in Matlab with five-fold cross validation. The metrics of the regression models for seven mice were: accuracy, $0.923 \pm 0.007$ (mean ± s.e.m.); precision, $0.857 \pm 0.008$; recall, $0.674 \pm 0.021$; F1-score, $0.754 \pm 0.015$.

The needle tip was continuously monitored during the injection process. As the needle approached the surface, it paused when the tip was situated 0.2 mm above the designated insertion point. At this moment, an image of the cortical surface was captured. An area of $8 \times 8$ pixels directly below the needle tip was then extracted and subsequent classification of these pixels was carried out. If the pixels were identified as blood vessels, the insertion was immediately cancelled. Following this, pixels were extracted in three different directions, excluding the negative Y-axis direction where the surface was overlaid with the glass pipette. These pixels were classified in the same manner as before. The robot was set to move a random distance ranging from 0.05 mm to 0.15 mm in a direction not identified as vascular structure. Then, the needle insertion process was set to be re-initiated.

## Animals

All animal experiments were approved by the Animal Experimental Committee of the University of Tokyo. *C57BL/6* mice (8–14 weeks old; Japan SLC) were used. All mice were provided with food and water ad libitum, and housed in a 12:12 h light-dark cycle (light cycle; 8 AM–8 PM). Ambient temperature and humidity were 22–25 °C and 40–80%, respectively. One laboratory-bred common marmoset (*Callithrix jacchus*, female, 2 years and 5 months old) was also used in the present study. The marmoset was kept on a 12:12 h light-dark cycle and was not used for other experiments prior to the present study.

## Head plate implantation for mice

Head plate implantation was performed as the protocol described previously[47]. Mice were anesthetized by intraperitoneal injection of a mixture of ketamine (74 mg/kg) and xylazine (10 mg/kg), followed by topical administration of eye ointment (Tarivid; 0.3% w/v ofloxacin, Santen Pharmaceutical, Japan) to prevent eye-drying and infection. During surgery, body temperature was maintained at 36–37 °C with a heating pad. The head of each mouse was sterilized with 70% ethanol, the hair was shaved, and the skin covering the neocortex was incised. The exposed skull was cleaned, and a head plate (Tsukasa Giken, Japan) was attached to the skull using dental cement (Fuji lute BC; GC, Japan; and Bistite II; Tokuyama Dental, Japan). The surface of the intact skull was coated with dental adhesive resin cement (Super bond; Sun Medical, Japan) to prevent drying. An isotonic saline solution containing 5 w/v% glucose and the anti-inflammatory analgesic carprofen (5 mg/kg, Remadile; Zoetis, NJ, USA) was injected intraperitoneally after all surgical procedures. The number of mice per cage was 2–5

before the head plate was attached, and then after attachment the mice were housed singly to avoid damage to the head plate and glass window.

## Head plate implantation in the marmoset

All surgical procedures on the marmoset were performed under aseptic conditions, as described previously[21]. Following anesthetization, the marmoset was placed in a stereotaxic instrument (SR-6C-HT; Narishige, Japan), with anesthesia maintained by inhalation of isoflurane (1.5–4.0% in oxygen). Oxygen saturation (SpO2), heart rate, and rectal temperature were monitored continuously. The antibiotic ampicillin (16.7 mg/kg) and the antiemetic maropitant (1000 mg/kg) were administered intramuscularly. The anti-inflammatory drug carprofen (4.4 mg/kg) was intramuscularly injected to reduce pain and inflammation in the perioperative period. Acetated Ringer's solution (10 mL) containing riboflavin sodium phosphate (200 μg) was also administered subcutaneously. Hair was removed from the head using a depilatory, and the head was sterilized with povidone-iodine, followed by exposure of the skull. Lidocaine jelly was applied to wound sites to reduce pain. A head plate was attached to the skull using universal primer (Tokuyama Dental), dual-cured adhesive resin cement (Estecem II; Tokuyama Dental), and dental resin cement (Superbond).

## Virus production

pGP-AAV-syn-jGCaMP7c variant 1513-WPRE was a gift from Douglas Kim & GENIE Project (Addgene viral prep # 105321-AAV1; http://n2t.net/addgene:105321; RRID: Addgene_105321)[72]. The AAV plasmid of human synapsin I promoter (hSyn)-tetracycline-controlled transactivator 2 (tTA2) was constructed by subcloning the DNA fragments containing hSyn and tTA2 into pAAV-MCS (Agilent Technologies, CA, USA). The AAV vector was produced as described previously[21,42,73]. The generation of pAAV-TRE-GCaMP6s-P2A-GCaMP6s-WPRE (tandem GCaMP6s) is described in detail in Matsui et al.[14]. AAV plasmids were packaged into AAV serotype 9 using the AAV Helper-Free system and AAV-293 cells (240073, Agilent Technologies).

## Virus injections into the mouse cerebral cortex

Following anesthetization, a craniotomy measuring $8 \times 8$ mm and centered at the bregma was made in the skull of each mouse, with the dura mater being left as intact as possible. This was because the dura mater is thin enough to allow clear observation of the vasculature, and removing it posed a risk of damaging the vessels. However, keeping the dura intact increased the force needed for pipette insertion, which can lead to larger surface dimpling and potential tissue damage[25]. Additionally, the intact dura can cause the needle to slip if there is a significant angle discrepancy between the surface normal and the insertion angle. This is one of the reasons why we introduced angle adjustments for the needle during insertion. Anesthesia was maintained throughout surgery by inhalation of 1% isoflurane. To prevent cerebral edema, dexamethasone sodium phosphate (1.32 mg/kg; Decadron, Aspen Japan, Japan) was administered intraperitoneally 30 min before the start of the craniotomy. The AAV solution (AAV1-syn-jGCaMP7c; $7.86 \times 10^{12}$ vg/mL) was subsequently injected using a pulled quartz-glass capillary (broken and beveled to an outer diameter of 25–30 μm; Nakahara Opto-Electronics Laboratories, Inc., Japan) and an injector (Nanoject III). The capillary was carefully manufactured by sharpening it for more than 30 min to minimize physical damage to the cortical tissue during insertion[25]. The glass capillary was filled with mineral oil (Nacalai Tesque, Japan) and the solution of virus was drawn up from the tip prior to injection.

The number of AAV injection sites was 74 and 72 in two male mice (12 and 14 weeks old) with the injections in one hemisphere and 92, 128, 138, and 125 in four mice (three males [12–14 weeks old] and one female [14 weeks old]) with the injections in both hemispheres. The volume of each AAV injection was 30 nl/site; therefore, the total injection volume was $3.1 \pm 0.4$ μl. This volume was larger than that in our previous study, in which AAV was injected into only the primary motor cortex (0.25–1 μl)[74]. However, if the injected two-dimensional area was considered (the cortical depth was not considered), the injected solution density in the current mouse experiments was approximately 60.5 nl/mm$^2$ (assuming the craniotomy area was $4 \times 8$ mm in one hemisphere and $8 \times 8$ mm in both hemispheres), and this value was less than that in a previous report (79.6–318.5 nl/mm$^2$; the craniotomy was a circle of radius 1 mm). Thus, the virus injection volume used in the current system was not much larger than that used in our previous study using mice.

To validate the accuracy of the robotic manipulation, fluorescent dye (Vybrant™ CM-DiI Cell-Labeling Solution; Invitrogen, CA, USA) was injected in a male mouse instead of virus solution. The pipette was inserted into the target (cortical depth of 500 μm) and held for 15 s, and 3 nL of the dye stock solution were injected into each site at a rate of 1 nL/s with a 1 s interval, followed by a 30 s wait until the next injection. The total volume of dye was 150 nl. After all injections were finished, a mixture of ketamine (74 mg/kg) and xylazine (10 mg/kg) was administered intraperitoneally, and the cortical surface was viewed with an epi-fluorescent microscope (THT macroscope; Brainvision, Japan) to determine the fluorescent spots. Subsequently, the animals were deeply anesthetized for slice preparation.

During the multiple-injection procedure, the experimenter observed the cortical surface through Camera S in real-time. Long bleeding was visible to the experimenter. The number of occurrences of short bleeding was counted by checking the video acquired by Camera S after the injection procedure had ended. It was reported that even when laser irradiation induces micro-bleeding with ~100 μm diameter hematoma in the mouse cortex, which is comparable to the short bleeding in the current study, significant neural pathology does not occur, and the activity of neurons and astrocytes near the hematoma is almost restored within a day[75,76]. Therefore, we considered that some occurrences of short bleeding were acceptable and the effect of slight bleeding that we could not detect on the cortex was negligible.

## Virus injections into the frontoparietal cortex in the marmoset

For wide-field one-photon calcium imaging of a marmoset, a rectangular craniotomy of $8 \times 16$ mm was made over the right dorsal cortex from area 8A to area PE, and the dura mater was removed. The AAV in the pipette was injected as described above. The virus solution consisted of AAV9-hsyn-tTA ($5.38 \times 10^{12}$ vg/mL) and AAV9-TRE3G-tandem-GCaMP6s ($1.365 \times 10^{13}$ vg/mL). The pipette was inserted to a depth of 500 μm from the cortical surface and held for 2.5–5 s, and the virus solution was injected. A total of 30 nL of solution was injected into each site at a rate of 2 nL/s with a 5 s delay after each injection. The total volume of virus solution was 7.98 μL. This was also comparable to that in our previous study using the marmoset (15 μl)[42]. In addition, the injected solution density was estimated to be 62.3 nl/mm$^2$ (7.98 μL/[$8 \times 16$ mm]), which was lower than the 333.3 nl/mm$^2$ in Ebina et al.[42] (15 μl/[$5 \times 9$ mm]). Thus, the total injection volume was comparable to that in our previous study using marmosets. A rectangular glass coverslip of $15 \times 8$ mm (approximately 150 μm thick; Matsunami Glass, Japan) that was attached to the bottom of a polyacetal chamber (height, 1.2 mm) with UV-curing optical adhesive (NOR-61) was pressed onto the cortical surface, and the space between the chamber and the cortical surface was filled with dental cement (Fuji Lute BC). The edge of the chamber was sealed with dental adhesive resin cement (Super bond). An area of $14.2 \times 7.2$ mm, excluding the width of the chamber's frame, formed the field of view for cortical imaging. The marmoset was allowed to recover for 7 weeks prior to imaging. Upon observation, the marmoset exhibited no signs of deleterious immune responses, such as significant body weight loss, reduced eating, or severe fever. Throughout the experiment, the marmoset's body weight remained between 85–95% of its pre-surgery weight of 339 g, which is considered

normal for animals under water restriction for behavioral experiments. The experiment was completed two months after surgery, and during this time the marmoset displayed normal behaviors with no abnormal response, such as epilepsy.

## Histological evaluation

To evaluate tissue damage due to multiple local injections, two male mice in which AAV solution was injected into only the right hemisphere were post-fixed on days 25 and 28 after the injection, leaving time for sufficient expression of GCaMP7c. During the fixation, the mice were deeply anesthetized by intraperitoneal administration of ketamine (74 mg/kg) and xylazine (10 mg/kg), and were then perfused transcardially with PBS, followed by 4% formaldehyde (Wako, Japan). Their brains were removed, post-fixed with the same fixative overnight, and sectioned. To visualize cytoarchitecture, fluorescent nucleus staining was performed with DAPI (1:1000; D8417, Sigma-Aldrich). Images were acquired under a fluorescence microscope (Apexview APX100; Olympus, Japan) equipped with a 10× objective lens and fluorescence filters (U-FUNA, U-FGFP; Olympus). All the sections were obtained with the same LED intensity and exposure time. For each slice, the brightest area of $500 \times 500 \, \mu m$ (in green) and its corresponding area in the left hemisphere were chosen for DAPI counts. DAPI signals were counted as particles above 5 pixels after applying the 'Auto Local Threshold' command with the Niblack algorithm and the 'Watershed' command to the images, which are involved in Fiji[77]. To evaluate the spread of each injection, we chose nine slices with relatively isolated injections in the most lateral position and measured the fluorescence profile along the medial-to-lateral axis in layer 2/3. We assumed that the injection fluorescence that was relatively distant from other fluorescence spots was derived from the most lateral injection sites. The baseline fluorescence was calculated as the 7th percentile of the measured signal, and the half fluorescence value was the average of the baseline and the peak value of the Gaussian-smoothed fluorescence with a sigma value of 10. The half-decay distance of the fluorescence was defined as the distance between the peak point and the point where the Gaussian-smoothed fluorescence signal fell below the half-decay fluorescence value. The half-decay distance calculation was performed in MATLAB (R2024a; MathWorks). All the other analyses were performed using Fiji[77].

## Measurement of the insertion points and depths in the mouse cortex

The location of each insertion was observed as CM-DiI fluorescence using an epifluorescence microscope. Elastic and consistent image registration using bUnwarpJ[78] was applied to the fluorescence and surface vascular pattern images to determine the injection sites. Because the registration was not accurate at the periphery of the images, the images were cropped around the targeted injection site before application of the algorithm. The distance between the center of fluorescence and the planned injection site was measured on the registered image using Fiji[77]. If the insertion point was observed as a shadow surrounded by fluorescence, the center of the observed shadow was used for measurement, instead of the center of fluorescence.

The trajectory of the pipette insertion within the cortex was traced according to the fluorescence derived from CM-DiI in the slice. Each mouse was deeply anesthetized with a mixture of ketamine and xylazine, and then transcardially perfused with PBS (10 mL) followed by 4% formaldehyde (30 mL). Their brains were post-fixed with the same fixative overnight at 4 °C and then coronally sectioned at a width of 100 μm. The sections were counterstained with a fluorescent Nissl stain (NeuroTrace blue; 1:500; Invitrogen). Images were acquired with an epifluorescence microscope (BX53F; Olympus) and a digital CCD camera (RETIGA2000R; QImaging, Canada). Since the plane of the slice and the trajectory of the injection did not always coincide, fluorescence derived from the same injection was sometimes observed

across multiple slices. Therefore, the putative depth of the pipette tip was estimated as follows. First, the distance from the deepest point of the pipette track (observed as fluorescence) to the surface was measured for each slice. If the pipette trajectory was visible as a shadow surrounded by fluorescence, the distance from the deepest point of the shadow to the surface was measured. Adjacent slices were compared to determine whether the trajectories originated from the same injection. These processes were performed using Fiji[77].

## Estimation of bleeding risk probability

The distribution of misalignment of the actual injection sites from the target injection sites (32 sites from one male mouse) was fitted using a non-parametric Gaussian kernel restricted to positive values. Based on the data, the following methods were used to determine the relationships between the distance of the injection sites from blood vessels and the hemorrhage probability. First, in one cerebral cortex image obtained with Camera S, the distance between each target injection site was set to 0.4 mm and the total number of injection sites per trial was set to approximately 120. The safety margin distance (the minimum distance of the target site from blood vessels), $D$, was specified (0–80 μm). In this condition, using the clustering methods as described, the target injection sites were determined. Next, to determine the simulated injection site, the position of the target site was shifted according to the estimated misalignment distribution. Those simulated injection sites that hit a vessel were considered as bleeding sites and their number was counted. The simulation was conducted 100 times for each $D$ value. Finally, the proportion of the simulated injection sites that hit the vessels was averaged for each $D$ value.

## Comparison of the blood vessels using microscopy and vessel segmentation

To examine the limits of the ability of the SA-UNet to extract blood vessels from the camera image, the results of the vessel segmentation were compared with vessels observed under 5× epi-illumination microscopy. One microscopic image of $1 \times 1$ mm size was taken and vessels were masked manually. The following process written in MATLAB (R2021b; MathWorks) was then used to measure the thickness. 1. Distance transformation was applied to the vessel mask. The brightness values of the pixels in the mask were determined according to their distance from the nearest non-zero pixels. 2. The vessel mask was skeletonized to obtain the furthest line from the boundary. 3. The distance map and the skeletonized vessels were multiplied. This yielded a skeleton of the blood vessel mask where the pixel brightness corresponded to the vessel thickness (skeletonized distance map). The vessel mask obtained by the SA-UNet was also prepared for this analysis. The actual output from the SA-UNet is an 8-bit grayscale image, and this was binarized by applying the threshold (205) that was used to determine the injection site. Then, projective transformation was applied to the segmentation result to match the microscopic image using manually extracted feature points. The thickness that the SA-UNet was able to recognize was obtained by calculating the overlaid area between the segmentation and the skeletonized distance map.

## Wide-field calcium imaging of the marmoset cortex

In the marmoset, images were acquired using a fluorescent zoom macroscope equipped with a high-numerical aperture macro lens (PlanNEOFLUAR Z 1.0×, NA 0.25; Carl-Zeiss, Germany) and an EM-CCD camera (iXon Ultra 888; Andor Technology, UK). The microscope body with the objective was rotated 16.5° around the anterior-posterior axis and the marmoset chair was tilted 7° so that the optical axis was nearly perpendicular to the glass window. Images of $512 \times 512$ pixels were captured using a Solis camera system (Andor Technology) at a sampling rate of 30 frames/s. Optical zoom magnification was set to cover the entire window in the marmoset. The center wavelength of the LED for the wide-field calcium imaging was set to 470 nm, and the light

intensity under the device was 6 mW. The animal was trained to sit in an awake head-fixed condition, and the position of a spout was adjusted so that the marmoset could lick it. A series of 18000 images were acquired while 40 μL of sucrose water was delivered as a reward every 10–20 s. No apparent fading of the fluorescent signal was observed over a duration of several days under this experimental condition.

## Two-photon calcium imaging of the marmoset cortex

The marmoset cortex was imaged with a custom-built two-photon microscopy (Olympus) system equipped with a water immersion objective lens (Olympus XLPLN10XSVMP, NA 0.6; working distance 8 mm) and an Nd-based fiber-delivered femtosecond laser (Femtolite FD/J-FD-500, pulse width of 191–194 fs, repetition rate of 51 MHz; IMRA, USA) at a wavelength of 920 nm. Details of this implementation were described previously[21]. The optical axis was adjusted to be nearly perpendicular to the cranial window by tilting the microscope body 4.8° and the marmoset chair 5°. To shield the microscope objective from possible stray light, an aluminum foil dish was attached to the implanted metal chamber using silicone elastomer (Dent Silicone-V, Japan), and the space over the animal's head was covered with light-proof cloth. The head-fixed animal was habituated under the microscope. A series of 18000 images were acquired while 40 μL of sucrose water was delivered as a reward every 10–20 s. The imaging field area was 636 × 636 μm and 512 × 512 pixels. Two different regions were imaged on different days.

## Identification of cortical areas in the marmoset

To infer the imaging areas of the marmoset frontoparietal cortex, intracortical microstimulation was performed 1 week after the imaging experiments, as described previously[42]. The glass window was removed and a tungsten microelectrode with an impedance of 0.5 MΩ (World Precision Instruments, Sarasota, FL, USA) was inserted to a depth of 1.5 mm from the cortical surface, and a train of 15 cathodal pulses (0.2 ms duration at 333 Hz) were applied. Current ranging from 10 to 100 μA was sequentially applied for each region. The minimum current required to elicit body part movements, as confirmed by an expert experimenter, was defined as the threshold current for that stimulation site. The type of body part movement (extension, flexion, or rotation) induced was determined by visual inspection by the experimenters. The cortical map in Fig. 6d and Supplementary Fig. 8c was determined as follows. First, we identified the cortical area that had a relatively low movement threshold in comparison with more posterior or anterior areas. We considered this area to be the main part of M1. Then, the border between M1 and the caudal part of the dorsal premotor cortex (area 6DC) in this animal was defined as the line at which the threshold current apparently increased along the posterior-to-anterior direction. The marmoset cortical map by Paxinos et al.[79] was aligned with the dorsal cortex of this animal so that the midline and the border between M1 and area 6DC matched between the Paxinos map and this animal.

## Evaluation of the brain pulsation of the marmoset

During the measurement of the marmoset cortical surface, the laser distance sensor moved in two phases: 0.1 mm steps in the Y direction and 8-mm steps in the Z direction. To isolate small fluctuations in the distance between the laser distance sensor and cortical surface, a continuous period where the laser distance sensor was at the most medial and lateral positions without moving in the Z direction was defined as a segment for further analysis. To remove signal changes due to laser distance sensor movement during this period, a linear regression was performed using the XYZ positions of the laser distance sensor to fit the measured depth of the cortical surface in 36 segments (both medial and lateral sides) and extract their residuals (Supplementary Fig. 9d). The peaks and troughs of the moving average of the

residual signals with a minimum prominence of 0.02 s and a minimum separation of 0.67 s were detected. The peak-to-peak intervals and amplitude of the oscillation were calculated for each segment. Assuming that brain pulsation caused by heartbeat is mostly perpendicular to the cortical surface[46], and that the cortical surface curvature is negligible, the perpendicular pulsation component was approximated by multiplying the sensor-measured vector by the cosine of the angle between the perpendicular vector of the cortical surface and the sensor trajectory (Supplementary Fig. 9g). Finally, the possible maximal displacement of the reference position in a plane perpendicular to the optical axis of Camera S was estimated by multiplying the amplitude of the perpendicular pulsation component by the sine of the angle between the normal vector of the brain surface and the optical axis of Camera S (Supplementary Fig. 9i).

## Monitoring licking and body movement

Two CMOS cameras (DMK33UP1300; ImagingSource, Taipei, Taiwan) equipped with a fixed-focus lens (focal length, 3.0 mm; #89-341; Edmund, NJ, USA) and a varifocal lens (focal length, 2.7–12.0 mm; Spacecom, Japan) were set at a distance of 270 mm and an angle of 35° from the front of the marmoset face. During the imaging experiment, two 480 × 480-pixel images were simultaneously obtained at 50 frames/s. Licking was detected as a change in the average luminance within a region of interest (ROI) set around the mouth using ImageJ software (National Institute of Health, USA), followed by processing in MATLAB (R2022a; MathWorks). A high-pass filter was applied to the signal to remove baseline trends and it was dichotomized at a threshold of two standard deviations.

## Processing of wide-field calcium imaging data for the marmoset

The data obtained from the marmoset across three sessions on different days were analyzed using custom MATLAB routines (R2021a, R2022a; MathWorks). Fluorescence data were obtained as 512 × 512-pixel tiff images, which were cropped to 241 × 442-pixel images prior to further processing. The time-varying baseline fluorescence of each pixel ($F$) was estimated for a given time point as the 10th percentile value over 30 s around it, and the relative fluorescence change ($\Delta F/F$) of each pixel was calculated. Because three sessions were performed on different days, the images were registered across sessions. The average frames of each imaging session were used for registration. The averaged frame of the first session was regarded as the reference image for alignment. A geometric transformation matrix was generated to align the average frames of the second and third sessions to the reference image.

To assess the functional connectivity that was not directly related to movements, we extracted the neuronal activity during the quiet state, which we defined as the time period excluding that from 0.3 s before to 2 s after the occurrence of licking or body movement. Then, correlations in the fluorescence change between the areas during this quiet state were calculated. Hierarchical clustering was applied to the resulting correlation matrix, using the Calinski-Harabasz criterion, and two distinct clusters were identified. Seed correlation maps were created using the ROIs of areas PFG and 4ab as seed points. The correlation coefficients between these seeds and all other pixels were visualized as seed correlation maps.

A linear regression model was constructed using a design matrix composed of multiple variables. Each variable was generated as follows. For the "body_analog" measurement, the absolute difference between two adjacent frames was calculated, and its average within that frame was regarded as the motion intensity at that time. The z-scored value of the motion intensity was assigned to the "body_analog" variable. The "body" variable was binarized "body_analog" data with a threshold value of 2. For the "eye_analog" measurement, ROIs were placed around the pupils of both eyes, and movement traces were extracted using DeepLabCut[80]. The position of each eye was then

determined by calculating the average position of the ROIs around its pupil. To quantify the intensity of eye motion, the speed of each eye's movements was averaged. This calculated intensity was then normalized using z-scoring and recorded as the 'eye_analog' variable. The "eye" variable was derived from this "eye_analog" data, binarized with a threshold value of two. For the "eyelid_analog" measurement, ROIs were placed on eyelids of both eyes, and movement traces were extracted using DeepLabCut. Subsequently, the z-scored values of their vertical coordinates were averaged and assigned to the "eyelid_analog" variable. For the "jaw_analog" measurement, an ROI was placed on the jaw and the movement traces of the jaw were extracted using DeepLabCut. Then, the speed was calculated to quantify the intensity of jaw motion. The z-scored value of this intensity was assigned to the "jaw_analog" variable. The "jaw" variable was derived from the "jaw_analog" data, binarized with a threshold value of 0. For the "lick" measurement, an ROI was placed on the tongue, and its appearances were extracted using DeepLabCut. The times when the tongue was recognized were assigned a value of 1, while all other times were assigned a value of 0. For each vector of the "body", "eye", "jaw", and "lick" variables, time-shifted copies were generated, covering frames from 1 s before to 3 s after each event. For the reward variable, a vector containing a pulse at the time of reward delivery was copied in time, spanning the frames from the timing of the reward delivery to 3 s after each event. The "body_analog", "eye_analog", "eyelid_analog", and "jaw_analog" variables were not subjected to time-shifting.

A ten-fold cross-validated linear regression was used to fit the design matrix to the fluorescence signals of each pixel, using ridge regularization and stochastic gradient descent solver. Subsequently, the variance explained ($R^2$) by the full model was calculated. The selection of lambda for ridge regression was as follows: Fifteen candidate lambda values that ranged from $10^{-4}$ to $10^{-1}$ on a logarithmic scale were used to predict the time series data of a representative pixel's fluorescence signal in a 10-fold ridge regression model. We then plotted the mean squared error and frequency of non-zero coefficients for each lambda value. The value of 0.0139 was chosen as it adequately balanced the model's accuracy and the sparsity of coefficients. We confirmed the adequacy of this lambda value by verifying that the mean squared errors for other pixels were also comparably low. To estimate the explanatory power of each variable, shuffled models were created by randomizing all time points for only the specified variable group. The difference in the explained variance between the full and the shuffled model was considered as the unique contribution ($\Delta R^2$) of that variable group. The "mouth" variable group included the "jaw", "jaw_analog", and "lick" variables, while the "body", "eye", and "eyelid" groups included their respective analogue and binarized variables.

### Processing of two-photon calcium imaging data for the marmoset

The observed images were registered with rigid motion correction using NoRMCorre[81] and their contrast was adjusted in field 1 and field 2. Then, the CNMF algorithm[82] was applied to each set of data to obtain ROIs of active neuronal somata. The relative change in fluorescence for each ROI at a time point $t$ was calculated as $\Delta F/F(t) = \frac{F(t) - F_0(t)}{F_0(t)}$, where $F(t)$ was the mean of the fluorescence intensity value of the pixels within each ROI at $t$, and $F_0(t)$ was the baseline fluorescence at $t$ calculated as the 8th percentile of the $F(t)$ value across $t \pm 15$ s. The max $\Delta F/F$ values of the ROIs were statistically compared between two regions.

Linear regression models were constructed to analyze the relationships between the time-shifted neuronal activity and each behavioral variable (licking and body movement). To capture the potential influence of neuronal activity on the behavior, and its temporal dynamics, we created time-shifted versions of the activity data for each neuron. For each session, we used a time frame range from −30 to +30

(or ± 1 s range) to shift the activity data. This process resulted in 61 time-shifted traces for each neuron, including the original time point. These shifted traces were then aligned to form a new matrix for each neuron, with dimensions of 18000 frames by 61 shifted activity traces. A linear regression model was fitted for each neuron, using its shifted activity matrix as predictors and the behavioral variable as the response variable. For licking decoding, a Gaussian filter with a 30-frame window was applied to the binarized lick signal and the filtered signal was used as the response variable. For body movement decoding, the motion intensity was used as the response variable. The predictive power of each model was assessed using $R^2$, which quantified the variance in the behavioral variable explained by the neuronal activity.

### Statistics and reproducibility

Statistical analyses were performed using MATLAB (R2021a, R2022a; MathWorks). Comparisons were made using Welch's test, paired t-test (two-sided), and one-way ANOVA followed by post-hoc one-sided Student's t-test with Bonferroni correction. No statistical tests were run to predetermine the sample size. However, sample sizes were estimated according to previous methodically comparable laboratory experiments and were similar to those generally employed in the field. Data are presented as mean ± s.e.m. unless otherwise noted. Blinding and randomization were not performed. For the data shown in Fig. 2c, the experiment was performed once. For one-photon imaging of the marmoset shown in Fig. 6b, three sessions were obtained on different days from one marmoset. For two-photon imaging of the marmoset shown in Fig. 8c, e, two sessions were obtained on different days from one marmoset.

### Reporting summary

Further information on research design is available in the Nature Portfolio Reporting Summary linked to this article.

## Data availability

All data supporting the findings of this study are available within the article and its supplementary files. Any additional requests for information can be directed to, and will be fulfilled by, the corresponding authors. Source data are provided with this paper. The imaging data generated in this study for training SA-UNet are deposited in the GitHub under accession code https://doi.org/10.5281/zenodo.13220760 https://github.com/nomurshin/ARViS_Automated_Robotic_Virus_injection_System/tree/main/Data/Final_model_dataset.

## Code availability

All computer codes to realize the ARViS have been deposited in the GitHub under accession code https://doi.org/10.5281/zenodo.13220760 https://github.com/nomurshin/ARViS_Automated_Robotic_Virus_injection_System.

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

## Acknowledgements

We thank M. Nishiyama, Y. Hirayama, M. Hirokawa, and Y. Takahashi for animal handling, and Y. Watakabe, M. Takaji, and T. Yamamori for providing AAV constructs. This work was supported by Grants-in-Aid for Scientific Research on Innovative Areas (17H06309 to M.M., 19H05307 to T.E., and 21H00302 to T.E.), for Transformative Research Areas (A) (22H05160 to M.M. and 23H04977 to T.E.), for Scientific Research (A) (19H01037 and 23H00388 to M.M.), for Scientific Research (B) (20H03546 to T.E.), and for Early-Career Scientists (20K15927 and 23K14284 to S.-I.T) from the Ministry of Education, Culture, Sports, Science, and Technology, Japan; AMED (JP19dm0207069, JP15dm0207001, and JP18dm0207027 to M.M., JP21dm0207014 to K.O, JP23wm0625001 to K.O. and M.M.; JP19dm0207085 to T.E. and M.M.); and the Tokyo Society of Medical Sciences (to S.-I.T. and T.E.). This work was also supported by the program for Brain Mapping by Integrated Neurotechnologies for Disease Studies (Brain/MINDS) from AMED under Grant number JP21dm0207111.

## Author contributions

S.N., S.-I.T., and M.M. designed the experiments. S.N. and S.-I.T. constructed the robotic system and conducted mouse experiments. S.N. performed most of the programming, optimized and manipulated the robot system, and analyzed the data. T.E. conducted surgery and imaging for the marmoset experiments. M.U., Y.M., and K.O. designed and prepared AAVs. S.N., S.-I.T., and M.M. wrote the paper, with comments from all authors.

## Competing interests

The authors declare no competing interests.
