## [Peer Review File · Nature Communications]

REVIEWER COMMENTS

Reviewer #1 (Remarks to the Author):

In this paper, Nomura and colleagues present a method for injecting a given biosensor across many sites in the dorsal cortex of mice and marmosets. They paid special attention to avoiding vasculature and achieving perpendicular insertion using robotic control of the injection arm.

I found this to be intriguing technology, and very much appreciated the foray into the marmoset brain, as a larger brain will require more injection sites, which both increases bleed probability and becomes painfully time-consuming for experimenters. I was impressed by the careful consideration of accuracy, safety, and procedure duration. I think that as we try to achieve transfection of more and more of the cortex, this could be an incredibly valuable tool.

I have just a few suggestions for improvement:

-The immediate focus on genetically encoded calcium indicators in the introduction seemed off. I understand that calcium imaging was the first application they were using, but I could really see this as a tool for the injection of viruses for many purposes, including optogenetics and chemogenetics. I would recommend opening in a more general way, to reach the right audience.

-I was a little bit confused about how the technology interacts (or does not interact) with the dura mater. It seems as though it penetrates through the dura mater for the mice, but does not for the marmoset (because the dura was removed). How does this impact the vascular imaging and the injection process?

-I believe that the mice have been sacrificed and histology performed; it looks like the marmoset has not. It would be nice to show some mouse histology in the supplement, to give a sense of the cortical damage and the spread of each injection.

-Although I could find the injection volumes for the mouse dye experiment (3nl/site—it would be nice to also say the total volume, though) and the marmoset experiment (30nl/site—same thing), I could not find the number for the mouse virus experiment. Are these particularly large total volumes for these species? Was there any sign of a deleterious immune response? And, is the marmoset still healthy?

-It would be helpful to add a bit to the Discussion about potential applications to rats and macaques.

-I visited the github page, and was so pleased to find much more than was promised! From the perspective of a potential user, I recommend explaining in the manuscript that you have a wiki page with a full parts list, assembly guide, and workflow instructions, along with the code.

Reviewer #1 (Remarks on code availability):

I looked at what was available, but I did not actually open the code.

Reviewer #2 (Remarks to the Author):

COMMENTS FOR AUTHORS

Nomura et al. developed an Automated Robotic Virus injection System (ARViS) which can express genetically encoded fluorescent sensors in multiple brain areas. ARViS consists of two key technologies: image recognition of vasculature structures on the cortical surface to determine multiple injection sites, and robotic control of micropipette insertion perpendicular to the cortical surface with 50- μm precision. ARViS successfully achieved 266-site injections over the frontoparietal cortex of a common marmoset, and the authors demonstrated *in vivo* calcium imaging in the marmoset frontoparietal cortex. As also shown in the study by Jendritza et. al. (Nature Comm, 2023, 14:577), this is one of the important steps for the understanding of the neural networks in the marmoset cortex. However, there are several limitations to consider for the publication in Nature Communications.

Firstly, while ARViS identifies the injection sites with minimal bleeding through image recognition of vasculature structures, multiple injections may have negative impacts not only on blood vessels but also on various cells including neurons and the extracellular matrix (ECM) within the cortical tissues. The authors should validate through histological studies to ensure that multiple injections by ARViS do not compromise the normal structure and function of cortical tissues. Also, it is crucial to demonstrate the optimal density and number of injection sites that allow cortex imaging with

minimized negative issues. Overall, the authors need to show the significant advantages of ARViS over the current method that manually injecting into a few regions.

Secondly, there is a limitation regarding the depth of injection. How deep into the brain region can ARViS facilitate injections? Imaging only at the cortex level restricts the scope of future research for understanding brain functions. Thus, it would be highly significant if the authors could demonstrate that ARViS can inject viruses into multiple brain regions with varying depths, enabling research across different layers of the neural network of the brain.

Lastly, in addition to calcium imaging, it is important to demonstrate the imaging of neurotransmitter receptor sensors, such as dopamine sensors and glutamate sensors.

Reviewer #3 (Remarks to the Author):

In this study, the authors aimed to develop an Automated Robotic Virus injection System (ARViS) for broad expression of a biosensor, that integrates two technologies: one for the image-based identification of vascular structures on the cortical surface to select multiple injection locations without impacting them, and another for the robotic management of micropipette insertion at a 90-degree angle to the cortical surface with precision up to 50 micrometers. In the mouse cortex, ARViS methodically administered virus solution injections at 100 sites over a period of 100 minutes, maintaining a low bleeding risk of only 0.1% per site. Additionally, ARViS efficiently completed injections at 266 sites across the frontoparietal cortex of a common marmoset. Both one-photon and two-photon calcium imaging in the marmoset's frontoparietal cortex were also demonstrated.

This study contains an analysis of good clinical significance but there are several factors concerning the methodology.

Please find some of my comments below:

1. The authors initially note a discrepancy in gene expression levels between transgenic animals and those subjected to high-titer AAV-mediated gene delivery. Could you please provide further insight into the reasons behind this observed difference? Considering that transvenous transfection of AAV may result in diminished bioavailability at critical sites, particularly within the brain, due to potential locoregional uptake and metabolism, as well as non-target organ degradation.
2. In deep learning applications for vessel segmentation, the acquisition of finely-focused surface images were accomplished by automatically adjusting the distance between Camera S and the

cortical surface, capturing data at 62 points across the neocortex. Nonetheless, the potential for minor movements in the anesthetized animal model due to respiration could introduce errors. It is unclear how these types of inaccuracies could be mitigated.

3. Similarly, while the approach described for avoiding blood vessels by meticulously mapping the three-dimensional (3D) structure of the cortical surface and utilizing deep learning for image recognition of vascular structures, complemented by clustering-based optimization for selecting multiple injection sites, is innovative, it does not fully address the inherent complexities of blood vessel avoidance. This technique, although advanced, may not account for the dynamic nature of the vasculature, including variations in vessel depth and the potential for movement or changes over time. Additionally, the reliance on clustering-based optimization does not guarantee the elimination of risk to all blood vessels, particularly in highly vascularized areas where the density and proximity of vessels may pose significant challenges. Further refinement and validation of this method are necessary to ensure its effectiveness and safety in practical applications.

4. The authors describe their initial training of the convolutional neural network (CNN) model using only twenty images of the retinal fundus, which exhibited limited accuracy, particularly for thick caliber vessels. They then supplemented this dataset with six cortical images of the brain. Notably, cortical vessels lack an internal elastic lamina crucial for cerebral autoregulation, unlike retinal vessels, which could compromise the internal validity of their training dataset. Why was it not possible to exclusively utilize mouse cortex vasculature images for training the dataset?

5. The author is requested to provide a more detailed explanation regarding the technique of multiple injections into the mouse dorsal cortex, particularly emphasizing the insertion and withdrawal process. It would be beneficial to illustrate this procedure clearly in a separate figure. Also, the timing of insertion and pullback should be specified to ensure the experimental subject's safety.

6. While the authors suggest that the limitation of ARViS regarding the minimal volume capacity of each pipette ($\sim 4 \mu\text{L}$) and the lengthy calibration time (90 minutes) for glass pipette positioning can be mitigated by employing a syringe with a larger capacity, this solution appears overly simplistic and does not adequately address the complexity of the issues at hand. The suggestion fails to consider the nuanced requirements of precise delivery mechanisms needed for effective and safe injections into the cortical tissue, which involve more than just the volume capacity of the injector. For instance, the use of a larger capacity syringe does not inherently solve the challenge of accurately calibrating the injector to ensure precise delivery at the desired location, nor does it address potential alterations in injection pressure that could affect tissue integrity. Additionally, the proposal overlooks the need for a comprehensive solution that integrates improvements in both the hardware for delivery and the software for control and monitoring, ensuring that the increased volume does not compromise the precision and safety of the injections. Therefore, a more detailed and holistic strategy is required to effectively overcome these limitations.

Reviewer #4 (Remarks to the Author):

The authors present a technical setup and computational controlling workflow for automatic injection of viruses in the marmoset cortex, with the aim of automatizing cortex-wide GECI measurements with approximately uniform distribution of injection sites. The presented system aims to find optimal sites so that bleeding is prevented and micropipette injection is exact and perpendicular. To this end, the injector with micropipette are mounted on a robot stage, which measures the brain surface in the fixated head with skull removed using combined laser distance sensing and cameras.

The proposed methodology is highly significant for measuring spatiotemporal dynamics in the cortex. It provides significant potential to advance the development of detailed atlases of functional organization in animal brains, and improve our understanding of structure/function relationships. It extends existing work by increasing automation levels in the technical setup, and introduces AI algorithms for performing vessel localization in camera images.

The focus of this review is on the image processing part of the system, and the calibration of the pipette tips. I have little expertise regarding virus injection and the protocols and requirements for in-vivo animal experiments.

The paper is well written and organized. The Figures are appropriately chosen and described to support the text effectively. The drawn conclusions are well supported by the work. While the experiments and results are generally appropriate, I suggest some minor revisions before publication.

A shortcoming of the approach is the significant recalibration effort of about 90 minutes after changing a pipette. This aspect is clearly described in the paper, and understandable given the requirements in precision. I was a bit surprised however about the 17 minutes required to finish a particular measurement (l. 167). Is this due to the scanning or processing? I suggest to break down this time estimate into the different measurement and processing parts. E.g. how much of the time is taken for the vessel segmentation? Could that be reduced by using a (better?) GPU?

For identifying the vessel structures, an SA-UNet is used which is trained on an existing retinal vessel dataset. The model choice is not much elaborated, but appears reasonable to me, and while not the single obvious choice it can be considered state of the art for such type of application. Unfortunately, the paper does not provide the necessary information to reproduce the model

training. Essential parameters such as batch size, possible cropping of the training images, learning rate, etc. should be specified. Furthermore, while the text refers to “the modified SA-UNet” (l. 873), I could not identify which modifications exactly were made to the originally proposed network. This should be described more clearly.

Regarding the performance of the model, I am missing a more detailed quantitative evaluation. While some scores are provided (l. 873), I could not capture from the text whether the segmentation was evaluated on a test set and whether any cross-validation was performed, and which images exactly went into the evaluation. The authors report that the pre-trained net didn't work well on small vessels, and improved after adding 6 newly annotated images for re-training. Here I am missing a quantification of the initial performance and the improvement after retraining (e.g. l. 130 quantifies vessel thickness in the final model, but corresponding numbers of the initial model are lacking). It should also be described why 6 images were added to the training set, not less or more (also in line 865, does “several images” refer to those 6?). Did you see a saturation point in the performance, or is the annotation effort too high to try out more? It would also be helpful that the paper discusses briefly whether new annotations are likely needed when measuring a new subject, or setting up a new installation of the apparatus with e.g. different lighting conditions.

Some more minor issues:

- L. 888: What is the fitgeotrans function? Which software? Matlab?
- L. 108: Refer to infrared laser as "infrared laser sensor", as it is later (145) referred to as "laser".

Reviewer #5 (Remarks to the Author):

We thank all the reviewers for their careful consideration of our manuscript and for making helpful comments. Our detailed responses (in black) to the reviewers' comments (in blue) are provided below.

Reviewer #1 (Remarks to the Author):

In this paper, Nomura and colleagues present a method for injecting a given biosensor across many sites in the dorsal cortex of mice and marmosets. They paid special attention to avoiding vasculature and achieving perpendicular insertion using robotic control of the injection arm.

I found this to be intriguing technology, and very much appreciated the foray into the marmoset brain, as a larger brain will require more injection sites, which both increases bleed probability and becomes painfully time-consuming for experimenters. I was impressed by the careful consideration of accuracy, safety, and procedure duration. I think that as we try to achieve transfection of more and more of the cortex, this could be an incredibly valuable tool.

Thank you for appreciating our meticulous efforts to achieve high accuracy and safety with our method. Reviewer #1 also recognized our contribution to the neuroscience community through the Github pages, which we found highly encouraging.

I have just a few suggestions for improvement:

-The immediate focus on genetically encoded calcium indicators in the introduction seemed off. I understand that calcium imaging was the first application they were using, but I could really see this as a tool for the injection of viruses for many purposes, including optogenetics and chemogenetics. I would recommend opening in a more general way, to reach the right audience.

We have re-written the second and third sentences in the Introduction as follows:

“Furthering their use, the recent development of other genetically encoded biosensors and optogenetic and chemogenetic tools has opened new opportunities to study the spatiotemporal dynamics of neuronal and glial activity and synaptic transmission (Marvin et al., Nat. Methods 15, 936–939, 2018; Sun et al., Cell 174, 481–496, 2018; Sternson & Roth, Ann. Rev. Neurosci. 37, 387–407, 2014; Kim et al., Nat. Rev. Neurosci. 18, 222–235, 2017). . If these probes could be expressed uniformly across multiple broad cortical areas in non-human primates (NHP), it would be possible to measure and manipulate the spatiotemporal dynamics of the inter-areal

spreading of a variety of activities occurring in motor, cognitive, and sensory processing, activities that are well developed in primates (Galvan et al., J. Neural. Transm. 125, 543–563, 2018; Matsuzaki & Ebina, Curr. Opin. Neurobiol. 64, 103–110, 2020; Kalaska, F1000Res 8, 749, 2019)”.

-I was a little bit confused about how the technology interacts (or does not interact) with the dura mater. It seems as though it penetrates through the dura mater for the mice, but does not for the marmoset (because the dura was removed). How does this impact the vascular imaging and the injection process?

As described in the Methods section, the mouse dura mater was left intact during the injection process. This is because the dura is thin enough to allow clear observation of the vasculature, and removing it posed a risk of damaging the vessels. However, leaving the dura intact increases the force needed for pipette insertion, which can lead to larger surface dimpling and potential tissue damage (Sharp et al., IEEE Trans. Biomed. Eng. 56, 1, 2009). Additionally, the intact dura can cause the needle to slip if there is a significant discrepancy between the surface normal and the needle insertion angle. This is one of the reasons why we introduced angle adjustments for the needle during insertion. We have described this in the Methods section (lines 1167–1173). For the marmoset experiment, the dura mater was carefully removed because the thicker dura made it difficult to observe fine vasculature patterns and to insert the glass pipette through the dura mater into the cortical tissue.

-I believe that the mice have been sacrificed and histology performed; it looks like the marmoset has not. It would be nice to show some mouse histology in the supplement, to give a sense of the cortical damage and the spread of each injection.

We have added histology data from two mice in which AAV-jGCaMP7c was injected into only one hemisphere and DAPI staining of the cell nuclei was conducted (Response Fig. 1a–e). There were no linear DAPI-negative areas that could be considered to be marks from the glass pipette insertion. Using the DAPI signal, we also counted the cell numbers around the areas with jGCaMP7c fluorescence, and compared the numbers to those in corresponding areas in the contralateral hemisphere that did not undergo multiple injections. The values were not very different between the injected hemisphere and the non-injected hemisphere (Response Fig. 1d). In Response Fig. 1a, the most lateral expression of GCaMP was probably from the most lateral injection site, which was relatively distant from the other injection sites. In the fluorescence profile of this expression in the lateral direction, the half decay distance was roughly estimated

to be 234 μm (Response Fig. 1e). The average half decay distance in the GCaMP expression in the most lateral region, which was relatively far from other expression sites, was $185.3 \pm 16.1 \mu\text{m}$ ($n = 9$ slices from two mice). Considering that the average distance between neighboring injection sites was approximately 0.36 mm (Fig. 2i), the injection site spacing and the spread of GCaMP expression in mice seemed reasonable. In addition, in the marmoset experiment, we estimated the half-decay time of calcium transients in individual neurons that were measured with two-photon imaging (Response Fig. 1f,g). This value was 0.94 ± 0.05 s, which was comparable to that in the original paper that reported GCaMP6s (Chen et al., Nature 499, 295–300, 2013). These results suggest that the multiple injections did not induce apparent cortical damage. We have added these results in the main text (lines 281–287 and 406–409) and Supplementary Figures 7 and 10.

Response Figure 1. Evaluation of tissue structure and function after multiple injections.

a, A section of a mouse brain with multiple injection of AAV-hSyn-jGCaMP7c in only the right hemisphere. The white bounding boxes indicate the areas shown in panels **(b,c)**. Scale bar, 1 mm. **b,c**, Magnified images of the left **(b)** and right **(c)** hemispheres with jGCaMP7c signal (left) and DAPI signal (right). Scale bar, 100 μm . **d**, Quantification of DAPI counts in the AAV-injected hemisphere vs. the control hemisphere. The difference between the two groups was not significant according to a paired t -test (4711.9 ± 33.6 vs. 4671.1 ± 42.7 ; $n = 73$ slices from two mice; $p = 0.89$, two-sided). For each slice, the brightest area of $500 \times 500 \mu\text{m}$ in green, and its corresponding area in the left hemisphere, were chosen for DAPI counts. **e**, The raw fluorescence signal (gray)

and gaussian-smoothed signal (black) of jRCaMP7c along the yellow line shown in (a). The blue horizontal line represents the baseline fluorescence, calculated as the 7th percentile value of the fluorescence in the measured data. The half-decay fluorescence value (horizontal dashed gray line) is the average of the peak value and the baseline. The half-decay distance (red dashed line; 234.22 μm) was measured at the points where the smoothed signal fell below the half-decay fluorescence value. **f**, Four representative traces of calcium transients that were obtained with two-photon imaging of the marmoset motor cortex. The traces were aligned to the peak timing and the amplitude was normalized to the peak amplitude. Scale bar, 1 s. **g**, Histogram of the half-decay time (0.94 ± 0.05 s; $n = 119$ ROIs from four sessions). We calculated the half-decay time of calcium transients with peaks more than four standard deviations of the $\Delta F/F$ traces that had no other peak from 3 s before to 6 s after the peak timepoint and averaged over each imaging session in each of the detected ROIs. To calculate the half-decay time of bursts with fewer than five spikes, the calcium transients with peak values of less than 200% were subjected to analysis (Chen et al., Nature, 499, 295-300, 2013).

-Although I could find the injection volumes for the mouse dye experiment (3nl/site—it would be nice to also say the total volume, though) and the marmoset experiment (30nl/site—same thing), I could not find the number for the mouse virus experiment. Are these particularly large total volumes for these species? Was there any sign of a deleterious immune response? And, is the marmoset still healthy?

For the virus injection in mice, the number of AAV injection sites was 74 and 72 in two male mice with the injections in one hemisphere and 92, 128, 138, and 125 in four mice (three males and one female) with the injections in both hemispheres. The volume of each AAV injection was 30 nl/site; therefore, the total injection volume was 3.1 ± 0.4 μl . This volume was larger than that in our previous study, in which AAV was injected into only the primary motor cortex (0.25–1 μL ; Masamizu, et al., Nat. Neurosci., 17, 987–994, 2014). However, if the injected two-dimensional area was considered (the cortical depth was not considered), the injected solution density in the current mouse experiments was approximately 60.5 nl/mm² (assuming the craniotomy area was 4×8 mm in one hemisphere and 8×8 mm in both hemispheres), and this value was less than that in a previous report (79.6–318.5 nl/mm²; the craniotomy was a circle of radius 1 mm). For the marmoset experiment, the total volume of virus solution was 7.98 μL . This was also comparable to that in our previous study using marmosets (15 μL ; Ebina et al., PNAS 116, 22844–22850, 2019). In addition, the injected solution density was estimated to be 62.3 nL/mm² (7.98 $\mu\text{L}/[8 \times 16$ mm]), which was lower than the 333.3 nL/mm² in Ebina et al. (2019) (15 $\mu\text{L}/[5 \times 9$ mm]). Thus, overall, the injection volume used in the current system was not much larger than those in our previous studies that used mice and marmosets. We have added these results to the Methods section (lines 1182–1192 and 1210–1214).

Upon observation, the marmoset exhibited no signs of deleterious immune responses, such as significant body weight loss, reduced eating, or severe fever. Throughout the

experiment, the marmoset's body weight remained between 85%–95% of its pre-surgery weight of 339 g, which is considered normal for animals under water restriction for behavioral experiments. The experiment was completed 2 months after surgery, during which the marmoset displayed normal behaviors with no abnormal response, such as epilepsy. Overall, the marmoset was in apparent good health and fit to participate in behavioral experiments. We have added a description of the health of the marmoset after describing the surgery (lines 1221–1226).

-It would be helpful to add a bit to the Discussion about potential applications to rats and macaques.

We have added the following discussion at lines 484–491:

“ARViS could also be applied to the neocortex of other mammals, including rats and macaques. The area of the whole dorsal cortex in the rat is approximately 12×12 mm, which is close to the current craniotomy area in the marmoset (8×16 mm). Thus, ARViS may be applied to the whole rat dorsal cortex. By contrast, the macaque cortex is much larger than the marmoset cortex, and it would therefore be difficult to use the present ARViS to perform cortex-wide injection. However, ARViS would be more efficient than manual injections, even if the injected area is limited to a part of the neocortex (probably one functional cortical area) with an area that is comparable to the area imaged in the current study”.

-I visited the github page, and was so pleased to find much more than was promised! From the perspective of a potential user, I recommend explaining in the manuscript that you have a wiki page with a full parts list, assembly guide, and workflow instructions, along with the code.

Thank you for appreciating our efforts to distribute this technology. We have added this information to the main text (lines 242 and 243).

Reviewer #1 (Remarks on code availability):

I looked at what was available, but I did not actually open the code.

We appreciate that you accessed the site and our code.

Reviewer #2 (Remarks to the Author):

COMMENTS FOR AUTHORS

Nomura et al. developed an Automated Robotic Virus injection System (ARViS) which can express genetically encoded fluorescent sensors in multiple brain areas. ARViS consists of two key technologies: image recognition of vasculature structures on the cortical surface to determine multiple injection sites, and robotic control of micropipette insertion perpendicular to the cortical surface with 50- μm precision. ARViS successfully achieved 266-site injections over the frontoparietal cortex of a common marmoset, and the authors demonstrated in vivo calcium imaging in the marmoset frontoparietal cortex. As also shown in the study by Jendritzka et. al. (Nature Comm, 2023, 14:577), this is one of the important steps for the understanding of the neural networks in the marmoset cortex. However, there are several limitations to consider for the publication in Nature Communications.

Thank you for recognizing the importance of our work in regard to exploring coordination among multiple brain regions. We appreciate the reviewer's acknowledgment of the significance of our Automated Robotic Virus Injection System (ARViS) for advancing the understanding of neural networks in the marmoset cortex.

Firstly, while ARViS identifies the injection sites with minimal bleeding through image recognition of vasculature structures, multiple injections may have negative impacts not only on blood vessels but also on various cells including neurons and the extracellular matrix (ECM) within the cortical tissues. The authors should validate through histological studies to ensure that multiple injections by ARViS do not compromise the normal structure and function of cortical tissues. Also, it is crucial to demonstrate the optimal density and number of injection sites that allow cortex imaging with minimized negative issues. Overall, the authors need to show the significant advantages of ARViS over the current method that manually injecting into a few regions.

Thank you for raising an important concern about tissue damage resulting from the use of our method. First, we would like to point out that the glass capillary with a tip diameter of 30 μm was carefully manufactured by sharpening it for more than 30 minutes. This minimizes physical damage to brain tissue during insertion (Sharp et al., IEEE Trans. Biomed. Eng. 56, 1, 2009). We have added histology data from two mice in which AAV-jGCaMP7c was injected into only one hemisphere and DAPI staining of the cell nuclei was conducted (Response Fig. 2a–d).

There were no linear DAPI-negative areas that could be considered to be marks from the glass pipette insertion. Using the DAPI signal, we also counted the cell numbers around the areas with jGCaMP7c fluorescence, and compared the numbers to those in corresponding areas in the contralateral hemisphere that did not undergo multiple injections. The values were not very different between the injected hemisphere and the non-injected hemisphere (Response Fig. 2d). In addition, for the marmoset experiment, we estimated the half-decay time of the calcium transients of individual neurons that were measured with two-photon imaging (Response Fig. 2e,f). This time was 0.94 ± 0.05 s, which is comparable to that in the original paper that reported GCaMP6s (Chen et al., Nature 499, 295–300, 2013). These results suggest that the multiple injections did not induce apparent cortical damage. We have added these results to the main text (lines 281–283, 406–409, and 1178–1180) and Supplementary Figures 7 and 10.

Response Figure 2. Evaluation of tissue structure and function after multiple injections.

a, A section of a mouse brain with multiple injection of AAV-hSyn-jGCaMP7c in only the right hemisphere. The white bounding boxes indicate the areas shown in panels **(b,c)**. Scale bar, 1 mm. **b,c**, Magnified images of the left **(b)** and right **(c)** hemispheres with jGCaMP7c signal (left) and DAPI signal (right). Scale bar, 100 μ m. **d**, Quantification of DAPI counts in the AAV-injected hemisphere vs. the control hemisphere. The difference between the two groups was not significant according to a paired *t*-test (4711.9 ± 33.6 vs. 4671.1 ± 42.7 ; $n = 73$ slices from two mice; $p = 0.89$, two-sided). For each slice, the brightest area of 500×500 μ m in green and its corresponding

area in the left hemisphere were chosen for DAPI counts. **e**, Four representative traces of calcium transients that were obtained with two-photon imaging of the marmoset motor cortex. The traces were aligned to the peak timing and the amplitude was normalized to the peak amplitude. Scale bar, 1 s. **f**, Histogram of the half-decay time (0.94 ± 0.05 s; $n = 119$ ROIs from four sessions). We calculated the half-decay time of calcium transients with peaks more than four standard deviations of the $\Delta F/F$ traces that had no other peak from 3 s before to 6 s after the peak timepoint and averaged over each imaging session in each of all the detected ROIs. To calculate the half-decay time of bursts with fewer than five spikes, the calcium transients with peak values of less than 200% were subjected to analysis (Chen et al., *Nature*, 499, 295-300, 2013).

We agree that users should optimize the density and number of injection sites to obtain a balance between minimizing cortical damage and the efficacy of gene expression. However, this balance needs to be optimized for each virus type individually, and users should consider this optimization according to their specific objectives and methods, which is beyond the scope of the current study. We have stated that the injection conditions of ARViS need to be optimized to individual experiments in the Limitations subsection of the Discussion section (lines 511–514).

What we emphasize in the current study is that the advantage of ARViS is its ability to realize spatially-selective gene expression independent of endogenous expression patterns and anatomical boundaries in both rodent and marmoset cerebral cortices. For example, some of the biggest neuroscience questions are which areas of the neocortex, and how much neuronal and glial cell activity and molecular activity in the neocortex, are required for sleep regulation (Kim et al., *Nature* 612, 512–518, 2022; Bojarskaite et al., *Nat. Commun.* 11, 3240, 2020; Tsunematsu et al., *J. Neurosci.* 41, 5440–5452, 2021). Both subcortical areas and the cerebral cortex regulate sleep (Krone et al., *Nat. Neurosci.* 24, 1210–1215, 2021). ARViS can label many cerebral cortical neurons, but not subcortical neurons, which is very difficult to perform when AAV.PHP is used. Systematically varying the transgenic area is easy with ARViS, but manual injection requires a great deal of labor and skill. ARViS will also be useful for expressing optogenetic or chemogenetic probes, and RNAi or constitutively active forms of related genes, in any cell type and in any mouse cerebral cortical region. ARViS promises a cross-species approach to many questions related to a variety of functions of the cerebral cortex, questions that have been difficult to solve with previous methods alone. Thus, we believe that the current study will contribute to the advancement of neuroscience by presenting details of how ARViS is constructed and by showing that it works in both the mouse and marmoset cerebral cortex. We have added a description of future possible applications in the Discussion section (lines 475–483).

Secondly, there is a limitation regarding the depth of injection. How deep into the brain region can ARViS facilitate injections? Imaging only at the cortex level restricts the scope of future

research for understanding brain functions. Thus, it would be highly significant if the authors could demonstrate that ARViS can inject viruses into multiple brain regions with varying depths, enabling research across different layers of the neural network of the brain.

Thank you for suggesting these limitations and future directions of our system. Currently, our system focuses solely on the preservation of cerebral cortical vasculature. Therefore, we have modified our title to “ARViS: A bleed-free multi-site automated injection robot for accurate, fast, and dense delivery of viruses to the mouse and marmoset **cerebral cortex**”, to avoid any misleading statements or overstatements. However, we believe that this technology will serve as an important basis for innovative future research. For example, in vivo whole-cell patch clamping and patch-sequencing are critical methods that link accumulated electrophysiological knowledge, gene expression profiles, and animal behavior, although they are renowned for being low-throughput, time-consuming, and requiring years to master (Marx, Nat. Methods 19, 1340–1344, 2022). We believe that in combination with automation of in-vivo patch clamping technology (Kodandaramaiah et al., Nat. Methods 9, 585–587, 2012; Anecchino et al., Neuron 95, 1048–1055, 2017), our system could provide a significant step towards large-scale automation.

Lastly, in addition to calcium imaging, it is important to demonstrate the imaging of neurotransmitter receptor sensors, such as dopamine sensors and glutamate sensors.

Thank you for suggesting the demonstration of neurotransmitter imaging other than calcium imaging. First, our paper is a methods paper, and its main contribution is the implementation of a safe experimental procedure that combines image recognition with precise robotic operation, reducing dependency on the experimenter’s skill and concentration. We provide a complete parts list, assembly instructions, and all necessary codes. In addition, we demonstrate that one-photon and two-photon imaging of GCaMP6s expression by ARViS can detect motor representation in the marmoset frontoparietal cortex at both cortical-area and single-neuron levels. While demonstrating the application with different neurotransmitter sensors is important, as the reviewer suggests, it is evident that our method can be applied to various sensors. Therefore, we believe that our paper significantly contributes to the neuroscience community, even without specific examples of different neurotransmitter sensors. However, as Reviewer #1 pointed out, ARViS is a tool for the injection of viruses for many purposes, including optogenetics and chemogenetics. Thus, we have weakened the statement on neurotransmitter sensors and re-written the second and third sentences in the Introduction as follows: “Furthering their use, the recent development of other genetically encoded biosensors and

optogenetic and chemogenetic tools has opened new opportunities to study the spatiotemporal dynamics of neuronal and glial activity and synaptic transmission (Marvin et al., *Nat Methods* 15, 936–939, 2018; Sun et al., *Cell* 174, 481–496, 2018; Sternson & Roth, *Ann. Rev. Neurosci.* 37, 387–407, 2014; Kim et al., *Nat. Rev. Neurosci.* 18, 222–235, 2017). If these probes could be expressed uniformly across multiple broad cortical areas in non-human primates (NHP), it would be possible to measure and manipulate the spatiotemporal dynamics of the inter-areal spreading of a variety of activities occurring in motor, cognitive, and sensory processing, activities that are well developed in primates (Galvan et al., *J. Neural. Transm.* 125, 543–563, 2018; Matsuzaki & Ebina, *Curr. Opin. Neurobiol.* 64, 103–110, 2020; Kalaska, *F1000Res* 8, 749, 2019”).

Reviewer #3 (Remarks to the Author):

In this study, the authors aimed to develop an Automated Robotic Virus injection System (ARViS) for broad expression of a biosensor, that integrates two technologies: one for the image-based identification of vascular structures on the cortical surface to select multiple injection locations without impacting them, and another for the robotic management of micropipette insertion at a 90-degree angle to the cortical surface with precision up to 50 micrometers. In the mouse cortex, ARViS methodically administered virus solution injections at 100 sites over a period of 100 minutes, maintaining a low bleeding risk of only 0.1% per site. Additionally, ARViS efficiently completed injections at 266 sites across the frontoparietal cortex of a common marmoset. Both one-photon and two-photon calcium imaging in the marmoset's frontoparietal cortex were also demonstrated.

This study contains an analysis of good clinical significance but there are several factors concerning the methodology.

Thank you for recognizing the significance of our results. We believe that our efforts represent an important step towards understanding the multifaceted functions of the neocortex in non-human primates.

Please find some of my comments below:

1. The authors initially note a discrepancy in gene expression levels between transgenic animals and those subjected to high-titer AAV-mediated gene delivery. Could you please provide further insight into the reasons behind this observed difference? Considering that transvenous transfection of AAV may result in diminished bioavailability at critical sites, particularly within the brain, due to potential locoregional uptake and metabolism, as well as non-target organ degradation.

In this study, when we refer to “high-titer AAV-mediated gene delivery”, we mean only direct injection of high-titer AAV into the brain, not transvenous transfection of AAV. However, our wording was confusing, so we have changed it to “direct injection of high-titer AAV into the brain”. In comparison with transgenic animal generation, direct virus injection into the brain tissue achieves a larger copy number of transgene in target neurons. This is because transgenic animal generation requires a process by which the transgene is incorporated into the host genome. In addition, inhibitory positional effects and repeat-induced gene silencing frequently occur in the transgene in the host genome (Zeng and Madisen, *Prog. Brain Res.* 196, 193–213, 2012). We have added this description to the Introduction section (lines 56–58).

2. In deep learning applications for vessel segmentation, the acquisition of finely-focused surface images were accomplished by automatically adjusting the distance between Camera S and the cortical surface, capturing data at 62 points across the neocortex. Nonetheless, the potential for minor movements in the anesthetized animal model due to respiration could introduce errors. It is unclear how these types of inaccuracies could be mitigated.

We agree with the notion that movements controlled by the autonomic nervous system could affect the cortical surface measurement. We have re-analyzed our marmoset data obtained during measurement of the cortical surface by the laser distance sensor to estimate errors in inferring the positions of the cortical surface points.

We ensured that the animals' skulls were tightly fixed to the surgery apparatus, so we presume that cortical surface movement was primarily caused by intracranial pressure variations. Previous studies indicate that intracranial pressure depends on blood pressure and cerebrospinal fluid pressure (Czosnyka & Pickard, *J. Neurol. Neurosurg. Psychiatry* 75, 813–821, 2004), with respiratory movement having only a minor effect (Cardoso, et al., *J. Neurosurgery* 59, 817–821, 1983). Since cerebrospinal fluid pressure fluctuates with the heartbeat (Czosnyka and Pickard, 2004), the heartbeat is thought to be the major origin of cortical tissue movement in head-fixed anesthetized animals.

During the measurement of the cortical surface, the laser distance sensor moved in a zigzag pattern in the YZ plane (Response Fig. 3a). We reanalyzed the marmoset time-series data acquired during this measurement to isolate components of putative animal movements. The sensor moved in two phases: 0.1-mm steps in the Y direction and 8-mm steps in the Z direction (Response Fig. 3a,b). Closer observation of the first phase with slow-speed sensor movement revealed depth signal fluctuations of up to 0.1 mm, which did not depend on the sensor's movement timing in the Y direction (Response Fig. 3c). We speculate that these fluctuations were induced by animal movements. We selected one continuous period of time when the sensor position was in the most medial and lateral sides without sensor movement in the Z direction (red and blue lines in Response Fig. 3a,b, respectively) as a segment for further analysis to isolate these fluctuations. We constructed a linear regression model to fit the XYZ positions of the laser distance sensor to the measured depth of the cortical surface in 36 segments (in the medial and lateral sides) and extracted the residual (Response Fig. 3d). This residual showed oscillation, so we suspected that the residual reflected the oscillatory heartbeat. In fact, the peak-to-peak intervals were 0.398 ± 0.0125 s (2.51 ± 0.07 Hz, $n = 18$ segments) in the medial side and 0.396 ± 0.006 s (2.53 ± 0.04 Hz, $n = 18$ segments) in the lateral side, matching our recording of the heartbeat rate during surgery (2.42 ± 0.14 Hz, $n = 6$ recorded

timepoints), and the reported heartbeat rate of anesthetized marmosets (3.13 ± 0.10 Hz; Moussavi, et al., *Sci. Rep.*, 10, 10221, 2020) (Response Fig. 3e). The amplitude of the oscillation was 30.6 ± 3.6 μm ($n = 18$ segments) in the medial side and 46.2 ± 1.2 μm ($n = 18$ segments) in the lateral side (Response Fig. 3f). Thus, the oscillation amplitude in the Z direction was larger in the lateral side than in the medial side.

Assuming that brain pulsation caused by heartbeat is mostly perpendicular to the cortical surface (Greinz et al., *Neuroradiology* 34, 370–380, 1992), using vector decomposition, the maximal heartbeat-induced displacement perpendicular to the cortical surface was estimated to be 29.9 ± 3.5 μm ($n = 18$ segments) in the medial side and 39.9 ± 1.1 μm ($n = 18$ segments) in the lateral side (Response Fig. 3g,h). These values are only $5.98\% \pm 0.70\%$ and $7.98\% \pm 0.22\%$ of the planned depth (500 μm), respectively. If this heartbeat-induced displacement perpendicular to the cortical surface also occurred while picturing the cortical surface with Camera S (Response Fig. 3i), the possible maximal displacement of the reference position in a plane perpendicular to the optical axis of Camera S was estimated to be 1.6 ± 0.2 μm ($n = 18$ segments) in the medial side and 13.0 ± 0.4 μm ($n = 18$ segments) in the lateral side (Response Fig. 3j). Given that the distance from the planned injection site to the nearest blood vessels was set at more than 65 μm , this possible displacement would not have a substantial impact on the injection safety. Thus, we consider that heartbeat-induced misalignment of the injection point during the cortical surface measurement and AAV injection is not negligible, but would have only a minor effect. We have added these results in the main text (lines 313–348) and Supplementary Figure 9.

However, the effect of pulsation on the estimation of the injection site may be more critical in larger non-human primates such as rhesus monkeys because the heartbeat would cause larger movement of the brain. Real-time monitoring of brain phasic movement, allowing data recording at specific phases and compensational movement during pipette injection, might be necessary to decrease the rate of vascular injury due to injection. This suggestion of real-time monitoring of the cortical surface during the injection process has been added to the revised version of the manuscript in response to the following comment.

Response Figure 3. Effect of pulsation of the marmoset brain on the estimation of the positions of the cortical surface.

a, Schematic representation of the method used to measure the surface of the marmoset brain. Top, a part of the trajectory of YZ movement of the laser distance sensor (magenta) viewed from the X (depth) axis. The depth fluctuation due to the animal movement was estimated in the period without laser distance sensor movement along the Z axis (red line on the medial side and blue line on the lateral side). Bottom, laser distance sensor movement viewed from the Y (anterior-to-posterior) axis. Camera S (blue) is positioned at approximately 16 degrees tilt relative to the optical axis of the laser distance sensor (yellow; equal to X axis). From the laser distance sensor signal, the depths of individual points on the cortical surface in the X direction were measured. The marmoset was tilted at an angle of approximately 13 degrees to the Z axis so that the cortical surface within the craniotomy was as close to perpendicular to the optical axis of camera S as possible. **b**, Top, representative time course of the laser distance sensor signal, shown along with the corresponding time courses of positions of the laser distance sensor in the X (second top), Y (third top), and Z (bottom) axes. Periods with no Z direction movement are shown in red (medial side) and blue (lateral side). The time segments shown in red and blue were used to isolate these

fluctuations because, in the other periods when the sensor moved fast in the Z direction, depth fluctuations caused by animal movements were masked by depth changes of the cortical surface scanning in the Z-axis direction (ranging from -4 to 4 mm), which were nearly a hundred times larger. **c**, Example laser distance sensor signal (black line) shown in the cyan box in **(b)**, and corresponding linear regression curve explained by the position of the laser distance sensor (orange line). **d**, Smoothed residual signal of the linear regression curve subtracted from the raw data (black line). The peaks and troughs are indicated by black circles. **e**, Distribution of peak-to-peak interval (red, medial side; blue, lateral side). **f**, Distribution of the amplitude of waves (red, medial side; blue, lateral side). **g**, Schematic illustrating calculation of the deviation perpendicular to the cortical surface (black arrow) from the deviation in the X direction measured by the laser distance sensor (magenta arrow). The black curve represents the brain surface, while the gray one shows the deviated surface with pulsation. Assuming the brain surface curvature is negligible, the perpendicular pulsation component can be approximated by multiplying the laser-measured vector by the cosine of the angle between the perpendicular vector of the surface and the laser trajectory (θ), which was 12.9 and 30 degrees in the medial and lateral sides, respectively. **h**, Distribution of deviation perpendicular to the cortical surface (red, medial side; blue, lateral side). **i**, Schematic illustrating estimation of the deviation in a plane parallel to the cortical surface in the Camera S measurement (black arrow). **j**, Distribution of the estimated deviation in Camera S measurement; red, medial side; blue, lateral side.

3. Similarly, while the approach described for avoiding blood vessels by meticulously mapping the three-dimensional (3D) structure of the cortical surface and utilizing deep learning for image recognition of vascular structures, complemented by clustering-based optimization for selecting multiple injection sites, is innovative, it does not fully address the inherent complexities of blood vessel avoidance. This technique, although advanced, may not account for the dynamic nature of the vasculature, including variations in vessel depth and the potential for movement or changes over time. Additionally, the reliance on clustering-based optimization does not guarantee the elimination of risk to all blood vessels, particularly in highly vascularized areas where the density and proximity of vessels may pose significant challenges. Further refinement and validation of this method are necessary to ensure its effectiveness and safety in practical applications.

As Reviewer #3 pointed out, we did not explicitly treat the variation of the vessels under the cortical surface. However, two-photon microscopic observation revealed that nearly 70% of major (>5 μm) vessels deviated less than 49 μm from their surface origin up to a depth of 0.5 mm (Kozai et al., J. Neural. Eng. 7, 046011, 2010). Although it is possible that the pipette penetrated intracortical vessels, rather than vessels in the cortical surface, in such a case, blood would be expected to appear on the cortical surface through the pipette insertion area. Thus, at least in our experimental condition, our clustering-based optimization for selecting multiple injection sites worked effectively. However, only blood vessels with a width of > 30 μm were considered to be avoided. Thus, our method does not guarantee the elimination of risk to all blood vessels. We have described this limitation (lines 507–511).

We cannot rule out the possibility that blood vessel patterns changed to some extent between the time of the cortical surface measurement and the time of the pipette injection. Nevertheless, the bleeding probability was very low. During the AAV injection period, the vessel pattern on the cortical surface can be monitored. We confirmed that if the planned injection site was found to correspond to a vessel, the injection site was slightly moved to avoid it. We have exactly described this optional method, “realtime closed-loop evaluation of vessels”, in the revised manuscript (lines 290–293 and 1083–1102) and to GitHub.

We agree that our approach avoiding setting injection sites within 65 μm of blood vessels for safety prevents injections into highly vascularized areas. However, if there is a small safety region within a highly vascularized area, an injection site can be placed there, and such areas can be covered to some extent. For instance, in the example shown in Response Figure 4, the total area of spot-like safety regions of 30 pixels or less (cyan; a pixel size of 7.5 μm) was 753 pixels. Although these small regions occupied only 0.92% of the entire area of the safety regions (8.17×10^4 pixels), 3.94% of injection sites (five out of 127 sites) were set in these small regions (arrows). When ARViS is used for expression of transgenes for fluorescence imaging and optogenetics, the light exposure on highly vascularized areas is not effective, meaning that injection into these areas is not practically important. We have added this example and interpretation in the main text (lines 183–186) and Supplementary Figure 3.

Response Figure 4. Example vessel pattern with safety regions and injection sites.

An example injection site arrangement. The injection sites (red circles) were set in the safety regions and the number of the sites were 127. The safety regions are divided into those with individual areas of more than 30 pixels (black; a pixel size of 7.5 μm) and those with individual areas of 30 pixels or less (cyan). Gray shading illustrates the vessel patterns. White pixels are

those which were less than 65 μm from the nearest blood vessel. Red arrows indicate the injection sites arranged in the safety regions with area of 30 pixels or less. Scale bar, 1 mm.

4. The authors describe their initial training of the convolutional neural network (CNN) model using only twenty images of the retinal fundus, which exhibited limited accuracy, particularly for thick caliber vessels. They then supplemented this dataset with six cortical images of the brain. Notably, cortical vessels lack an internal elastic lamina crucial for cerebral autoregulation, unlike retinal vessels, which could compromise the internal validity of their training dataset. Why was it not possible to exclusively utilize mouse cortex vasculature images for training the dataset?

We evaluated the segmentation performance of SA-UNets that were trained with four different datasets: the DRIVE dataset only (DRIVE model), DRIVE dataset and six-mouse dataset (initial model), 33-mouse dataset (non-DRIVE model), and DRIVE dataset and 33-mouse dataset (final model). To compare the segmentation performance of these four models, we used three images that were common to all four models as the test dataset (Response Fig. 5a,b). Then, we compared six metrics of model performance (Response Fig. 5c). Among them, we consider accuracy, AUC, F1, and MCC to be good indicators for evaluating model performance. The final model showed the best accuracy, F1, and MCC, and the second best AUC. However, the AUC value of the final model was almost the same as the highest AUC value, which was achieved by the non-DRIVE model. Thus, we consider that the final model was the best of the four models, and used the final model in the subsequent mouse and marmoset experiments. The non-DRIVE model showed the second best AUC, F1, and MCC, with values that were not largely different from those of the final model. In addition, the performance of the DRIVE model was much lower than that of the initial model. Thus, for training SA-UNet for the purpose of vessel segmentation on the mouse cortical surface, the contribution of the DRIVE dataset was much smaller than that of the mouse data set. We have added this result in the main text (lines 134–145) and Supplementary Figure 2i–k.

Response Figure 5. Segmentation performance of SA-UNet with four different training data sets.

a, Three input images of the test dataset. Scale bar, 1 mm. **b**, Manually labeled images of the test dataset. Scale bar, 1 mm. **c**, Six metrics applied to the models trained on different training datasets. Accuracy (the proportion of correctly classified instances): 0.881 for the final model, 0.876 for the non-DRIVE model, 0.860 for the initial model, and 0.792 for the DRIVE model. AUC (Area Under the ROC Curve, a measure of the model's performance across all classification thresholds): 0.941 for the final model, 0.957 for the non-DRIVE model, 0.888 for the initial model, and 0.724 for the DRIVE model. F1 Score (the harmonic mean of precision and recall, balancing both metrics): 0.774 for the final model, 0.767 for the non-DRIVE model, 0.718 for the initial model, and 0.511 for the DRIVE model. MCC (Matthews Correlation Coefficient; a robust metric for evaluating binary classifications, especially in imbalanced datasets, ranging from 1 for perfect prediction to -1 for complete disagreement): 0.706 for the final model, 0.696 for the non-DRIVE model, 0.628 for the initial model, and 0.382 for the DRIVE model. Sensitivity (recall, the proportion of actual positive instances correctly identified): 0.884 for the final model, 0.885 for the non-DRIVE model, 0.773 for the initial model, and 0.472 for the DRIVE model. Specificity (the proportion of actual negative instances correctly identified): 0.881 for the final model, 0.873 for the non-DRIVE model, 0.886 for the initial model, and 0.888 for the DRIVE model. With sensitivity and specificity, if the number of positive and negative examples is biased in one direction or the other, the poorer performing model will have a higher rate of biased positive or negative determinations, and may produce higher values than the better performing model. In fact, in the current case of classification training with a high percentage of negative examples relative to the total sample, specificity was higher with the initial and DRIVE models, which would not be able to correctly determine positive examples. Thus, sensitivity and specificity were not considered in our evaluation of segmentation performance.

5. The author is requested to provide a more detailed explanation regarding the technique of multiple injections into the mouse dorsal cortex, particularly emphasizing the insertion and withdrawal process. It would be beneficial to illustrate this procedure clearly in a separate figure. Also, the timing of insertion and pullback should be specified to ensure the experimental subject's

safety.

We have added Supplementary Figure 6 (Response Fig. 6, below) to illustrate the insertion and withdrawal process. This shows all processes from the initial position of the pipette before the first injection to the position just before preparation for injection in the second injection. The timing of insertion and pullback are also shown. We hope this figure satisfies Reviewer #3. We have also rewritten this method in more details (lines 1058–1081).

Response Figure 6. Sequential injections into the target injection sites in mice.

The series of pipette tip movements were made according to the following steps. **a**, First, the pipette tip was set 3 mm above the bregma. **b**, It moved to 1 mm above the first target injection point and it then vertically approached 0.2 mm above the target injection point on the cortical surface at a speed of 20 mm/s. **c**, The angle of the pipette was changed to be perpendicular to the surface. **d**, A new coordinate system with the injection point on the cortical surface as the origin was set as frame {I}. The X axis of the device was aligned with the vertical line to the

cortical surface. **e**, The pipette tip was pulled 2 mm back along the X axis in frame {I}. **f**, It was inserted to depths of 0.6–0.8 mm from the cortical surface at the injection point. **g**, It was pulled back to the target depth (0.3 mm) at 0.1 mm/s. **h**, The pipette was held for a time setting of 15–30 s before the injection and then the solution was injected at a speed of 2 nl/s for 15 s. Then, the pipette was hold for 30 s. **i**, The pipette was pulled back 0.2 mm above the surface along X axis in frame {I} at the speed of 0.1 mm/s. **j**, The pipette was pulled back 1 mm along X axis in frame {I} at the speed of 20 mm/s. **k**, The angle was aligned to the initial state. **l**, The pipette was moved to 4 mm above the injection point.

6. While the authors suggest that the limitation of ARViS regarding the minimal volume capacity of each pipette (~4 μ L) and the lengthy calibration time (90 minutes) for glass pipette positioning can be mitigated by employing a syringe with a larger capacity, this solution appears overly simplistic and does not adequately address the complexity of the issues at hand. The suggestion fails to consider the nuanced requirements of precise delivery mechanisms needed for effective and safe injections into the cortical tissue, which involve more than just the volume capacity of the injector. For instance, the use of a larger capacity syringe does not inherently solve the challenge of accurately calibrating the injector to ensure precise delivery at the desired location, nor does it address potential alterations in injection pressure that could affect tissue integrity. Additionally, the proposal overlooks the need for a comprehensive solution that integrates improvements in both the hardware for delivery and the software for control and monitoring, ensuring that the increased volume does not compromise the precision and safety of the injections. Therefore, a more detailed and holistic strategy is required to effectively overcome these limitations.

We appreciate Reviewer #3's feedback on our limitations section. We agree with the reviewer's comment that the use of a larger capacity syringe as a solution for volume capacity limitation is over simplified. We have described possible problems that can occur when a larger capacity syringe is used (lines 500–502). We have also added text (lines 502–504 and 511–514) stating that it is necessary to achieve wide and uniform expression levels that are sufficient for practical experiments by optimizing the virus titer, the number of injection sites, injection time, and injection pressure, and/or applying clearance of the debris attached to the pipette tip with detergent (such as that used for repetitive patch-clamp recordings) (Kolb et al., Sci. Rep. 6, 35001, 2016) and re-filling with the virus solution.

Reviewer #4 (Remarks to the Author):

The authors present a technical setup and computational controlling workflow for automatic injection of viruses in the marmoset cortex, with the aim of automatizing cortex-wide GECI measurements with approximately uniform distribution of injection sites. The presented system aims to find optimal sites so that bleeding is prevented and micropipette injection is exact and perpendicular. To this end, the injector with micropipette are mounted on a robot stage, which measures the brain surface in the fixated head with skull removed using combined laser distance sensing and cameras.

The proposed methodology is highly significant for measuring spatiotemporal dynamics in the cortex. It provides significant potential to advance the development of detailed atlases of functional organization in animal brains, and improve our understanding of structure/function relationships. It extends existing work by increasing automation levels in the technical setup, and introduces AI algorithms for performing vessel localization in camera images.

The focus of this review is on the image processing part of the system, and the calibration of the pipette tips. I have little expertise regarding virus injection and the protocols and requirements for in-vivo animal experiments.

The paper is well written and organized. The Figures are appropriately chosen and described to support the text effectively. The drawn conclusions are well supported by the work. While the experiments and results are generally appropriate, I suggest some minor revisions before publication.

Thank you for recognizing the significance of our results and understanding our efforts to achieve precision.

A shortcoming of the approach is the significant recalibration effort of about 90 minutes after changing a pipette. This aspect is clearly described in the paper, and understandable given the requirements in precision. I was a bit surprised however about the 17 minutes required to finish a particular measurement (l. 167). Is this due to the scanning or processing? I suggest to break down this time estimate into the different measurement and processing parts. E.g. how much of the time is taken for the vessel segmentation? Could that be reduced by using a (better?) GPU?

As we describe in Supplementary Table 1, it took approximately 5 minutes to scan the cortical

surface, 1 minute to perform the segmentation, 1 minute to review and manually define the injection area, and 5 minutes to calculate an appropriate arrangement of the injection sites. In the original work for the manuscript, we had already made efforts to speed up the code by utilizing the Parallel Computing Toolbox in MATLAB during the calculation of the injection sites, and a GPU (NVIDIA GeForce RTX 2070 SUPER) for the vessel segmentation. We have added descriptions of these steps to the Methods section (lines 992, 993, 1030, and 1031).

For identifying the vessel structures, an SA-UNet is used which is trained on an existing retinal vessel dataset. The model choice is not much elaborated, but appears reasonable to me, and while not the single obvious choice it can be considered state of the art for such type of application. Unfortunately, the paper does not provide the necessary information to reproduce the model training. Essential parameters such as batch size, possible cropping of the training images, learning rate, etc. should be specified. Furthermore, while the text refers to “the modified SA-UNet” (l. 873), I could not identify which modifications exactly were made to the originally proposed network. This should be described more clearly.

We apologize for omitting this information necessary for reproducibility. The parameters for model training were as follows: the captured image was cropped to 512×512 pixels, the number of filters for the first convolution layer was 16, total epochs was 150, learning rate was 10^{-3} for the first 100 epochs, decaying to 10^{-4} for the last 50 epochs, the probability for the DropBlock layer was kept at 0.82, its block size was seven, and the batch size for training was four. We have added this information to the Methods section (lines 993–997).

As the reviewer pointed out, we have not modified the model architecture itself. We apologize for the confusion. After we have explained that the SA-UNet model was trained on all the training dataset (defined as final model), we have now stated “Thus, we considered that the final model was the best of the four models, and used this final model for all subsequent experiments” (lines 144 and 145). In the revised manuscript, we have used “SA-UNet” instead of “modified SA-UNet”.

Regarding the performance of the model, I am missing a more detailed quantitative evaluation. While some scores are provided (l. 873), I could not capture from the text whether the segmentation was evaluated on a test set and whether any cross-validation was performed, and which images exactly went into the evaluation. The authors report that the pre-trained net didn't work well on small vessels, and improved after adding 6 newly annotated images for re-training. Here I am missing a quantification of the initial performance and the improvement after retraining (e.g. l. 130 quantifies vessel thickness in the final model, but corresponding numbers of the initial

model are lacking). It should also be described why 6 images were added to the training set, not less or more (also in line 865, does “several images” refer to those 6?). Did you see a saturation point in the performance, or is the annotation effort too high to try out more? It would also be helpful that the paper discusses briefly whether new annotations are likely needed when measuring a new subject, or setting up a new installation of the apparatus with e.g. different lighting conditions.

We apologize for the insufficient explanation of the segmentation evaluation with the SA-UNet models. We trained SA-UNet on four different training datasets: the DRIVE dataset only (DRIVE model), DRIVE dataset and six-mouse dataset (initial model), 33-mouse dataset (non-DRIVE model), and DRIVE dataset and 33-mouse dataset (final model). For each model, we used 95% of the training set to train the SA-UNet and used the remaining 5% as the test set. We performed 20-fold cross validation to optimize the SA-UNet. To compare the segmentation performance of these four models, we used three images that were common to all four models as the test dataset (Response Fig. 7a,b). Then, we compared six metrics for the model performance (Response Fig. 7c). Among them, we consider accuracy, AUC, F1, and MCC to be good indicators of the model performance. The final model showed the best accuracy, F1, and MCC, and the second best AUC. However, the AUC value of the final model was almost the same as the highest AUC value obtained with the non-DRIVE model. Thus, we considered that the final model was the best of the four models and used this final model in the subsequent mouse and marmoset experiments. The non-DRIVE model showed the second best AUC, F1, and MCC, with values that were not largely different from those of the final model. In addition, the performance of the DRIVE model was much lower than that of the initial model. Thus, for training the SA-UNet for vessel segmentation on the mouse cortical surface, the contribution of the DRIVE dataset was much smaller than that of the mouse data set. We have added this result in the main text (lines 134–145) and Supplementary Figure 2i–k.

Response Figure 7. Segmentation performance of SA-UNet with four different training data sets.

a, Three input images of the test dataset. Scale bar, 1 mm. **b**, Manually labeled images of the test dataset. Scale bar, 1 mm. **c**, Six metrics applied to the models trained on different training datasets. Accuracy (the proportion of correctly classified instances): 0.881 for the final model, 0.876 for the non-DRIVE model, 0.860 for the initial model, and 0.792 for the DRIVE model. AUC (Area Under the ROC Curve, a measure of the model's performance across all classification thresholds): 0.941 for the final model, 0.957 for the non-DRIVE model, 0.888 for the initial model, and 0.724 for the DRIVE model. F1 Score (the harmonic mean of precision and recall, balancing both metrics): 0.774 for the final model, 0.767 for the non-DRIVE model, 0.718 for the initial model, and 0.511 for the DRIVE model. MCC (Matthews Correlation Coefficient; a robust metric for evaluating binary classifications, especially in imbalanced datasets, ranging from 1 for perfect prediction to -1 for complete disagreement): 0.706 for the final model, 0.696 for the non-DRIVE model, 0.628 for the initial model, and 0.382 for the DRIVE model. Sensitivity (recall, the proportion of actual positive instances correctly identified): 0.884 for the final model, 0.885 for the non-DRIVE model, 0.773 for the initial model, and 0.472 for the DRIVE model. Specificity (the proportion of actual negative instances correctly identified): 0.881 for the final model, 0.873 for the non-DRIVE model, 0.886 for the initial model, and 0.888 for the DRIVE model. With sensitivity and specificity, if the number of positive and negative examples is biased in one direction or the other, the poorer performing model will have a higher rate of biased positive or negative determinations, and may produce higher values than the better performing model. In fact, in the current case of classification training with a high percentage of negative examples relative to the total sample, specificity was higher with the initial and DRIVE models, which would not be able to correctly determine positive examples. Thus, sensitivity and specificity were not considered in our evaluation of segmentation performance.

The reason why we used six mouse images for the initial model was because we had these images before we started adding the mouse dataset to the training dataset for SA-UNet. Since we had six images on hand, we considered that using all of them would be more effective for training SA-UNet than using a smaller number of images. Then, using the initial model, we

performed vessel segmentation on 27 images. We noticed that some vessels were not recognized even with this model, as shown in the middle panel of Figure 2b. We considered that manual labeling of the vessels for all these 27 images would be too time consuming. Therefore, we implemented the CNN-corrected segmentation method for these images. We have added this information to the Methods section (lines 979–983). Thus, we did not examine the relationship between the number of training images and segmentation performance. In addition, considering the low bleeding probability in the subsequent mouse experiments using this model, we concluded that the vessel segmentation performance of the final model was sufficient for the current study. Therefore, no further images were added to the training dataset and the saturation point was not determined.

The mice that were used for SA-UNet training were different from those that were used in the subsequent mouse experiments. The same final SA-UNet model was also used for the marmoset experiment. Thus, new annotations were not needed when measuring a new animal, at least if it was a mouse or marmoset. However, as the reviewer pointed out, lighting conditions are important. If reflective points appeared on the cortical surface, the vessel segmentation tended to be incorrect. The direction and intensity of the light need to be adjusted so that a reflective point does not appear. We have noted this in the Methods section (997–1002).

Some more minor issues:

- L. 888: What is the `fitgeotrans` function? Which software? Matlab?

The function “`fitgeotrans`” is a function in the Image Processing Toolbox in MATLAB. It fits a geometric transformation to control point pairs. By fitting the specified transformation type to control points from moving and fixed images, it returns a geometric transformation object that can then be used for image registration and alignment tasks. We have added this information to the Methods section (lines 1019 and 1020).

- L. 108: Refer to infrared laser as "infrared laser sensor", as it is later (145) referred to as "laser".

We have unified this term to “laser distance sensor” throughout the manuscript.

Reviewer #5 (Remarks to the Author):

We thank Reviewer #5 for their careful consideration of our manuscript and for making helpful comments.

REVIEWER COMMENTS

Reviewer #2 (Remarks to the Author):

I have no further question for the revised manuscript.

Reviewer #3 (Remarks to the Author):

The authors designed an automated system that utilizes deep learning segmentation to identify cortical vessels and subsequently developed a robotic model to inject a viral solution into the cortical surface in mice and marmosets while avoiding cortical vessels. The performance was evaluated using genetically encoded calcium indicators, carried by an adeno-associated virus (AAV). The system (ARViS) performed sequential multipoint injections with a 50-micrometer accuracy and a bleeding risk of 0.1% per injection.

Overall, the authors present an intriguing model based on robust methodology and promising results. The proposed model has the potential to benefit the medical field both clinically and in research. Its direct implementation could automate and streamline research, particularly in studying functional relationships in the brain. Additionally, the automated segmentation model could provide neurosurgeons with real-time mapping of the cortex and vasculature, enhancing intraoperative decision-making.

Please find my comments below:

1. The deep-learning segmentation model was trained on the Digital Retinal Images for Vessel Extraction (DRIVE) dataset, which the authors have included in the supplementary material. Overall, the authors have done an excellent job of providing a very detailed explanation of the training process and the methods used. However, from the current version of the manuscript, it is slightly unclear what type and format the images employed in the training are and where they originated from. This information would be particularly helpful for the reader, especially when considering the translation of their findings to the clinical/neurosurgical context. This information is required to understand how the model can be implemented in neurosurgical practice and research and how their model compares to those currently reported in the literature(1).
2. Their model is based on the premise that their system, despite potentially higher costs, offers significant labor and time savings, along with high accuracy and low complication rates during multiple AAV microinjections, outperforming current manual methods. While the discussion

extensively covers the implications of their model, the benchmarking is somewhat indirect. It would be beneficial if the authors could clarify how ARViS directly compares to current practices and outline their vision for future development and potential clinical translation to the neurosurgical field.

3. The authors provide a detailed explanation for the characterization of bleeding (short < 30 seconds vs. long > 30 seconds) and how the probability risk was extrapolated. However, it is not clear how bleeding was determined in the first place. Was it direct visualization? What are the implications of their detection methods, and how sensitive were they? The authors performed histologic examinations of brain tissue damage from multiple local injections in two mice. However, would it be beneficial to include a histological examination for bleeding?

4. The authors performed one-photon and two-photon imaging of the marmoset cortex, revealing differential spatial representations of various movements across the cortex and multiple neurons. To underscore the significance of their findings from each imaging method, the authors should discuss the benefits of utilizing both one-photon and two-photon imaging.

5. In the last paragraph of the introduction, the authors state that they "...developed an Automated Robotic Virus injection System (ARViS) that enables bleed-free multi-site automated injections for accurate, fast, and dense virus delivery into living brains". They then describe their methodology but only specify the animal type in the final sentence. For improved clarity, it would be helpful to mention the animals used in the study throughout the last paragraph, and especially the last sentence.

References:

1. Moccia S, Foti S, Routray A, Prudente F, Perin A, Sekula RF, Mattos LS, Balzer JR, Fellows-Mayle W, De Momi E, Riviere CN. Toward Improving Safety in Neurosurgery with an Active Handheld Instrument. *Ann Biomed Eng.* 2018 Oct;46(10):1450-1464. doi: 10.1007/s10439-018-2091-x. Epub 2018 Jul 16. PMID: 30014286; PMCID: PMC6150797.

Reviewer #3 (Remarks on code availability):

The authors provide a commendable amount of details regarding their methodology, including the code data availability, and materials used for the development of ARViS. The details provided facilitate reproduction of their study. I did not try to download and run the code, but the information shared seems adequate for the reader to do so if interested.

Reviewer #4 (Remarks to the Author):

I thank the authors for careful consideration of my remarks on the first version of the paper.

All issues that I raised were convincingly answered and appropriately taken into account in the revised version.

I consider this revision as acceptable for publication.

Reviewer #5 (Remarks to the Author):

We thank all the reviewers for their careful consideration of our manuscript and for making helpful comments. Our detailed responses (in black) to Reviewer #3's comments (in blue) are provided below.

Reviewer #3 (Remarks to the Author):

The authors designed an automated system that utilizes deep learning segmentation to identify cortical vessels and subsequently developed a robotic model to inject a viral solution into the cortical surface in mice and marmosets while avoiding cortical vessels. The performance was evaluated using genetically encoded calcium indicators, carried by an adeno-associated virus (AAV). The system (ARViS) performed sequential multipoint injections with a 50-micrometer accuracy and a bleeding risk of 0.1% per injection.

Overall, the authors present an intriguing model based on robust methodology and promising results. The proposed model has the potential to benefit the medical field both clinically and in research. Its direct implementation could automate and streamline research, particularly in studying functional relationships in the brain. Additionally, the automated segmentation model could provide neurosurgeons with real-time mapping of the cortex and vasculature, enhancing intraoperative decision-making.

Please find my comments below:

1. The deep-learning segmentation model was trained on the Digital Retinal Images for Vessel Extraction (DRIVE) dataset, which the authors have included in the supplementary material. Overall, the authors have done an excellent job of providing a very detailed explanation of the training process and the methods used. However, from the current version of the manuscript, it is slightly unclear what type and format the images employed in the training are and where they originated from. This information would be particularly helpful for the reader, especially when considering the translation of their findings to the clinical/neurosurgical context. This information is required to understand how the model can be implemented in neurosurgical practice and research and how their model compares to those currently reported in the literature(1).

The DRIVE dataset contains 24-bit RGB images in Tag Image File Format (TIFF) as training images and 8-bit binary images in Portable Network Graphics (PNG) format as label images. These can be downloaded from <https://drive.grand-challenge.org/>. Our mouse images were aligned with the DRIVE dataset format. These images are available at

https://github.com/nomurshin/ARViS_Automated_Robotic_Virus_injection_System/tree/main/Data/Final_model_dataset. We have added this information to the Methods section (lines 800–805).

2. Their model is based on the premise that their system, despite potentially higher costs, offers significant labor and time savings, along with high accuracy and low complication rates during multiple AAV microinjections, outperforming current manual methods. While the discussion extensively covers the implications of their model, the benchmarking is somewhat indirect. It would be beneficial if the authors could clarify how ARViS directly compares to current practices and outline their vision for future development and potential clinical translation to the neurosurgical field.

Thank you for this important suggestion. We have added a table comparing ARViS to other neurosurgical techniques as Supplementary Table 2. We have also summarized the advantages and possible future applications of ARViS in Supplementary Fig. 11.

3. The authors provide a detailed explanation for the characterization of bleeding (short < 30 seconds vs. long > 30 seconds) and how the probability risk was extrapolated. However, it is not clear how bleeding was determined in the first place. Was it direct visualization? What are the implications of their detection methods, and how sensitive were they? The authors performed histologic examinations of brain tissue damage from multiple local injections in two mice. However, would it be beneficial to include a histological examination for bleeding?

During the multiple-injection procedure, the experimenter observed the cortical surface through Camera S in real-time. Long bleeding was clearly visible to the experimenter. The number of occurrences of short bleeding was counted by checking the video acquired by Camera S after the injection procedure had ended. It was reported that even when laser irradiation induces micro-bleeding with ~100- μ m diameter hematoma in the mouse cortex, which is comparable to the short bleeding in the current study, significant neural pathology does not occur, and the activity of neurons and astrocytes near the hematoma is almost restored within a day (Rosidi et al., PLoS One 6, e26612, 2011; Cianchetti et al., PLoS One 8, e65663, 2013). Therefore, we considered that some occurrences of short bleeding were acceptable and the effect of slight bleeding that we could not detect on the cortex was negligible. We have added this explanation and discussion to the Methods section (lines 994–1002).

4. The authors performed one-photon and two-photon imaging of the marmoset cortex, revealing differential spatial representations of various movements across the cortex and multiple neurons. To

underscore the significance of their findings from each imaging method, the authors should discuss the benefits of utilizing both one-photon and two-photon imaging.

We have added the following discussion (lines 496–504):

“Wide-field one-photon imaging can detect the dynamics of inter-areal interactions while two-photon imaging can detect the multicellular activity at the single-cell level. These methods can be applied to the same individual animal as demonstrated in the current study. Thus, once ARViS is used to express GECI or other probe over a broad region of the cerebral cortex, the strategy of clarifying the dynamics across multiple cortical areas and then revealing the multicellular activity at the single-cell level, especially in regions involved in those dynamics, would be beneficial for understanding cortical information processing across species”.

5. In the last paragraph of the introduction, the authors state that they “...developed an Automated Robotic Virus injection System (ARViS) that enables bleed-free multi-site automated injections for accurate, fast, and dense virus delivery into living brains”. They then describe their methodology but only specify the animal type in the final sentence. For improved clarity, it would be helpful to mention the animals used in the study throughout the last paragraph, and especially the last sentence.

In this paragraph, we have specified that this method was performed in the cerebral cortex of mice and a marmoset as follows (lines 84–94):

“In this study, we developed an Automated Robotic Virus injection System (ARViS) that enables bleed-free multi-site automated injections for accurate, fast, and dense virus delivery into the cerebral cortex of living mice and common marmoset. We solved the first problem of blood vessel avoidance by precisely measuring the three-dimensional (3D) structure of the cortical surface in these animals and applying deep learning-based image recognition of the vasculature structures with clustering-based optimization of multiple injection sites. We solved the second problem of perpendicular insertion by manipulating the micropipette with a six degrees-of-freedom robot under high-precision calibration. Bleeding probability after injection with ARViS was 0.1% in seven mice and 0% in one marmoset. To demonstrate the imaging of ARViS-assisted expression of the GECI, we imaged different neuronal representations of orofacial and body movements over a 14×7 mm area of the frontoparietal cortex in an awake adult marmoset”.

REVIEWERS' COMMENTS

Reviewer #3 (Remarks to the Author):

The authors have addressed all comments thoroughly. In its current state, the manuscript is suitable for publication.